# Adenosine signalling to astrocytes coordinates brain metabolism and function

Shefeeq M. Theparambil[1,2 ✉], Olga Kopach[3], Alice Braga[1], Shereen Nizari[1], Patrick S. Hosford[1], Virag Sagi-Kiss[4], Anna Hadjihambi[5], Christos Konstantinou[5], Noemi Esteras[3], Ana Gutierrez Del Arroyo[6], Gareth L. Ackland[6], Anja G. Teschemacher[7], Nicholas Dale[8], Tobias Eckle[9], Petros Andrikopoulos[10], Dmitri A. Rusakov[3], Sergey Kasparov[7] & Alexander V. Gourine[1 ✉]

Brain computation performed by billions of nerve cells relies on a sufficient and uninterrupted nutrient and oxygen supply[1,2]. Astrocytes, the ubiquitous glial neighbours of neurons, govern brain glucose uptake and metabolism[3,4], but the exact mechanisms of metabolic coupling between neurons and astrocytes that ensure on-demand support of neuronal energy needs are not fully understood[5,6]. Here we show, using experimental in vitro and in vivo animal models, that neuronal activity-dependent metabolic activation of astrocytes is mediated by neuromodulator adenosine acting on astrocytic A2B receptors. Stimulation of A2B receptors recruits the canonical cyclic adenosine 3′,5′-monophosphate–protein kinase A signalling pathway, leading to rapid activation of astrocyte glucose metabolism and the release of lactate, which supplements the extracellular pool of readily available energy substrates. Experimental mouse models involving conditional deletion of the gene encoding A2B receptors in astrocytes showed that adenosine-mediated metabolic signalling is essential for maintaining synaptic function, especially under conditions of high energy demand or reduced energy supply. Knockdown of A2B receptor expression in astrocytes led to a major reprogramming of brain energy metabolism, prevented synaptic plasticity in the hippocampus, severely impaired recognition memory and disrupted sleep. These data identify the adenosine A2B receptor as an astrocytic sensor of neuronal activity and show that cAMP signalling in astrocytes tunes brain energy metabolism to support its fundamental functions such as sleep and memory.

Brain neurons lack significant metabolic reserves and require a continuous supply of energy substrates. Astrocytes store chemical energy in the form of glycogen and respond to increases in the activity of neighbouring neurons with rapid activation of glucose metabolism[5,6]. Evidence exists that metabolic coupling between neurons and astrocytes is crucial for supporting the function of neural circuits that control core behaviours[4–9]. One of the defining features of metabolic activation of astrocytes is the increased production and release of lactate. Lactate supplements the extracellular pool of readily available energy substrates, and its local concentration rapidly increases in response to neuronal activity[10]. Significant experimental evidence suggests that the transfer of lactate from astrocytes to neurons is important for metabolic support of neuronal function[4,6,11,12]. However, it is not entirely clear how exactly astrocytes monitor the metabolic needs of neighbouring neurons, and which extracellular and intracellular signalling pathways control astrocyte glucose metabolism and ensure uninterrupted supply of chemical energy to support neuronal activity.

In peripheral tissues such as the liver and muscle, increased energy expenditure rapidly recruits intracellular stores of glucose via the actions of hormones like glucagon and catecholamines, and activation of the canonical cyclic adenosine 3′,5′-monophosphate (cAMP)–protein kinase A (PKA) signalling pathway[13]. Here we show that, in the brain, the activity of the same cAMP–PKA signalling pathway in astrocytes is regulated by adenosine and plays a major role in coordinating brain energy metabolism and function.

## cAMP signalling in astrocytes

Using the genetically encoded fluorescent cAMP sensor Epac-S[H187] (ref. 14) and the sensor of PKA activity AKAR4 (ref. 15) (Extended Data Fig. 1a,b), expressed under the control of glial fibrillary acidic protein (Gfap) promoter (Fig. 1a), we recorded robust increases in intracellular [cAMP] and PKA activity in astrocytes of the CA1 area of the rat hippocampus (acute and organotypic slice preparations) in response

[1]Centre for Cardiovascular and Metabolic Neuroscience, Neuroscience, Physiology and Pharmacology, University College London, London, UK. [2]Department of Biomedical and Life Sciences, Lancaster University, Lancaster, UK. [3]Institute of Neurology, University College London, London, UK. [4]Section of Bioanalytical Chemistry, Department of Metabolism, Digestion and Reproduction, Imperial College London, London, UK. [5]The Roger Williams Institute of Hepatology, Foundation for Liver Research & Faculty of Life Sciences and Medicine, King's College London, London, UK. [6]Translational Medicine and Therapeutics, William Harvey Research Institute, Queen Mary University of London, London, UK. [7]Physiology, Pharmacology, and Neuroscience, University of Bristol, Bristol, UK. [8]School of Life Sciences, University of Warwick, Coventry, UK. [9]Department of Anesthesiology, School of Medicine, University of Colorado Anschutz Medical Campus, Aurora, CO, USA. [10]Section of Biomolecular Medicine, Department of Metabolism, Digestion and Reproduction, Imperial College London, London, UK. ✉e-mail: s.theparambil@ucl.ac.uk; a.gourine@ucl.ac.uk

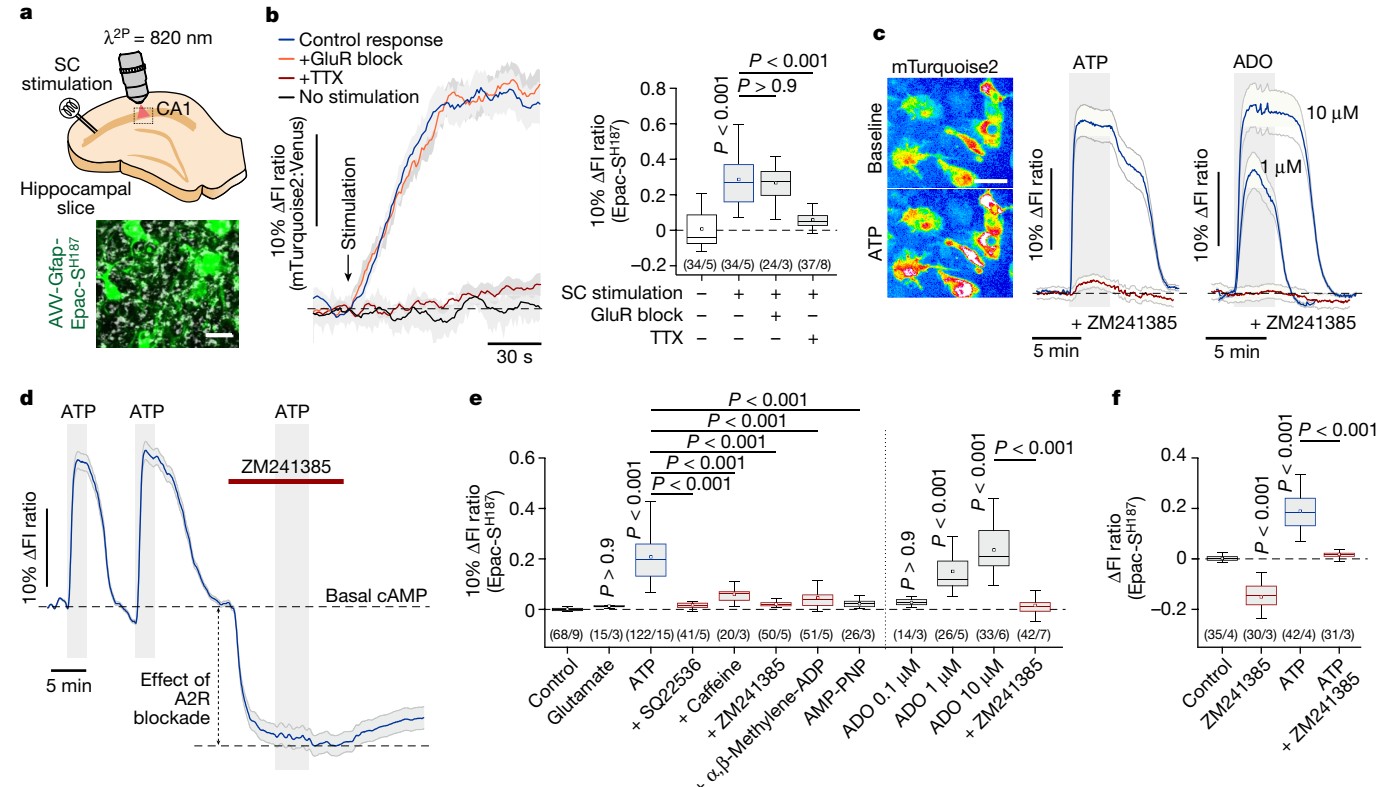

**Fig. 1 | Neuronal activity recruits cAMP–PKA signalling in astrocytes.**
**a**, Two-photon (2P) optical recordings of neuronal activity-induced cAMP responses in astrocytes transduced to express Epac-S$^{H187}$ in hippocampal slices. SC, Schaffer collateral fibres. Scale bar, 20 μm. **b**, Representative traces and summary data illustrating changes in intracellular [cAMP] ([cAMP]$_i$; Epac-S$^{H187}$ fluorescence intensity (FI) ratio of mTurquoise2:Venus) in astrocytes of the CA1 area induced by stimulation of Schaffer collateral fibres (burst of 5 pulses at 20 Hz) in control conditions and under conditions of glutamate receptor (GluR) blockade, or in the presence of tetrodotoxin (TTX). **c**, Representative astrocyte cAMP responses induced by ATP, ATP in the presence of the adenosine A2 receptor antagonist ZM241385, adenosine (ADO) and adenosine in the presence of ZM241385. Images illustrate changes in the Epac-S$^{H187}$ sensor mTurquoise2 fluorescence in astrocytes in response to ATP. In this sensor, cAMP binding increases mTurquoise2 fluorescence. Scale bar, 20 μm. **d**, Representative trace showing a significant decrease in the Epac-S$^{H187}$ sensor signal in hippocampal astrocytes induced by adenosine A2 receptor (A2R) blockade, indicative of a

reduction in basal [cAMP]$_i$. **e**, Summary data showing peak changes in [cAMP]$_i$ (Epac-S$^{H187}$ FI ratio) in astrocytes induced by glutamate, ATP, AMP-PNP and ATP in the presence of the adenylyl cyclase inhibitor SQ22536, the ecto-5′-nucleotidase inhibitor α,β-methylene-ADP, the adenosine receptor inhibitor caffeine or ZM241385. Also shown are peak cAMP responses induced by adenosine. **f**, Summary data showing changes in [cAMP]$_i$ in astrocytes in response to ZM241385, ATP or ATP in the presence of ZM241385, recorded in hippocampal slices. In the box-and-whisker plots, the central dot indicates the mean, the central line indicates the median, the box limits indicate the upper and lower quartiles, and the whiskers extend to 1.5× the interquartile range from the quartiles. In panels **b**–**d**, traces show averaged (mean ± s.e.m.) recordings from several individual cells in a representative experiment. In panels **b**,**e**,**f**, the numbers in parentheses indicate the number of individual cells/number of separate slices or cultures prepared from the same number of animals. $P$ values were determined by one-way analysis of variance (ANOVA) followed by Sidak's post-hoc test.

to stimulation of Schaffer collateral fibres (Fig. 1b and Extended Data Fig. 1c–e), supporting the results of previously published studies[16]. Blockade of glutamate receptors (10 μM CPP, 20 μM NBQX and 200 μM MCPG) did not prevent these cAMP and PKA responses (Fig. 1b and Extended Data Fig. 1e), suggesting that increased neuronal activity recruits the cAMP–PKA pathway in astrocytes via signals other than glutamate.

There is significant evidence that in the brain, increased neuronal activity is associated with the release of purines into the extracellular space[17–22]. We next found that astrocytes in culture and brain slices respond to the purine nucleotides ATP (30 μM) and ADP (30 μM) (but not to glutamate (100 μM)) with elevations in intracellular [cAMP] and PKA activity (Fig. 1c–f and Extended Data Figs. 1a,b and 2a–e). As expected, strong [cAMP] increases in astrocytes were induced by the specific adenylyl cyclase activator forskolin (5 μM; Extended Data Fig. 2a,d). ATP-induced cAMP responses were significantly reduced by the adenylyl cyclase inhibitor SQ22536 (100 μM; Fig. 1e and Extended Data Fig. 2b,d), but were not affected by blockade of Ca$^{2+}$ signalling (in zero Ca$^{2+}$ and 1 μM thapsigargin), inhibition of ionotropic P2X (with PPADS (100 μM)) or metabotropic P2Y$_1$

(with MRS2179 (20 μM) or MRS2500 (2 μM)) receptors for ATP (Extended Data Fig. 2c,d). cAMP responses induced by a non-hydrolysable ATP analogue AMP-PNP (10 μM) were much smaller than the responses induced by ATP applied in the same concentration (Fig. 1e and Extended Data Fig. 3a). Moreover, the effects of ATP on astrocyte [cAMP] were significantly reduced or abolished by the ecto-5′-nucleotidase inhibitor α,β-methylene-ADP (200 μM), the adenosine receptor inhibitor caffeine (1 mM) or by the more specific adenosine A2 receptor antagonist ZM241385 (10 μM; Fig. 1c–f and Extended Data Fig. 3b).

Collectively, these data suggested that the effect of ATP on cAMP–PKA signalling in astrocytes is indirect, independent of intracellular Ca$^{2+}$ signalling, and mediated by adenosine formed in the extracellular space following ATP breakdown by ectonucleotidase activity[20]. Indeed, adenosine (1–10 μM) triggered robust increases in intracellular [cAMP] and PKA activity in astrocytes (often leading to saturation of the Epac-S$^{H187}$ and AKAR4 sensors when applied at higher concentrations) (Fig. 1c,e and Extended Data Fig. 3c,d). The A2 receptor antagonist ZM241385 (10 μM) completely blocked the effects of adenosine (Fig. 1c,e and Extended Data Fig. 3c). Furthermore, adenosine-induced PKA responses

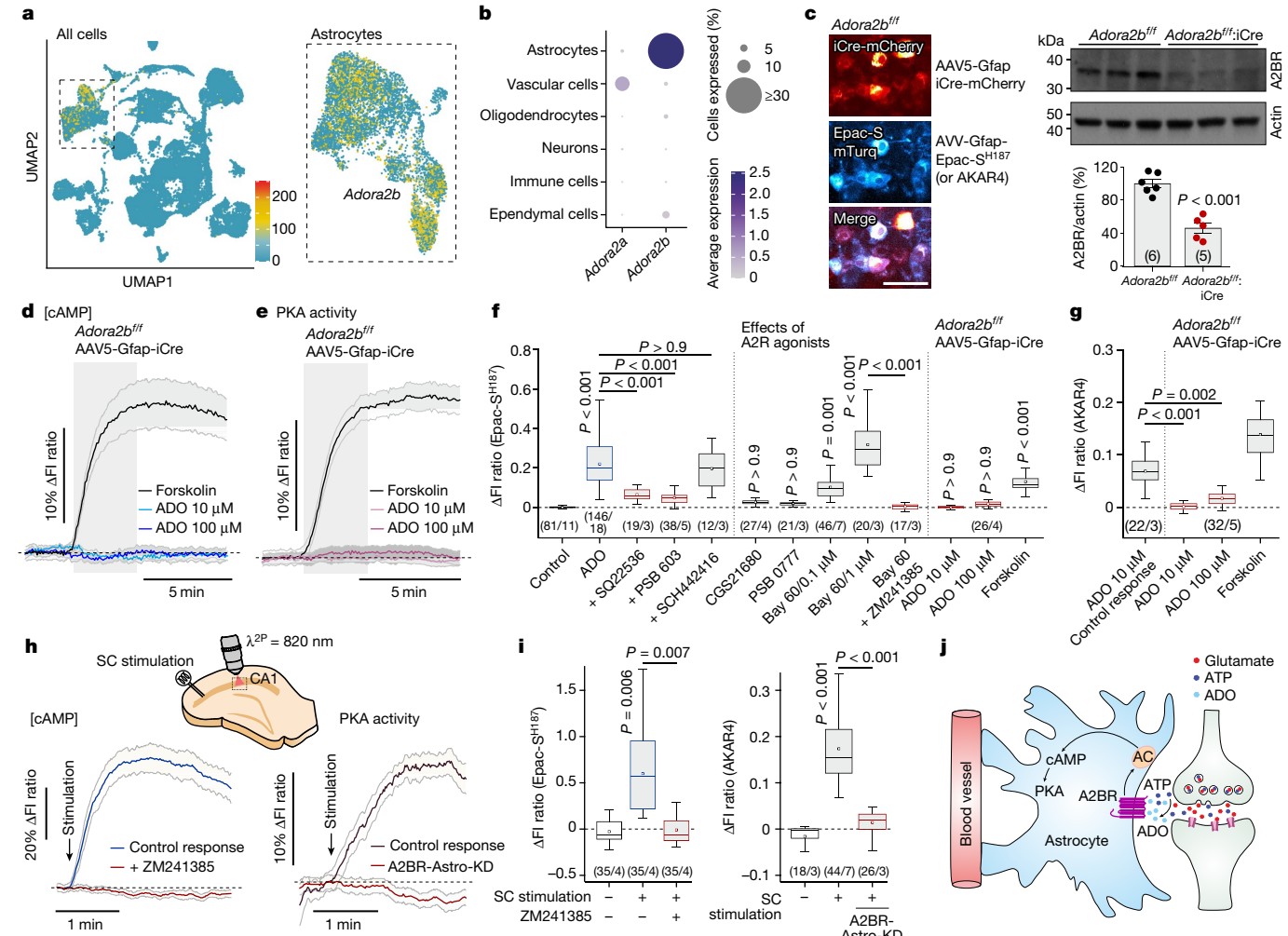

**Fig. 2 | Neuronal activity-dependent recruitment of cAMP–PKA signalling in astrocytes is mediated by adenosine A2B receptors. a**, Visualization of single-cell RNA-seq data from the mouse brain[24] using uniform manifold approximation and projection (UMAP), illustrating specific expression of *Adora2b* in astrocytes. The expression level of *Adora2b* is represented by the colour scale (counts per cell). **b**, The percentage of brain cells (identified by characteristic marker genes) that express *Adora2a* and *Adora2b* is represented by the size of the circle and the relative expression is represented by the colour gradient. **c**, A2B receptor (A2BR) knockdown in astrocytes (A2BR-Astro-KD) of *Adora2b flox/flox* (*Adora2b f/f*) mice transduced to express iCre recombinase. Data are shown as individual values and mean ± s.e.m. *P* value was determined by two-tailed Student's *t*-test. For gel source data, see Supplementary Fig. 1. mTurq, mTurquoise2. Scale bar, 50 µm. **d,e**, A2BR knockdown prevented cAMP and PKA responses to adenosine, whereas the effects of the direct adenylyl cyclase activator forskolin were not affected. **f**, Summary data showing peak astrocyte cAMP responses induced by adenosine; adenosine in the presence of SQ22536, PSB 603 (an A2BR antagonist) or SCH442416 (an A2A receptor

antagonist); A2A receptor agonists CGS21680 or PSB 0777; and the A2BR agonist BAY 60-6583 or BAY 60-6583 in the presence of ZM241385. Also shown are peak cAMP responses induced by adenosine or forskolin in conditions of A2BR deletion. **g**, Changes in PKA activity in astrocytes in response to adenosine or forskolin in conditions of A2BR deletion. **h,i**, Representative traces (**h**) and summary data (**i**) illustrating changes in [cAMP]ᵢ and PKA activity in astrocytes of the CA1 area induced by Schaffer collateral fibre stimulation in control conditions and under conditions of pharmacological or genetic A2BR blockade. **j**, Schematic of neuronal activity-dependent recruitment of cAMP–PKA signalling in astrocytes, mediated by adenosine and A2B receptors. AC, adenylyl cyclase. In panels **d,e,h**, traces show averaged (mean ± s.e.m.) recordings from several individual cells in a representative experiment. In panels **c,f,g,i**, the numbers in parentheses indicate the number of individual cells or number of separate cultures/slices prepared from the same number of animals. *P* values were determined by one-way ANOVA followed by Sidak's or Dunnett's post-hoc test.

were prevented by the PKA inhibitor H89 (10 µM; Extended Data Fig. 3c). In slice experiments, we observed that application of ZM241385 led to a significant reduction of the Epac-S^H187 sensor signal (Fig. 1d,f), suggesting that the adenosine A2 receptor-mediated cAMP signalling pathway in astrocytes is constitutively active.

## Adenosine A2B receptors in astrocytes

Vertebrates express two Gₛ protein-coupled adenosine receptors that stimulate adenylyl cyclase and activate cAMP signalling pathways. These are A2A and A2B receptors, encoded by two paralogous genes: *Adora2a* and *Adora2b*[23]. Analysis of single-cell RNA sequencing

(RNA-seq) data from the mouse brain[24] demonstrated strong and specific expression of *Adora2b* in astrocytes (Fig. 2a,b), consistent with the recently published evidence placing the A2B receptor among the most highly expressed G protein-coupled receptors in astrocytes from all regions of the brain[25]. The RNA-seq data suggested that astrocytes do not express *Adora2a* (Fig. 2b). This conclusion was further supported by the results of the experiments showing that adenosine-induced cAMP responses in astrocytes were effectively blocked by the selective A2B antagonist PSB 603 (10 µM), but were unaffected by the A2A antagonist SCH442416 (10 µM; Fig. 2f and Extended Data Fig. 3d). Moreover, the effects of adenosine on intracellular [cAMP] in astrocytes were mimicked by the A2B receptor agonist BAY 60-6583 (0.1–1 µM), whereas

the A2A agonists CGS21680 (1 µM) and PSB 0777 (1 µM) had no effect (Fig. 2f and Extended Data Fig. 3e). In astrocytes of floxed *Adora2b* mice (*Adora2b^flox/flox*)[26] transduced with AAV5-Gfap-iCre-mCherry vector to express the improved Cre (iCre) recombinase and delete A2B receptors (Fig. 2c), adenosine (up to 100 µM) had no effect on intracellular [cAMP] and PKA activity, whereas cAMP and PKA responses induced by direct adenylyl cyclase activation with forskolin (10 µM) were unaffected (Fig. 2d–g).

We next found that increases in intracellular [cAMP] and PKA activity in CA1 astrocytes evoked by stimulation of Schaffer collateral fibres were blocked by inhibition of A2 receptors with ZM241385 or upon genetic deletion of A2B receptors (Fig. 2h,i). Collectively, these results suggested that adenosine acting via A2B receptors mediates synaptic activity-dependent recruitment of cAMP–PKA signalling in astrocytes (Fig. 2j). We next sought to investigate the significance of this signalling pathway in the regulation of astrocyte glucose metabolism.

## Adenosine signalling and brain metabolism

Using the genetically encoded fluorescent sensor of glucose FLIP[12]glu-700μΔ6 (ref. 27) and the sensor of cytosolic NADH–NAD⁺ redox state Peredox[28], we recorded robust increases in glucose consumption and glycolytic rate in astrocytes in response to ATP and adenosine (Fig. 3a–d and Extended Data Fig. 4a). The A2B receptor agonist BAY 60-6583 (1 µM) increased glucose consumption and glycolytic rate in astrocytes and this effect was prevented by A2B receptor deletion (Fig. 3a and Extended Data Fig. 4b). Increases in astrocyte glucose consumption and the rate of glycolysis would be expected to facilitate the production of lactate and its release into the extracellular space. We next used enzymatic microelectrode biosensors to record the release of lactate in acute slices of the rat brain. Both ATP and adenosine triggered significant lactate release (Extended Data Fig. 5a,b). ATP-induced lactate release was not affected by the ATP receptor antagonist PPADS (100 µM), but was blocked by ZM241385 (10 µM; Extended Data Fig. 5b). The effect of adenosine on lactate release was mimicked by the A2B receptor agonist BAY 60-6583 (Extended Data Fig. 5f) and blocked by ZM241385 (10 µM), PKA inhibition with H89 (10 µM) or inhibition of glycogen metabolism with 1,4-dideoxy-1,4-imino-D-arabinitol (DAB; 200 µM; Extended Data Fig. 5b). The A1 adenosine receptor antagonist DPCPX (1 µM) had no effect on adenosine-induced lactate release (Extended Data Fig. 5b).

In studies involving simultaneous adenosine and lactate biosensor recordings in acute brain slices, we measured robust release of both adenosine and lactate in response to AMPA (5 µM; Extended Data Fig. 5c,e). The application of AMPA was used in these experiments as a model of generalized increases in neuronal activity. AMPA-induced lactate release was significantly reduced by ZM241385 (Fig. 3e and Extended Data Fig. 5d,e), suggesting that it is largely driven by the actions of adenosine. In these experiments, we observed that A2B receptor blockade had a major effect on the basal level of lactate release (Fig. 3e,f). Inhibition of lactate dehydrogenase with oxamate (10 mM) reduced basal lactate release by approximately 90%, whereas blockade of A2B receptors with PSB 603 (10 µM) or ZM241385 (10 µM) reduced lactate release by 60–70% (Fig. 3f). These data suggested that A2B receptor-mediated signalling controls both basal and stimulated astrocyte glucose metabolism and prompted us to investigate the effect of astrocyte-specific A2B receptor deletion on the brain metabolome.

To produce animals with conditional deletion of A2B receptors in astrocytes, *Adora2b^flox/flox* mice were crossed with *Aldh1l1^Cre/ERT2+/−* mice[29] (Fig. 3g). Tamoxifen treatment of *Adora2b^flox/flox*:*Aldh1l1^Cre/ERT2* mice reduced the brain A2B transcript level by 60% (Fig. 3g) and the basal level of lactate release (recorded in acute brain slices) by 41% (Fig. 3h). Biosensor recordings also confirmed effective deletion of A2B receptors in this model, as the A2B agonist BAY 60-6583 (1 µM)

failed to trigger lactate release in brain slices of tamoxifen-treated *Adora2b^flox/flox*:*Aldh1l1^Cre/ERT2* mice (Extended Data Fig. 5f). To understand how A2B receptor deletion in astrocytes impacts global brain energy metabolism, we performed targeted brain tissue metabolomics focusing primarily on metabolites of the central carbon metabolism (Supplementary Table 1). A cross-validated model built on brain metabolic features using partial least squares-discriminant analysis (PLS-DA) significantly predicted variance associated with A2B receptor deletion through a permutation test (Fig. 3j), indicating that A2B receptor-mediated signalling in astrocytes has a global effect on brain metabolism. The loading plot of the PLS-DA model (Fig. 3k) showed that cAMP was the most prominent analyte driving the separation between the two experimental groups. The levels of cAMP were depleted in the brains of tamoxifen-treated *Adora2b^flox/flox*:*Aldh1l1^Cre/ERT2* mice (Fig. 3i), suggesting that the major pool of brain cAMP is maintained by the activity of A2B receptors in astrocytes. Metabolic pathway enrichment analysis showed that the deletion of A2B receptors in astrocytes results in brain metabolic reprogramming with the most significantly downregulated processes, including the citric acid cycle, ketone body metabolism and the Warburg effect (Fig. 3l). The Warburg effect, defined as an increase in the rate of glucose uptake and preferential production of lactate, is a characteristic feature of astrocyte glucose metabolism[30]. This analysis supported the hypothesis that A2B receptor-mediated signalling controls astrocyte glucose metabolism and is essential to maintaining global brain metabolic activity.

## Astrocyte A2B receptors and synaptic function

We next found that A2B receptor-mediated signalling in astrocytes is important for metabolic support of synaptic activity in the brain. In experiments conducted in acute hippocampal slices, we performed recordings of field excitatory postsynaptic potentials (fEPSPs) in the CA1 area evoked by stimulation of Schaffer collateral fibres (Extended Data Fig. 6). Under control conditions, synaptic activity was maintained throughout the duration of the experiment, lasting for up to 1.5 h (Extended Data Fig. 6). When A2B receptors were blocked with PSB 603 (10 µM), a progressive reduction of the fEPSP slope starting from approximately 45 min after the application of the drug was observed (Extended Data Fig. 6). Supplemental lactate (5 mM) preserved the efficacy of synaptic transmission under conditions of A2B receptor blockade (Extended Data Fig. 6).

The neural circuits of the hippocampus have a crucial role in learning and memory[31]. We next deleted A2B receptors specifically in hippocampal astrocytes to determine the significance of the identified signalling pathway in the mechanisms of activity-dependent synaptic plasticity – the defining feature of neural circuits underlying information processing and storage in the brain. Hippocampal astrocytes of *Adora2b^flox/flox* mice were transduced to express iCre recombinase following microinjections of AAV5-Gfap-eGFP-iCre vector (Fig. 4a). Transduction of astrocytes with AAV5-Gfap-tdTomato vector was used as a control. The specificity of this approach in targeting astrocytes was confirmed by quantification of cells co-expressing eGFP with astrocytic (GFAP), microglial (Iba1), neuronal (NeuN) or oligodendrocytic (MBP) markers. The majority (85%) of transduced cells were found to be GFAP-positive hippocampal astrocytes (Extended Data Fig. 7a). Expression of iCre recombinase in astrocytes of *Adora2b^flox/flox* mice decreased A2B receptor mRNA and protein levels (Extended Data Fig. 7b), and reduced the release of lactate induced by adenosine (100 µM) or AMPA (5 µM) in acute hippocampal slices (Fig. 4b,c).

The next experiments showed that signalling via astrocyte A2B receptors is essential for maintaining synaptic function under conditions of acute metabolic strain. In the experiments performed in hippocampal slices (Fig. 4d), we observed that synaptic activity (assessed by measuring the slope of fEPSP evoked by stimulation of Schaffer collateral fibres) in the CA1 area was well maintained when the concentration

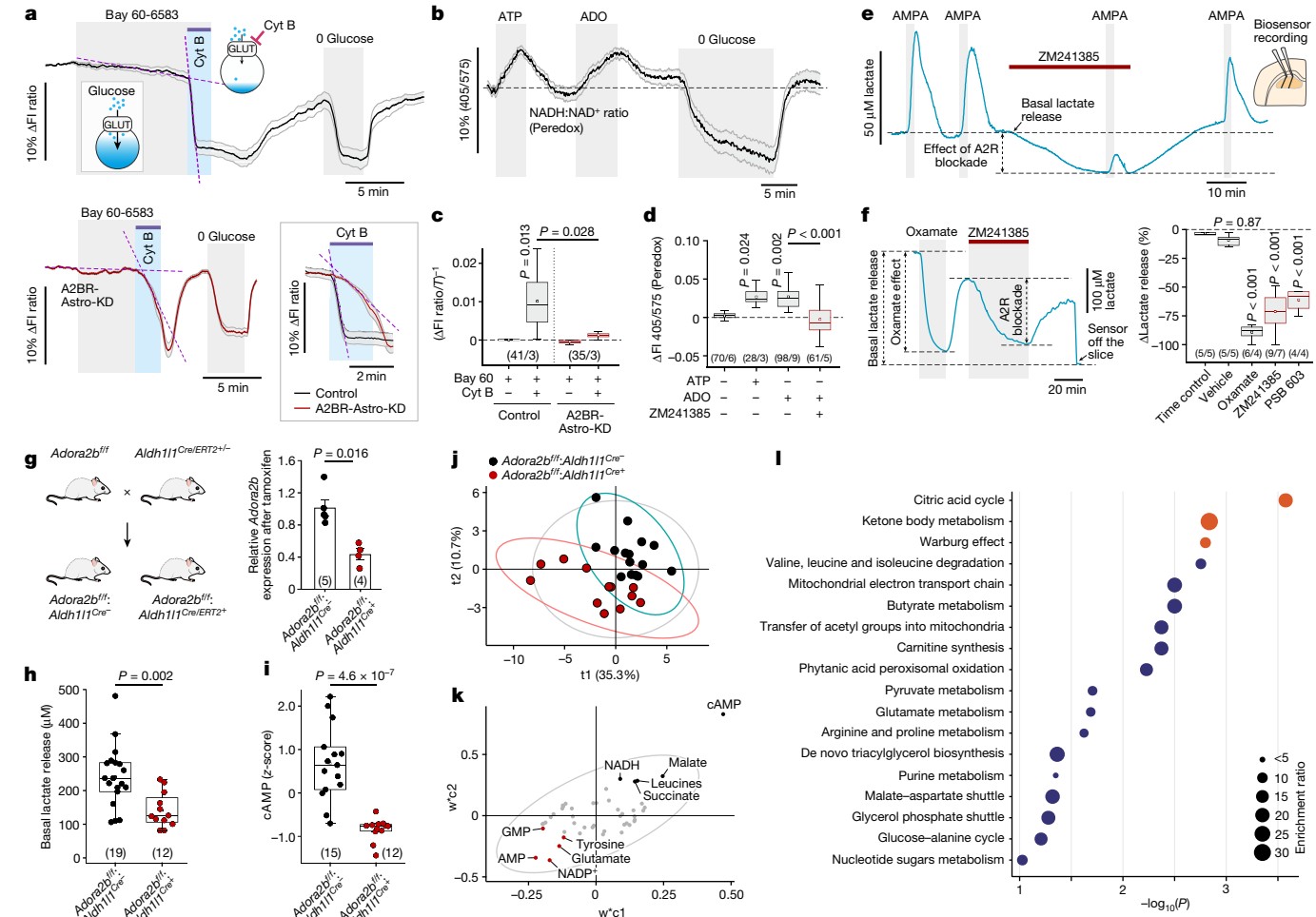

**Fig. 3 | Adenosine A2B receptor-mediated cAMP signalling in astrocytes regulates brain glucose metabolism. a,c**, Representative traces (**a**) and summary data (**c**) showing the effect of A2BR activation (with Bay 60-6583) followed by blockade of glucose transport with cytochalasin B (CytB; 20 μM) on changes in intracellular [glucose] (reporting the glycolytic rate), recorded using the FLIP[12]glu-700μΔ6 sensor in wild-type astrocytes and astrocytes with genetic A2BR deletion. Superimposed expanded traces show changes in astrocyte intracellular glucose concentration following application of CytB in the presence of Bay 60-6583 (inset). The slope of the sensor signal decline under conditions of glucose transport blockade was used to calculate the glycolytic rate, which was markedly reduced by A2BR deletion. **b,d**, Representative trace (**b**) and summary data (**d**) showing ATP-induced and adenosine-induced changes in the astrocyte cytosolic NADH–NAD+ redox state (reporting glucose consumption) recorded using the Peredox sensor. In panels **a,b**, traces show averaged (mean ± s.e.m.) recordings from several individual cells in a representative experiment. In panels **c,d**, the numbers in parentheses indicate the number of individual cells/number of separate cultures prepared from the same number of animals. P values were determined by one-way ANOVA followed by Sidak's post-hoc test. **e**, Representative trace showing the release of lactate in response to AMPA in the absence and presence of ZM241385, recorded using microelectrode biosensors in acute brain slices. **f**, Representative trace and summary data showing the effects of lactate

dehydrogenase inhibition with oxamate or A2BR blockade on the basal level of lactate release recorded in brain slices. The numbers in parentheses indicate the number of independent slice experiments/number of animals per experimental group. P values were determined by ANOVA followed by Dunnett's post-hoc test. **g**, *Adora2b[flox/flox]* mice were crossed with *Aldh1l1[Cre/ERT2+/−]* mice to produce animals with conditional deletion of A2BRs in brain astrocytes after tamoxifen treatment. Data are shown as individual values and mean ± s.e.m. **h**, A2BR deletion in astrocytes reduced the basal release of lactate recorded in brain slices. **i**, A2BR deletion in astrocytes depleted brain cAMP. In panels **g–i**, the numbers in parentheses indicate the numbers of animals per experimental group. P values were determined by two-tailed Mann–Whitney test. **j**, Multivariate analysis of brain metabolites by partial least squares-discriminant analysis (PLS-DA) showing a clear separation between the groups after sevenfold cross-validation for 10,000 iterations (pQ2 = 0.05). Ellipses denote 95% confidence intervals. **k**, Loading plot for PLS-DA in panel **j** showing the top five metabolites most enriched in the brains of tamoxifen-treated *Adora2b[flox/flox]:Aldh1l1[Cre−]* or *Adora2b[flox/flox]: Aldh1l1[Cre/ERT2+]* mice (black and red symbols, respectively). **l**, Illustration of the pathway enrichment analysis predicting metabolic pathways or processes that are more active in the brains of control animals versus the brains of mice with conditional deletion of A2BRs in astrocytes. The most significantly affected processes are highlighted in orange.

of glucose in the media was lowered from 10 mM to 2 mM (Fig. 4e,f). Genetic A2B receptor knockdown in astrocytes or pharmacological inhibition of A2B receptors (with PSB 603 (10 μM)) exposed the energetic vulnerability of excitatory transmission in the hippocampus. When A2B receptor-mediated signalling was blocked, lowering the concentration of extracellular glucose to 2 mM abolished the synaptic activity within 30 min (Fig. 4e,f). In slices of mice with A2B receptor deletion in astrocytes and under conditions of 2 mM extracellular glucose, the

efficacy of excitatory synaptic transmission was partially restored by supplemental lactate (5 mM; Fig. 4e,f).

Using this experimental model, we next found that signalling via astrocyte A2B receptors is also essential for synaptic plasticity in the hippocampus. We applied a classical long-term potentiation (LTP) induction protocol involving high-frequency stimulation of Schaffer collateral fibres with the recordings of evoked fEPSPs in the CA1 area[32]. The profiles of LTP were similar in hippocampi of two control groups

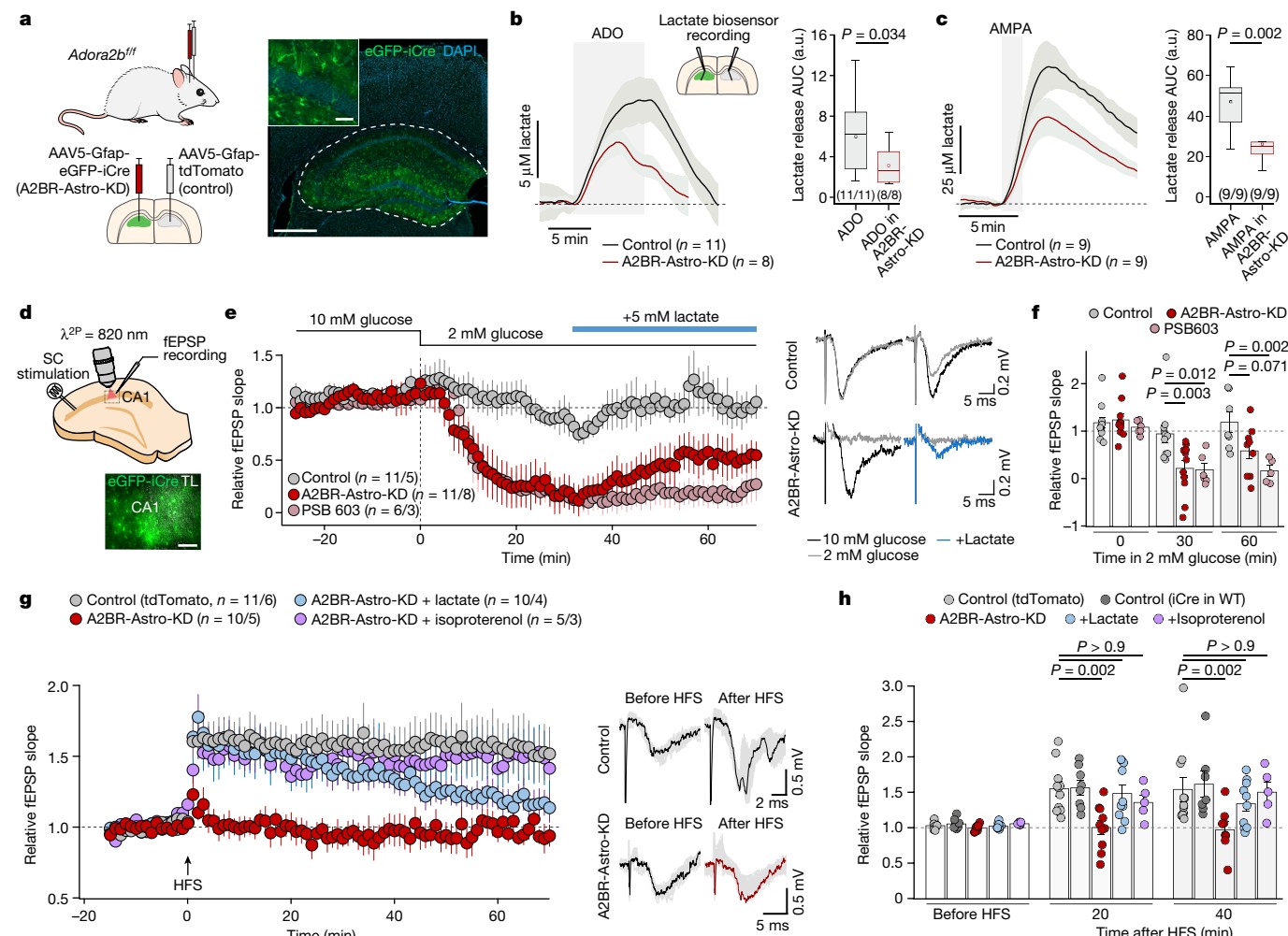

**Fig. 4 | Adenosine A2B receptor-mediated signalling in astrocytes is essential for metabolic support of synaptic activity and plasticity.**
**a**, Hippocampal astrocytes of *Adora2b^flox/flox* mice transduced to express iCre recombinase after the microinjections of AAV5-Gfap-eGFP-iCre vector. In control experiments, astrocytes were targeted with AAV5-Gfap-tdTomato vector. Scale bars, 500 µm and 50 µm (inset). **b,c**, A2BR knockdown in hippocampal astrocytes reduced the release of lactate induced by adenosine or increased neuronal network activity stimulated by AMPA (acute brain slices). Averaged trace data are shown as mean ± s.e.m. *P* values were determined by two-tailed Student's *t*-test. a.u., arbitrary units; AUC, area under the curve. **d**, CA1 astrocytes of *Adora2b^flox/flox* mice transduced to express iCre. TL, transmitted light. Scale bar, 100 µm. **e,f**, Time course (mean ± s.e.m.) (**e**) and summary data (individual values and mean ± s.e.m.) (**f**) of relative changes in the averaged field excitatory postsynaptic potential (fEPSP; representative examples are illustrated) slope recorded in the CA1 area in response to Schaffer collateral fibre stimulation, showing the effect of reduced glucose availability

on excitatory transmission under conditions of pharmacological (PSB 603) or genetic blockade of A2BR-mediated signalling. In A2BR-Astro-KD preparations, synaptic transmission was partially restored by supplemental lactate (5 mM). **g,h**, Time course (mean ± s.e.m.) (**g**) and summary data (individual values and mean ± s.e.m.) (**h**) of relative changes in the averaged fEPSP slope recorded in the CA1 area before and after induction of LTP by high-frequency Schaffer collateral fibre stimulation (HFS; 100 pulses at 100 Hz repeated 3 times with 60-s intervals) in hippocampal slices of *Adora2b^flox/flox* mice transduced to express tdTomato in astrocytes (control), wild-type (WT) mice transduced to express iCre (control; *n* = 8/3), *Adora2b^flox/flox* mice transduced to express iCre in astrocytes (A2BR-Astro-KD) and *Adora2b^flox/flox* mice transduced to express iCre and in the presence of supplemental lactate or the β-adrenoceptor agonist isoproterenol. In panels **b,c,e,g**, the numbers in parentheses indicate the number of independent slice experiments/number of animals per experimental group. *P* values were determined by one-way ANOVA followed by Sidak's post hoc test.

of mice: *Adora2b^flox/flox* transduced to express tdTomato and C57BL/6 wild-type mice transduced to express iCre recombinase in astrocytes (Fig. 4g,h). By contrast, high-frequency stimulation of Shaffer collateral fibres induced no LTP in hippocampal slices of animals with astrocyte-specific A2B receptor deletion (Fig. 4g,h). Supplemental lactate (5 mM) partially rescued LTP in these preparations (Fig. 4g,h). We then hypothesized that stimulation of other astroglial G_s-coupled receptors, such as β-adrenoceptors[33], could potentially support LTP in conditions of A2B receptor deficiency. Indeed, in slice preparations of mice with A2B receptor knockdown in hippocampal astrocytes, LTP induction was rescued by the β-adrenoceptor agonist isoproterenol (5 µM; Fig. 4g,h). The shorter duration of the effect of supplemental lactate than the effect of β-adrenoceptor stimulation suggested that,

although lactate signalling is important for the induction of LTP, it operates in concert with other cAMP-dependent astroglial mechanisms to ensure synaptic plasticity in the hippocampus.

## Astrocyte A2B receptors, sleep and memory

As synaptic plasticity is the key mechanism responsible for changes in brain function in response to experience, we hypothesized that deletion of A2B receptors in astrocytes would impair the brain mechanisms of learning and memory. *Adora2b^flox/flox* mice were transduced to express iCre recombinase (experimental group) or tdTomato (controls) in hippocampal astrocytes bilaterally (Fig. 5a), and the animals underwent memory testing using the object recognition test[34] (Fig. 5b).

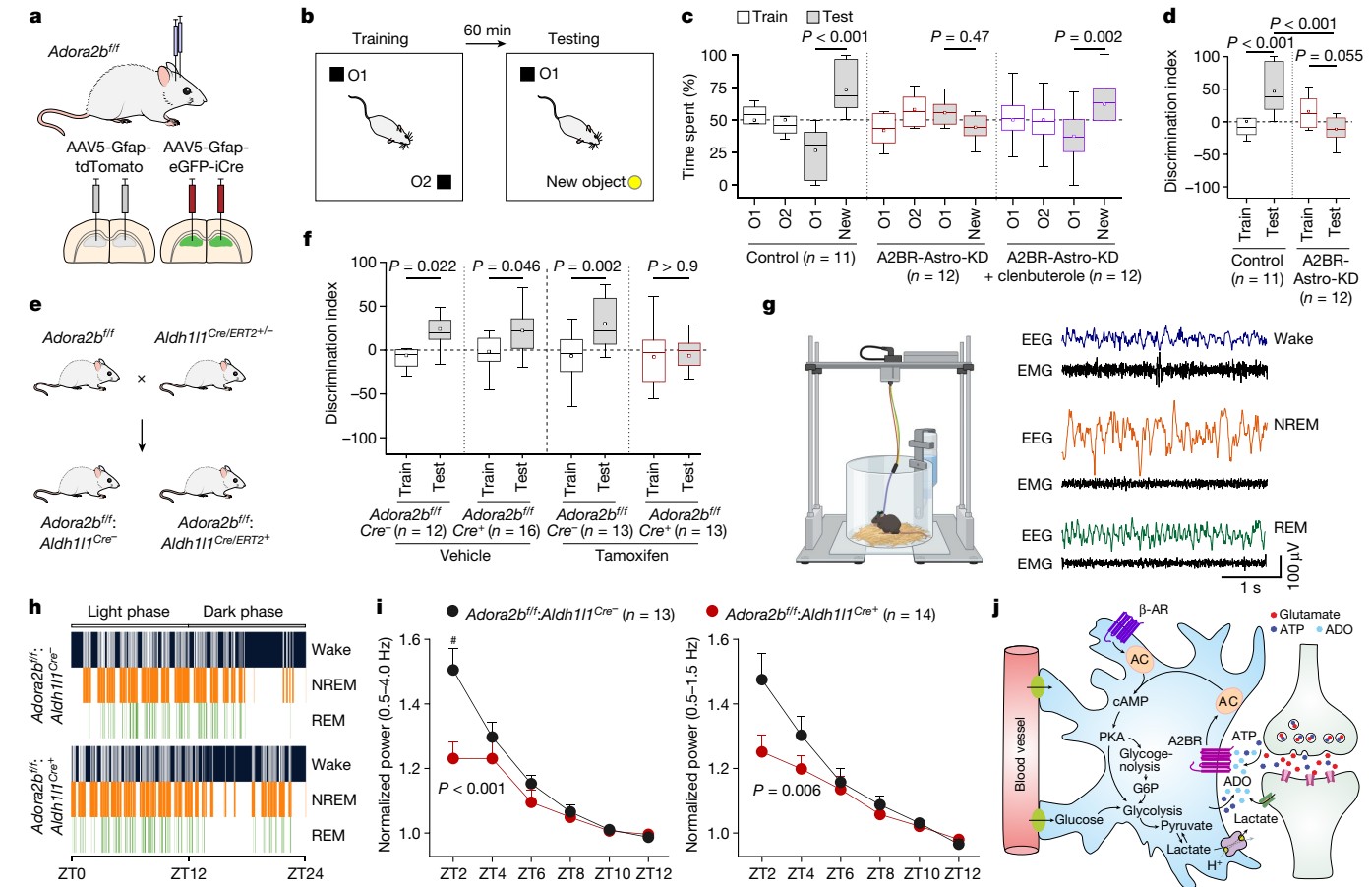

**Fig. 5 | Adenosine A2B receptor-mediated signalling in astrocytes is essential for recognition memory and accumulation of sleep pressure.**
**a**, Bilateral targeting of hippocampal astrocytes of *Adora2b*^*flox/flox*^ mice to express iCre recombinase (A2BR-Astro-KD) or reporter protein (tdTomato; control).
**b**, Schematic of the object recognition test. **c**, Summary data showing the time the animals spent exploring two identical objects (O1 and O2) during training and then one familiar (O1) and a new object (new) during testing. **d**, Calculated discrimination index as a measure of relative object preference, showing that A2BR deletion in hippocampal astrocytes impaired the recognition memory.
**e**, Generation of mice with conditional deletion of A2BR in astrocytes. **f**, Summary data showing the discrimination index calculated for *Adora2b*^*flox/flox*^:*Aldh1l1*^*Cre−*^ and *Adora2b*^*flox/flox*^:*Aldh1l1*^*Cre/ERT2+*^ mice treated with vehicle (oil) or tamoxifen. Deletion of A2BR in astrocytes impaired the recognition memory. **g**, Schematic of the EEG and EMG recording setup and representative EEG and EMG traces.
**h**, Representative hypnograms obtained by 24 h of EEG recordings showing

fragmentation of wake and NREM sleep during the light phase in mice with conditional deletion of A2BR in brain astrocytes. ZT, Zeitgeber time. Schematic in panel **g** was created using BioRender (https://biorender.com). **i**, A2BR deletion in astrocytes markedly decreased the slow-wave activity (0.5–4.0 Hz) and low-frequency slow-wave activity (0.5–1.5 Hz), indicative of a reduction in sleep pressure. Data are presented as mean ± s.e.m. ^#^$P = 0.010$; $P$ values indicated signify the differences between genotypes. **j**, Schematic illustrating the proposed mechanism that controls astrocyte glucose metabolism and provides metabolic support of neuronal activity. Potential sources of activity-dependent adenosine release are illustrated, including transporter-mediated release[19,22], and extracellular breakdown of ATP[20] released at synapses[18] and by astrocytes[19]. β-AR, β-adrenoceptor; G6P, glucose-6-phosphate. In panels **c,d,f,i**, the numbers in parentheses indicate the numbers of animals per experimental group. $P$ values were determined by one-way ANOVA followed by Sidak's post-hoc test (**c,d,f**) or two-way ANOVA (**i**).

During testing sessions, control animals spent significantly more time exploring the new objects (Fig. 5c). By contrast, mice with A2B receptor knockdown in hippocampal astrocytes spent the same amount of time exploring the familiar and the new object (Fig. 5c,d), indicative of impaired recognition memory. The recognition memory was also severely impaired in mice with generalized conditional deletion of A2B receptors in brain astrocytes (*Adora2b*^*flox/flox*^:*Aldh1l1*^*Cre/ERT2+*^ animals treated with tamoxifen; Fig. 5e,f). As stimulation of β-adrenoceptors rescued the synaptic plasticity in the hippocampus in slice experiments (Fig. 4g,h), we next tested whether systemic treatment with the brain-permeant β-adrenoceptor agonist clenbuterol can improve memory under conditions of genetic blockade of A2B receptor-mediated signalling in astrocytes. Partial rescue of recognition memory was achieved after administration of clenbuterol (0.25 mg kg^−1; intraperitoneal) in mice with A2B receptor deletion in hippocampal astrocytes (Fig. 5c).

Accumulation of adenosine in the brain during wakefulness is recognized as a major driver for the need to sleep[35]. We next explored

how astrocyte-specific A2B receptor deletion and associated metabolic reprogramming impacts sleep–wake regulation (Fig. 5g). It was found that deletion of A2B receptors led to fragmentation of sleep and wake during the light (resting) phase, with shorter periods of non-rapid eye movement (NREM) sleep, and increased the number of wakefulness events (Fig. 5h and Extended Data Fig. 8c). Sleep was not affected when A2B receptor deletion was limited to the astrocytes of the hippocampus (Extended Data Fig. 8a,b). Sleep homeostasis, or sleep pressure, can be measured by the analysis of slow-wave activity during the light phase[36]. We found that A2B receptor deletion in brain astrocytes was associated with a significant reduction of slow-wave activity (0.5–4.0 Hz) and low-frequency slow-wave activity (0.5–1.5 Hz) during NREM in the light phase (Fig. 5i), indicative of reduced sleep pressure. These data suggested that signalling via astrocyte A2B receptors promotes synchronous slow-wave activity in NREM sleep and, therefore, has a major role in the brain mechanisms of sleep–wake regulation.

## Concluding comments

Adenosine is arguably one of the most important neuromodulators in the brain. Acting via the inhibitory A1 receptors, adenosine modulates neuronal excitability, neural circuit activity and behaviour[17,20,35–37]. This study describes another important function of this signalling molecule in the brain. We present experimental data suggesting that adenosine is responsible for the neuronal activity-dependent recruitment of astroglial glucose metabolism via A2B receptor activation and cAMP signalling. This metabolic signalling pathway ensures continuous support of the energy-demanding processes at the central synapse and is essential for synaptic plasticity underlying learning and memory. Our finding that astrocyte A2B receptors mediate the sleep-promoting effects of adenosine (sleep pressure) was an unexpected outcome of the study, pointing to shared mechanisms that regulate sleep and maintain brain energy homeostasis[38].

The astrocyte-to-neuron lactate shuttle model was originally proposed by Pellerin and Magistretti, who observed stimulation of astroglial glycolysis following uptake of synaptically released glutamate[39]. More recently, it was suggested that astroglial glycolysis is driven by increases in extracellular $[K^+]$, causing astroglial depolarization and intracellular alkalinization[40]. We found that the effects of increased $[K^+]$ on astrocyte glycolysis and lactate release are largely dependent on A2B receptor-mediated signalling (Extended Data Fig. 9), indicative of a potentially critical permissive role of the identified mechanism, which is required for other signals of neuronal activity to modulate astrocyte glucose metabolism.

In conclusion, this study identifies the adenosine A2B receptor as an astrocyte sensor of neuronal activity and demonstrates that cAMP signalling in astrocytes tunes brain energy metabolism to support fundamental brain functions such as sleep and memory. Our data suggest that neuronal metabolic needs are communicated to astrocytes by adenosine, which is responsible for metabolic activation of these glial cells via the cAMP–PKA signalling pathway — the same mechanism that controls glucose metabolism in the muscle and the liver. There is significant evidence that during ageing, progressive impairment of brain energy homeostasis contributes to cognitive decline and the development of neurodegenerative disease[41]. Treatments that can rescue and preserve brain energetics may prove to be effective in preventing and/or countering neurodegenerative disorders of ageing[41]. We suggest that stimulation of A2B receptors expressed by brain astrocytes could potentially be an effective therapeutic strategy aimed at supporting brain energy metabolism, maintaining cognitive health and promoting brain longevity.

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

# Methods

All animal studies were performed in accordance with the European Commission Directive 2010/63/EU (European Convention for the Protection of Vertebrate Animals used for Experimental and Other Scientific Purposes) and the UK Home Office Animals (Scientific Procedures) Act (1986) with project approval from the Institutional Animal Care and Use Committee of the University College London. The animals were group-housed and maintained on a 12-h light cycle and had ad libitum access to water and food. The mice were housed at 24 °C ambient temperature with relative humidity kept at 60 ± 5%. The rats were housed at 22 °C ambient temperature and 55 ± 10% relative humidity.

## Hippocampal slice preparation

Male and female wild-type and *Adora2b^flox/flox* mice, transduced to express the improved Cre (iCre) recombinase or tdTomato in astrocytes or young Sprague–Dawley rats (postnatal day 21 (P21)–P30) were terminally anaesthetized with isoflurane, the brains were removed and hippocampal slices (300–350 µm) were cut in an ice-cold slicing solution containing: 64 mM NaCl, 2.5 mM KCl, 1.25 mM $NaH_2PO_4$, 0.5 mM $CaCl_2$, 7 mM $MgCl_2$, 25 mM $NaHCO_3$, 10 mM glucose and 120 mM sucrose, saturated with 95% $O_2$ and 5% $CO_2$ (pH 7.4). Slices were then left to recover for 1 h in artificial cerebrospinal fluid (aCSF) containing: 119 mM NaCl, 10 mM glucose, 3 mM KCl, 2 mM $MgSO_4$, 1.25 mM $NaH_2PO_4$, 26 mM $NaHCO_3$ and 2 mM $CaCl_2$ (pH 7.4; 300–310 mOsm).

## Organotypic slice preparation

Organotypic hippocampal slice preparations were obtained from the brain of rat (P5–P7 of either sex) or mouse (P8–P10 of either sex) pups as previously described[42]. The animals were terminally anaesthetized with isoflurane, and the brains were isolated and placed in ice-cold Hanks' balanced salt solution (HBSS; Thermo Fisher) without $Ca^{2+}$, with added 20 mM glucose (total 25.6 mM glucose), 10 mM $MgCl_2$, 1 mM HEPES, 1 mM kynurenic acid, 0.005% phenol red, 100 U $ml^{-1}$ penicillin and 0.1 mg $ml^{-1}$ streptomycin. Coronal brain slices (350 µm) were cut and plated on 0.4-µm membrane inserts (Millicell CM, Millipore). The slices were incubated in a medium containing 50% Opti-MEM-1 (Thermo Fisher), 25% FBS (Sigma-Aldrich), 21.5% HBSS, 25 mM glucose, 100 U $ml^{-1}$ penicillin and 0.1 mg $ml^{-1}$ streptomycin. After 3 days, the incubation medium was replaced with a fresh medium and subsequently replaced twice a week. Astrocytes were transduced to express the genetically encoded fluorescent cAMP sensor Epac-S^H187 (ref. 14) or the PKA activity sensor AKAR4 (ref. 15), under the control of the *GFAP* promoter[43]. Adenoviral vectors AVV-Gfap-Epac-S^H187 ($1.0 × 10^{10}$ PFU $ml^{-1}$) or AVV-Gfap-AKAR4 ($1.0 × 10^{10}$ PFU $ml^{-1}$) were added to the incubation medium after 9–14 days of incubation and the slices were used in the experiments 3–5 days after the transfection.

## Primary astrocyte cultures

Primary astrocyte cultures were prepared from the brains of rat and mouse pups (P2–P3 of either sex) as previously described[44]. The animals were terminally anaesthetized by isoflurane, and the brains were removed and the brain regions of interest separated by dissection. After isolation, the cells were plated on poly-D-lysine-coated coverslips and maintained in DMEM medium (Thermo Fisher) with 10% FBS, penicillin (100 U $ml^{-1}$) and streptomycin (0.1 mg $ml^{-1}$) at 37 °C in a humidified atmosphere of 5% $CO_2$ and 95% air for a minimum of 10 days before the experiments.

## Genetic targeting of astrocytes to express fluorescent sensors

Widespread expression of fluorescent sensors in the forebrain was achieved following microinjections of viral vectors in neonatal mice (P0–P2 of both sexes)[42]. The pups were prepared for aseptic surgery, and the solution containing the viral vector was administered into a lateral cerebral ventricle. The microinjections (volume 1–1.5 µl per side) were made 0.25 mm lateral to the sagittal suture, 0.50–0.75 mm rostral to the coronal suture and 2 mm ventral from the surface of the skull. Pups were kept in groups of litters and returned to their mothers in their home cages after the injections. Imaging experiments were performed 3–4 weeks after the injections.

## Genetic targeting of hippocampal astrocytes to express iCre recombinase

To knock down A2B receptor expression, hippocampal astrocytes of mice with floxed *Adora2b* gene (*Adora2b^flox/flox*; P0–P2 or 3–5-month-old of both sexes)[26] were transduced to express iCre recombinase using the adeno-associated viral vector AAV5-Gfap-eGFP-iCre (VB1131, Vector Biolabs). Transduction of astrocytes with the viral vector AAV5-Gfap-tdTomato (University of Pennsylvania Vector Core) was used as a control. Neonatal mice were injected with viral vectors as described above. To transduce astrocytes of adult mice, the animals were anaesthetized with isoflurane (5% induction, 2–3% maintenance, in $O_2$-enriched air). Adequate depth of surgical anaesthesia was maintained and confirmed by the absence of a withdrawal response to a paw pinch. With the head of the animal secured in a stereotaxic frame, a midline dorsal incision was made to expose the surface of the skull. A small craniotomy was performed and hippocampal CA1 regions were targeted with one microinjection per side of either AAV5-Gfap-eGFP-iCre vector or AAV5-Gfap-tdTomato vector. Microinjections (volume 0.3–0.5 µl given at a rate of 0.1 µl $min^{-1}$) were made 2.0 mm rostral, 1.5 mm lateral and 1.5 mm ventral from bregma. After the microinjections, the wound was sutured. For post-surgical analgesia, the animals received buprenorphine (0.5 mg $kg^{-1}$, subcutaneously). No complications were observed after the surgery, and the animals gained weight normally.

## Two-photon excitation imaging of changes in [cAMP] and PKA activity in astrocytes

Optical recordings of changes in [cAMP] and PKA activity in hippocampal astrocytes were performed in acute or organotypic brain slices placed in a custom-made flow-through recording chamber mounted on a stage of an Olympus FV1000 microscope or FemtoSmart imaging system, optically linked to a Ti:Sapphire MaiTai laser with $λ^{2P}$ = 820 nm (Spectra Physics), with the emission filters set for the detection of CFP and YFP fluorescence. Recordings were performed at approximately 33–35 °C in aCSF saturated with 95% $O_2$ and 5% $CO_2$ (pH 7.4). For timelapse recordings of [cAMP] or PKA activity in astrocytes of the CA1 region of the hippocampus before and after the stimulation of Schaffer collateral fibres (described in detail below), images were collected with 512 × 512 pixel frames (frequency 1 Hz) using a water immersion ×25 Olympus objective (NA 1.05). Control optical recordings were performed using the same experimental settings without Schaffer collateral fibre stimulation applied. The laser power intensity was kept below 4 mW throughout the experiment.

## Electrophysiology

Electrophysiological experiments in acute hippocampal slices were performed as previously described[42,44,45]. The slices were placed in a recording chamber mounted on a stage of an Olympus BX51WI upright microscope (Olympus) equipped with a LUMPlanFL/IR 40 × 0.8 objective coupled to an infrared DIC imaging system and an Evolve 512 EMCCD camera (Photometrics). A source of fluorescent light was an X-Cite Intelli lamp (Lumen Dynamics). Wide-field fluorescence images were acquired using Micromanager 4.1 (ImageJ plugin) software and various digital zooms to visualize transduced astrocytes. Schaffer collateral fibres were stimulated using a concentric bipolar electrode (pulse width of 100 µs; amplitude of 20–300 µA, corresponding to approximately one-third of the saturating response). Synaptic responses were induced by trains of Schaffer collateral fibre stimulations consisting of five pulses applied at 20 Hz and delivered 50 ms apart. Evoked field excitatory postsynaptic potentials (fEPSPs) were

recorded using glass electrodes (1–2 MΩ) placed at a distance of more than 200 µm from the stimulating electrode in the CA1 region showing strong astrocytic expression of transgenes. In a typical experiment, the evoked fEPSPs were recorded for at least 60 min.

Synaptic long-term potentiation (LTP) in the CA3–CA1 pathway was induced by high-frequency stimulation (HFS) of Schaffer collateral fibres[32,42,45]. Basal synaptic transmission was first tested by low-frequency Schaffer collateral fibre stimulation (trains of 5 pulses applied at 20 Hz every 30 s) and monitored with recordings of evoked fEPSPs for 15–20 min. The HFS was then applied to induce LTP — a protocol consisting of three trains of stimuli (100 pulses at 100 Hz), applied with 60-s intervals. The fEPSPs were recorded for 60–90 min after the induction of LTP. In these experiments, picrotoxin (100 µM) and CGP-52432 (5 µM) were added to the bath solution to block the inhibitory transmission. The recorded signal was amplified (Multipatch 700B) and processed using pClamp 10.2 software (Molecular Devices). Recordings were filtered and digitized; the fEPSP slope was measured for the first evoked response in each pulse train.

## Biosensor recordings of lactate and adenosine release
Lactate and adenosine were recorded in acute brain slices using amperometric enzymatic microelectrode biosensors (Sarissa Biomedical)[46–48]. The sensors were placed in direct contact with the surface of the slice placed on an elevated grid in a flow chamber at 35 °C. A dual recording configuration of a null sensor (lacking enzymes) and lactate or adenosine biosensor was used, as previously described[46,47]. The null sensor was used to determine whether any nonspecific electroactive interferents were detected and confounded the measurements. Null sensor currents were subtracted from the lactate or adenosine biosensor currents, and the resulting current profile was used to calculate the amount of the released analyte. In some experiments, adenosine and lactate signals were recorded simultaneously. Biosensors were calibrated with a known amount of lactate or adenosine added to the perfusate flowing through the recording chamber (in the identical temperature, aCSF composition and osmolarity conditions) immediately before and after the recordings. The mean of the initial and final calibrations was used to convert changes in sensor current to changes in lactate or adenosine concentration.

## Recordings of [cAMP], PKA activity, [glucose] and NADH–NAD$^+$ redox state in cultured astrocytes
Primary cultured astrocytes were transduced using viral vectors to express genetically encoded fluorescent sensors of cAMP (Epac-S$^{H187}$ (ref. 14)), PKA activity (AKAR4 (ref. 15)), the cytosolic NADH–NAD$^+$ redox state (Peredox[28]) or glucose (FLIP$^{12}$glu-700µΔ6 (ref. 27)). The cells were incubated with a viral vector for 12 h and used in the experiments after 3 days following transduction. Optical recordings of changes in [cAMP], PKA activity, [glucose] and NADH–NAD$^+$ redox state in astrocytes were performed using an inverted wide-field Olympus microscope equipped with a ×20 oil immersion objective lens, a cooled CCD camera (Clara, Andor, Oxford Instruments), a Xenon arc lamp, a monochromator and an Optosplit (Cairn Research). Recordings were performed in a custom-made flow-through chamber at 32–34 °C in aCSF saturated with 95% $O_2$ and 5% $CO_2$ (pH 7.4). The rate of chamber perfusion with aCSF was 1 ml min$^{-1}$. To record FRET signal changes (cAMP, PKA activity and glucose sensors), 415/10 nm excitation light was applied, and the fluorescence emission was recorded at 470/24 and 535/30 nm. To record changes in the cytosolic NADH–NAD$^+$ redox state, the Peredox sensor was excited with 405/10 nm and 575/10 nm light and the fluorescence emission was recorded at 535/30 nm and 630/35 nm.

## Conditional deletion of *Adora2b* in astrocytes
To induce conditional A2B receptor knockdown in brain astrocytes, mice carrying a loxP-flanked *Adora2b* allele (*Adora2b*$^{flox/flox}$)[26] were crossed with the mice expressing an inducible form of Cre (Cre/ERT2)

under the control of the astrocyte-specific *Aldh1l1* promoter[29]. Recombination specificity of *Aldh1l1*$^{Cre/ERT2}$ mice has been previously described[29]. Breeding was organized through PCR genotyping obtained from ear DNA biopsies. Tamoxifen (100 mg kg$^{-1}$ dissolved in corn oil; injected intraperitoneally (i.p.) daily for 5 consecutive days) was given to *Adora2b*$^{flox/flox}$:*Aldh1l1*$^{Cre/ERT2^+}$ and *Adora2b*$^{flox/flox}$:*Aldh1l1*$^{Cre^-}$ mice at 12–16 weeks of age. In separate groups of *Adora2b*$^{flox/flox}$:*Aldh1l1*$^{Cre/ERT2^+}$ and *Adora2b*$^{flox/flox}$:*Aldh1l1*$^{Cre^-}$ animals, corn oil was given as a vehicle control. The expression of the A2B receptor in the brain was examined 4 weeks after tamoxifen treatment.

The specificity of genomic recombination of the *Adora2b* locus was evaluated by PCR of the brain tissue of *Adora2b*$^{flox/flox}$:*Aldh1l1*$^{Cre/ERT2^+}$ and *Adora2b*$^{flox/flox}$:*Aldh1l1*$^{Cre^-}$ mice treated with tamoxifen. The following primers were used to identify the 719-bp-long recombination product: recombination forward 5′-CAGTGCTGAGGCTATTAAAAAGGG-3′ and recombination reverse 5′-GGTGACTGCATAGCCTAGGGAAAC-3′. For PCR genotyping, the following primers were used: *Adora2b*$^{flox/flox}$ forward: 5′-TTAAAAGGTGATTCCCAGCACG-3′; *Adora2b*$^{flox/flox}$ reverse: 5′-GGTGACTGCATAGCCTAGGGAAAC-3′; *Aldh1l1*$^{Cre/ERT2}$ forward: 5′-CTT CAACAGGTGCCTTCCA-3′; *Aldh1l1*$^{Cre/ERT2}$ reverse: 5′-GGCAAACGGAC AGAAGCA-3′. No *Adora2b* recombination was observed in tissues of control animals.

## Isolation of astrocytes and A2B receptor protein quantification
Hippocampal astrocytes were isolated from the brains of *Adora2b*$^{flox/flox}$ or wild-type mice injected with AAV5-Gfap-eGFP-iCre vector to determine the effectiveness of A2B receptor deletion in this model. The animals were euthanized by isoflurane overdose, perfused transcardially with ice-cold saline and the hippocampal regions were isolated. The tissue was enzymatically dissociated to obtain a suspension of individual cells. Astrocytes were identified and sorted by eGFP expression (ARIA II BD). The purified fraction of eGFP-expressing astrocytes was lysed by sonication at low frequency using a Soniprep 150 Sonicator (Sanyo). Supernatant was collected after centrifugation and used for quantification of A2B receptor protein by ELISA (E03A1281, BluGene Biotech). A2B receptor protein quantification was normalized to the number of eGFP-positive cells in each sample.

## Immunohistochemistry
At the end of the experiments, *Adora2b*$^{flox/flox}$ mice transduced to express iCre recombinase or tdTomato in hippocampal astrocytes were given an anaesthetic overdose (sodium pentobarbital; 200 mg kg$^{-1}$; i.p.), the brains were removed, fixed in 4% paraformaldehyde for 12 h and sliced (20 µm). Identification of eGFP expression in the brains of mice transduced to express iCre recombinase in hippocampal astrocytes was aided by antibody labelling. After slicing, free-floating sections were incubated with chicken with anti-GFP antibody (1:500; anti-GFP1020, Aves Labs or AB13970, Abcam) for 12 h at 4 °C. Sections were then incubated with secondary anti-chicken antibody AlexaFluor 568 (1:200; A-11041, Thermo Fisher) or AlexaFluor 488 (1:1,000; A-21441, Thermo Fisher) for 2 h at room temperature. Brain sections were mounted onto slides with Fluoroshield with DAPI (Sigma). Tiled images of coronal cross-sections were obtained using a Zeiss 800 confocal microscope.

To evaluate the cell specificity of transduction with the AAV5-Gfap-eGFP-iCre vector, astrocytes, oligodendrocytes, neurons and microglia were labelled using the following antibodies: rabbit anti-GFAP (1:500; 23935-1-AP, Proteintech), rabbit anti-MBP (1:200; MA5-35074, Thermo Fisher), rabbit anti-NeuN (1:200; AB236870, Abcam), and rabbit anti-Iba1 (1:200; GTX100042, GeneTex), respectively. Secondary anti-rabbit antibody AlexaFluor 568 (1:1,000; 175470, Abcam) was used to identify the transduced cells.

## Western blot
Astrocyte cultures prepared from *Adora2B*$^{flox/flox}$ mice, transduced with either AAV5-Gfap-eGFP-iCre or AAV5-Gfap-tdTomato, were washed

twice with PBS and the cells were collected in ice-cold lysis buffer supplemented with protease and phosphatase inhibitors (Thermo Fisher). Samples were snap frozen, sonicated and centrifuged at 14,000 rpm; the protein content of the extracts was determined by the Pierce BCA protein assay (Thermo Fisher). Twenty-five micrograms of protein was then fractionated on a Mini-PROTEAN TGX stain-free polyacrylamide gel (10%) (Bio-Rad) under denaturing and reducing conditions. Total protein levels were visualized in the gel after 3 min of exposure to UV light using a UV-transilluminator. Proteins were then transferred to polyvinylidene fluoride (PVDF) membranes (Bio-Rad), which were incubated with 5% non-fat milk. Membranes were next incubated overnight with the primary antibodies diluted in 5% BSA: rabbit anti-A2BR antibody (4 µg ml$^{-1}$; AB1589P, Merck Millipore) and mouse anti-actin antibody (1:5,000; 3700, Cell Signaling Technologies), followed by incubation with the corresponding species-specific horseradish peroxidase (HRP)-conjugated secondary antibodies (1:5,000; anti-rabbit-HRP, sc-2054, Santa Cruz and anti-mouse-HRP, sc-2005, Santa Cruz). The luminol-based Pierce ECL Western Blotting Substrate (Thermo Fisher) was used to detect the HRP activity. After scanning the X-ray films, protein band densities were quantified using ImageJ.

### Quantitative real-time PCR

Quantitative real-time PCR (rt–qPCR) assay was used to determine the level of *Adora2b* expression in the hippocampal tissue of *Adora2B*$^{flox/flox}$ mice transduced to express iCre recombinase or tdTomato in astrocytes and in *Adora2b*$^{flox/flox}$:*Aldh1l1*$^{Cre/ERT2+}$ and *Adora2b*$^{flox/flox}$:*Aldh1l1*$^{Cre-}$ mice treated with tamoxifen. Total RNA was extracted, purified (RNeasy mini kit, 74106, Qiagen) and reverse transcribed using the QuantiTect Reverse Transcription Kit (205311, Qiagen) as per the manufacturer's protocol. PCRs were performed in 20-µl volumes using the TaqMan Universal Master Mix II (4440040, Thermo Fisher) with a final volume of 9 µl cDNA, equivalent to 25 ng of RNA, sample template per reaction. PCRs were performed using the TaqMan assay (*Adora2b*, Mm00839292_m1, 61-bp amplicon length, Thermo Fisher) as detection method and an Agilent Technologies Aria Mx Real-time PCR system (Agilent). *Adora2b* expression was quantified using the comparative CT method ($^{\Delta\Delta}$Ct) and presented as arbitrary units of expression, normalized to the expression of the ubiquitin C gene (Mm01201237_m1, 92-bp amplicon length, Thermo Fisher).

### Analysis of single-cell RNA-seq data

Single-cell RNA-seq data of the mouse brain were obtained from a publicly available database collated and maintained by the Linnarsson group (Karolinska Institutet; http://mousebrain.org/). Cell dissociation, single-cell RNA-seq and quality control methods are described in detail in the original report of the database[24]. Data processing and visualization were performed using the Seurat package[49] in R (v.4.2.2, 'Innocent and Trusting'). The combined mouse cortical cell RNA-seq dataset was obtained from 50,478 cells with expression data for 27,998 genes. All cells that displayed nFeatures greater than 200 and less than 4,000 and a percentage of mitochondrial RNA of less than 30% were included in the analysis. Remaining were 49,703 cells with average UMI counts (absolute number of observed transcripts; nCount) of 3,124.92 and nFeatures (genes per cell) of 1,592.39. The data were then log normalized and scaled to 10,000 transcripts per cell. The FindVariableFeatures function[50] was used to identify the 4,000 most variable genes between the cells to be used in the principal component analysis (PCA). Before PCA, data were scaled with a linear transformation to ensure that all genes were given equal weight in the subsequent analyses. Dimensionality reduction was then performed by PCA on the scaled data up to and including the first 100 identified principal components. An elbow plot was used to determine the effective number of principal components, which was found to be 75. The *k*-nearest neighbour (KNN) graph was constructed using these 75 principal components. To cluster the cells, the Louvain method for community detection (Louvain algorithm)

was used with resolution set to 2.0 as recommended for the datasets of this size[51]. Uniform manifold approximation and projection (UMAP) was used to visualize the cell clusters in two dimensions based on the same 75 principal components used for clustering and yielded 63 distinct cell clusters. The identity of cells comprising these clusters was determined by the differential expression of characteristic cell-specific marker genes[24]. The distribution of *Adora2a* and *Adora2b* expression was then plotted across the identified clusters. In addition, to determine the distribution of *Adora2a* and *Adora2b* expression restricted to the astrocyte-like cells in the sample, the expression data from cells identified as astrocytes in the initial projection were pooled and reclustered by the same method described for the whole dataset with modifications. The KNN graph was constructed using 20 principal components with Louvain algorithm resolution set to 0.8. Finally, the distribution of *Adora2a* and *Adora2b* expression was determined in all the identified cell-type clusters with more than 150 members where the cluster had at least 10% of all included cells that were found to be positive for either *Adora2a* or *Adora2b*.

### Metabolomics

Mice were taken from their home cages, terminally anaesthetized with isoflurane overdose and transcardially perfused with ice-cold aCSF saturated with 95% $O_2$ and 5% $CO_2$. The brains were quickly isolated and snap-frozen in liquid nitrogen. The time taken from removing the animal from its habitat to the preparation of the frozen sample did not exceed 5 min in each case.

Small molecules from the brain tissue were extracted as previously described[52]. In brief, frozen brain samples (100–150 mg) were transferred into 2-ml soft tissue homogenizing tubes (CK14, Precellys), kept on dry ice and 0.6 ml pre-chilled methanol:chloroform (2:1 v:v) solution was added to the samples. Samples were transferred to a bead beater (Precellys) and homogenized (within 1 min of removing from dry ice to keep metabolite profiles) two times for 10 s at 10,000 rounds per min. Subsequently, 0.2 ml of water and 0.2 ml of $CHCl_3$ were added to each tube; the samples were vortex mixed and centrifuged at 13,000*g* for 10 min. The resulting top aqueous layer was aliquoted into microtubes, dried in Speedvac (Savant; 30 °C, overnight, VAQ setting) and then kept at −80 °C until assayed. A 'pooled quality control' sample was prepared by mixing 50 µl of the aqueous layer of each sample. The samples were reconstituted in 35 µl of water and analysed by ion-pairing liquid chromatography–mass spectrometry (LC–MS)[53], using a XEVO TQ-S tandem mass spectrometer and an Acquity ultraperformance liquid chromatography binary solvent manager equipped with a CTC autosampler (Waters). Data were acquired with an electrospray ionization in a negative-ion mode and chromatography using a Waters HSS T3 column (1.8 µm, 2.1 × 100 mm) with a binary solvent system of 10 mM tributylamine + 15 mM acetic acid in water (as mobile phase A) and 80% methanol + 20% isopropanol (as mobile phase B) with a gradient elution. The sample processing order was randomized. Injections of double blanks (water) and single blanks were performed to ensure system stability, and to identify carryover and solvent interference peaks. The pooled quality control sample was injected at the beginning of the run and then once every tenth injection throughout the run, to monitor the instrument stability across the entire analytical session. The pooled quality control sample was used to evaluate the normalization method. During sample preparation and mass spectrometry analysis, the investigators were blinded to the identity of the experimental samples. The list of all the annotated metabolites and raw data are provided in Supplementary Table 1 and available at https://doi.org/10.25345/C5X05XQ2B.

The LC–MS data were processed using Skyline[54]. Manually curated peaks were annotated based on in-house database of *m/z* and retention time of external standards; only peaks that passed the experimental ion ratios for the product ions (where more than one product existed) were integrated and peak area values were then exported.

All data were normalized using the probabilistic quotient and analysed using R (v4.1.3). Relative levels of metabolites were mean centred and variance adjusted using the scale function of R. Multivariate classification models were built using partial least squares-discriminant analysis (PLS-DA) with the ropls R package (v.1.26.4) using sevenfold cross-validation for 10,000 permutations. Metabolomics pathway enrichment analysis was performed using MetaboAnalyst[55] R package (v5.0) and the SMPDB database of metabolic pathways, with the top five metabolites differentiating between the experimental groups in the PLS-DA model as input.

## Novel object recognition test

The experiments were performed in an isolated room under dim light conditions. Before testing, the animals were handled by the investigator daily for at least 1 week before the main experiment. The animals in their home cages were brought into the behavioural testing room 1 h before the experiment. First, each mouse was allowed to explore an empty square experimental chamber (40 × 40 × 40 cm) for 5 min. The animal was presented for 5 min with two identical objects placed diagonally on the floor of the chamber and then the animal was returned to the home cage. After 1 h, the animal was returned to the testing arena in which one of the original objects was replaced with a new object. The behaviour of the animal in the arena was recorded by tracking the nose of the mouse using a video camera and Viewer III software (Biobserve). The recognition memory was assessed by calculating the time the animal spent and the frequency of visits the animal made near each object during the training and testing sessions. The discrimination index (DI) as a measure of recognition memory was calculated using the formula DI = (time spent at novel object − time spent at familiar object)/(time spent at novel object + time spent at familiar object) × 100.

## EEG and EMG recordings

For electrode implantation, the mice were anaesthetized with isoflurane (5% induction, 2–3% maintenance, in $O_2$) and received buprenorphine (0.5 mg kg$^{-1}$, subcutaneously) for perioperative analgesia. Adequate depth of surgical anaesthesia was maintained and confirmed by the absence of a withdrawal response to a paw pinch. With the head of the animal in a stereotaxic frame, a midline dorsal incision was made to expose the surface of the skull. EEG and EMG electrode headmounts (Pinnacle Technology) were secured to the skull using stainless steel screws and silver epoxy was used for optimal electrical connectivity. Two EMG leads were inserted into the nuchal muscles, and the headmounts were secured with dental acrylic. After a 10-day recovery period in a room with 12–12 light–dark cycle, mice were placed in individual Plexiglas circular recording cages (Pinnacle Technology) with unlimited access to water and food. The headmounts were connected to a lightweight EEG preamplifier (Pinnacle Technology) to enable unrestricted movement. Following a 3-day habituation period, EEG signals were sampled at 400 Hz using Sirenia software (Pinnacle Technology). Sleep stages were scored in 4-s epochs using SleepSign for Animal software (Kissei Comtec). Periods of wakefulness were identified by low-amplitude, high-frequency EEG and high EMG activity; NREM sleep was identified by high-amplitude, low-frequency EEG with minimal EMG modulation; and REM sleep was identified by low-amplitude, desynchronized EEG with low or absent EMG activity.

## Data analysis

Imaging data were acquired using IQ3 software (v6.3; Andor, Oxford Instruments) or Olympus FluoView software (v4; Olympus) and analysed using Fiji (ImageJ). Biosensor recordings were acquired using Power 1401 interface and analysed using Spike2 software (v7; Cambridge Electronic Design). Electrophysiological data were acquired and analysed using pClamp software (v10.2). Statistical analysis of the data was performed using Origin 2019 software (v9.6) and GraphPad Prism software (v8). Distribution of data was analysed by a Shapiro−Wilk normality test. Grouped data were analysed using one-way or mixed-model ANOVA or Kruskal−Wallis test (for non-normally distributed data) when comparing data between more than two groups. One-way ANOVA was followed by Dunnett's post-hoc test when comparing experimental groups against one control group or by Sidak's post-hoc test when multiple comparisons between the groups were made. Comparisons of data obtained in the experiments with two groups were made using $t$-test or Mann−Whitney $U$-test (for non-normally distributed data). EEG data were analysed by repeated-measures ANOVA when multiple measurements were made over time in the same groups followed by Tukey's post-hoc multiple comparisons test. The data are reported as individual values and mean ± s.e.m. or box-and-whisker plots. In the box-and-whisker plots, the central dot indicates the mean, the central line indicates the median, the box limits indicate the upper and lower quartiles, and the whiskers extend to 1.5 times the interquartile range from the quartiles. Details of the statistical tests applied are provided within the figure legends.

## Reporting summary

Further information on research design is available in the Nature Portfolio Reporting Summary linked to this article.

## Data availability

The data that support the findings in this study are included within the supplementary material. The source data underlying Figs. 1b,e,f, 2c,f,g,i, 3c,d,f,g–i,l, 4b,c,e–h and 5c,d,f,i and Extended Data Figs. 1e, 2d,e, 3c, 4a,b, 5b–f, 6, 7a,b, 8a–c and 9c,d are provided as source data files. Single-cell RNA-seq source data underlying Fig. 2a,b are available from a publicly available database (http://mousebrain.org/). The isotopically quantified LC−MS/MS data are deposited in MassIVE (https://doi.org/10.25345/C5X05XQ2B) with the accession number MSV000094445. Source data are provided with this paper.

## Code availability

The computer code used for the analyses of the RNA-seq data is available from an open repository (https://doi.org/10.5281/zenodo.10941579)[56].

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

**Acknowledgements** This work was supported by the Wellcome Trust (200893/Z/16/Z and 223057/Z/21/Z to A.V.G., and 212251/Z/18/Z and 223131/Z/21/Z to D.A.R.). A.V.G. was supported by a Wellcome Senior Research Fellowship (200893/Z/16/Z). D.A.R. was supported by a Wellcome Principal Research Fellowship (212251/Z/18/Z). P.A. is the recipient of a Medical Research Council Career Development Award (MR/Y010051/1). G.L.A. was supported by a NIHR Advanced Fellowship (NIHR300097). We thank B. Khakh for providing *Aldh1l1*[Cre/ERT2] mice, K. Jalink for providing the Epac-S[H187] sensor clones and P. Haydon for sharing the Pinnacle Technology EEG/EMG recording equipment. Schematic of the recording setup illustrated in Fig. 5g was created using BioRender (https://biorender.com).

**Author contributions** S.M.T. performed the experiments in cell culture, some of the optical recordings in slices, the biosensor recordings, the behavioural studies and analysed the data. O.K. performed the electrophysiological and optical recordings in hippocampal slices and analysed the data. A.B. performed the EEG and EMG recordings and analysis, and the immunohistochemistry. S.N. performed the biosensor recordings and immunohistochemistry. P.S.H. analysed the single-cell RNA-seq data. V.S.-K. and P.A. performed brain metabolomics and analysed the data. A.H. and C.K. analysed *Adora2b* expression by rt–qPCR. N.E. performed the western blot analysis of A2B receptor expression. A.G.D.A. and G.L.A. performed isolation of astrocytes and A2B receptor protein quantification. A.G.T. and S.K. developed and validated viral vectors for the expression of the cAMP sensor Epac-S[H187] in astrocytes. N.D. helped with the design of the experiments using biosensor recordings and data analysis. T.E. generated and provided *Adora2B*[flox/flox] mice for the study. D.A.R. helped with the design of the electrophysiological experiments and data analysis. A.V.G. helped S.M.T. in performing the biosensor recording and behavioural studies. A.V.G., S.M.T. and S.K. conceived the project. A.V.G. and S.M.T. directed the experiments and wrote the paper with help from the other co-authors.

**Competing interests** The authors declare no competing interests.

**Additional information**
**Correspondence and requests for materials** should be addressed to Shefeeq M. Theparambil or Alexander V. Gourine.

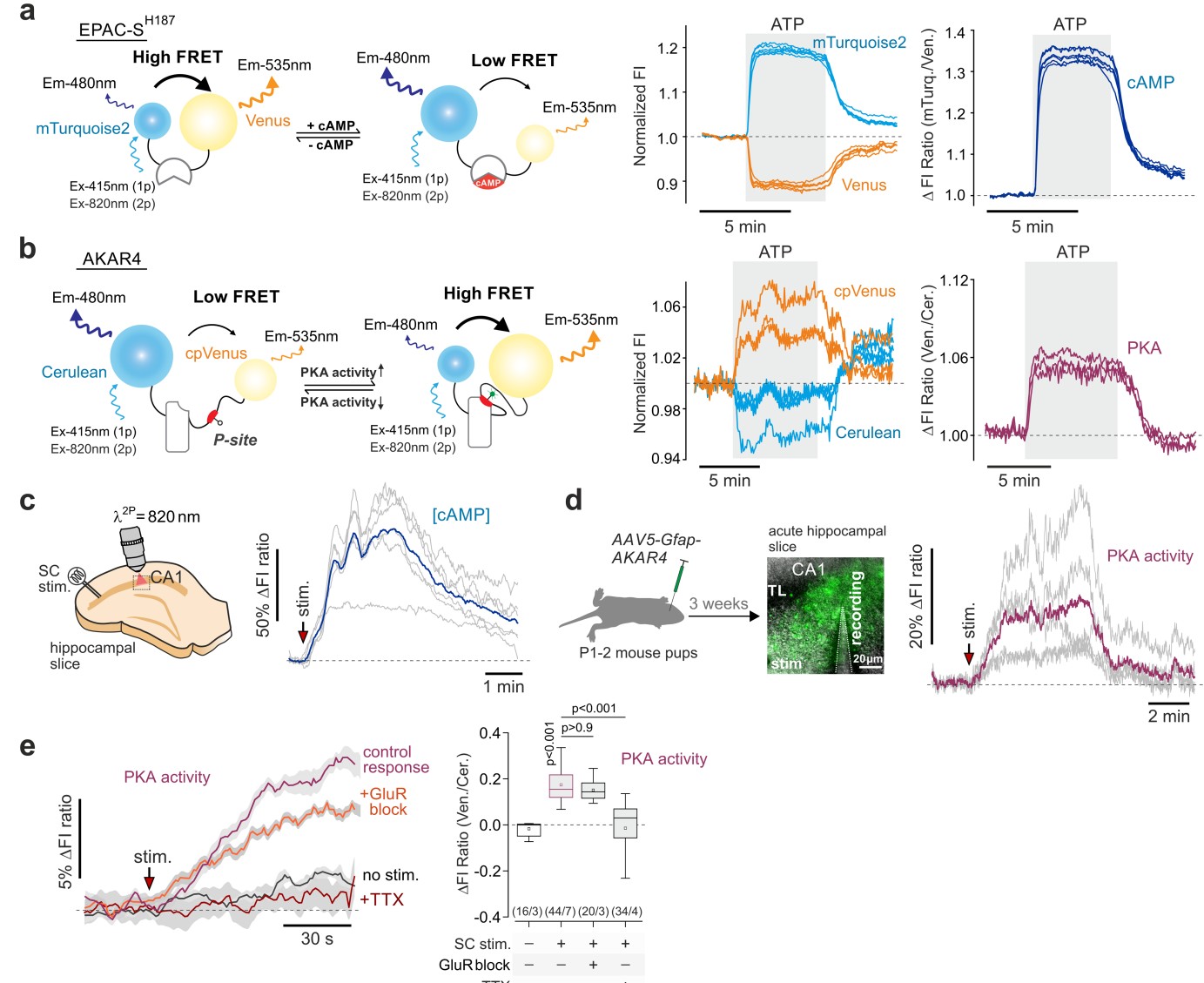

**Extended Data Fig. 1 | Recordings of changes in [cAMP] and PKA activity in astrocytes.** (**a**) Schematic illustrating the operational principle of the FRET-based cAMP sensor Epac-S[H187]. mTurquoise2 is excited with 415 nm wavelength (820 nm for 2p), and emissions from mTurquoise2 (donor) and Venus (acceptor) are collected at 480 nm and 535 nm. cAMP binding to the sensor induces a conformational change in the protein, resulting in increases and decreases in fluorescence of mTurquoise2 and Venus, respectively (low FRET). Traces illustrate ATP-induced changes in fluorescence of mTurquoise2 and Venus and changes in fluorescence intensity (FI) emission ratio between mTurquoise2 and Venus (mTurq./Ven.) of the Epac-S[H187] sensor, expressed in cultured astrocytes; (**b**) Schematic illustrating the operational principle of PKA activity sensor AKAR4. Cerulean is excited with 415 nm wavelength (820 nm for 2p), and emissions from Cerulean (donor) and cpVenus (acceptor) are collected at 480 nm and 535 nm. Phosphorylation of the sensor protein by PKA induces a conformational change, leading to decreases and increases in fluorescence of Cerulean and cpVenus, respectively (high FRET). Traces illustrate ATP-induced changes in fluorescence of Cerulean and cpVenus and changes in the FI ratio between cpVenus and Cerulean (Ven./Cer.) of the AKAR4 sensor, expressed in cultured astrocytes; (**c**) Schematic of the experimental design and representative recordings illustrating the profiles of changes in intracellular [cAMP] in astrocytes of the CA1 area induced by stimulation of Schaffer collateral (SC) fibers (burst of 5 pulses at 20 Hz) in hippocampal slices. Traces illustrate responses of individual astrocytes (grey) and averaged response profile (blue); (**d**) Intracerebroventricular injections of AAV5-Gfap-AKAR4 vector in neonates. The image illustrates AKAR4 expression in hippocampal astrocytes (acute brain slice of an adult mouse). Traces illustrate changes in PKA activity in individual CA1 astrocytes (grey) and averaged PKA response profile (purple) induced by stimulation of SC fibers; (**e**) Representative traces and summary data illustrating changes in PKA activity in astrocytes of the CA1 area induced by stimulation of SC fibers in control conditions and under conditions of glutamate receptor (GluR) blockade (CPP/NBQX/MCPG) or in the presence of tetrodotoxin (TTX). Traces illustrate averaged (mean ± s.e.m.) recordings from several individual cells in a representative experiment. In the box-and-whisker plot, the central dot indicates the mean, the central line indicates the median, the box limits indicate the upper and lower quartiles, and the whiskers extend to 1.5 IQR from the quartiles. Numbers in parentheses indicate the number of individual cells/number of separate slices, prepared from the same number of animals. *P* values, one-way ANOVA followed by Sidak's post hoc test.

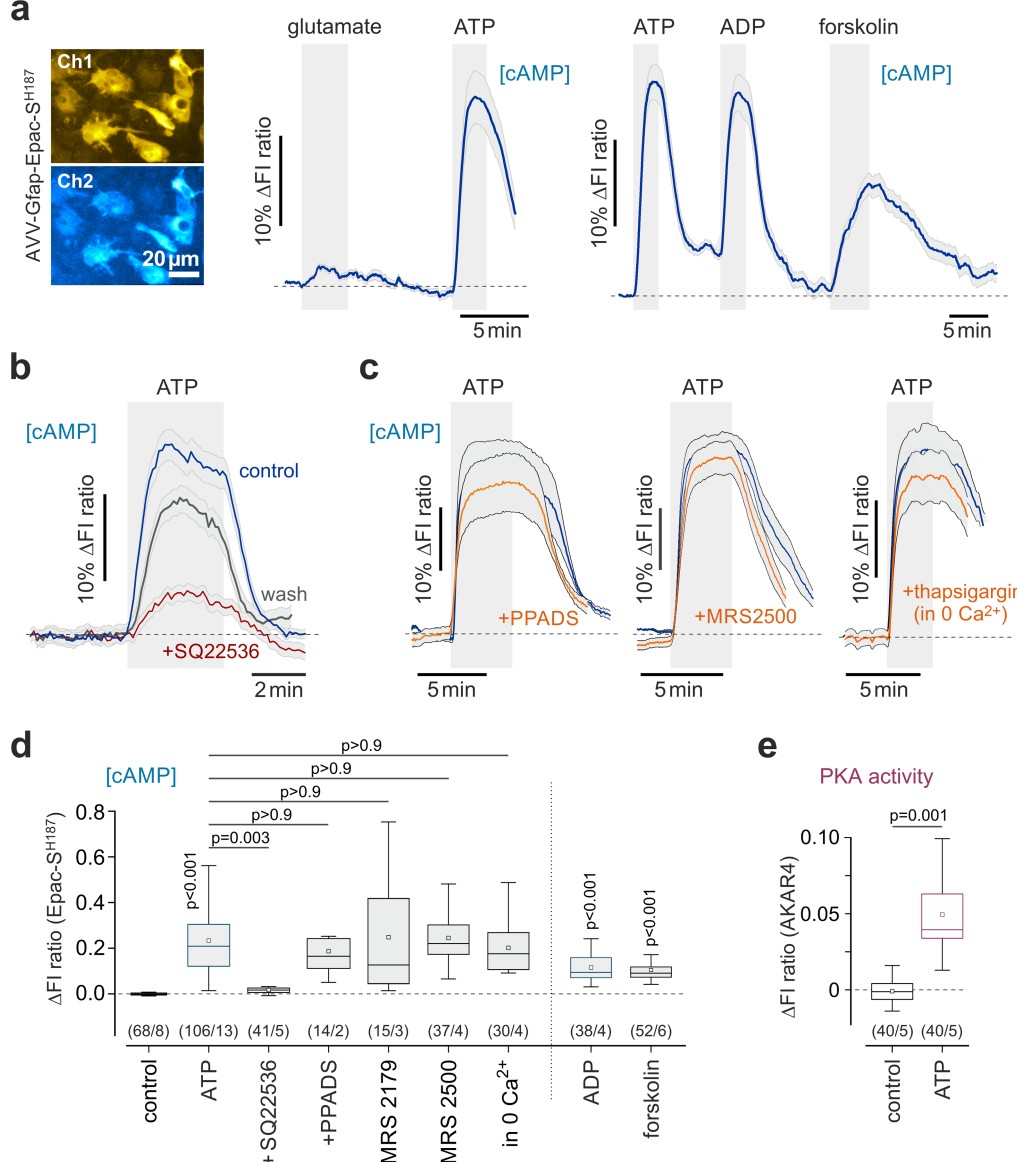

**Extended Data Fig. 2 | Activation of cAMP signalling pathway in astrocytes in response to ATP.** (**a**) Astrocytes in culture transduced to express Epac-S[H187] (Ch1: mTurquoise; Ch2: Venus). Traces illustrate changes in $[cAMP]_i$ (Epac-S[H187] FI ratio mTurq./Ven.) in response to glutamate (100 µM), ATP (30 µM), ADP (30 µM), and adenylyl cyclase activator forskolin (10 µM); (**b**) ATP-induced cAMP responses in astrocytes were inhibited by adenylyl cyclase inhibitor SQ22536 (100 µM); (**c**) ATP-induced cAMP responses in astrocytes were not affected by inhibition of ionotropic P2X (PPADS, 100 µM) or metabotropic P2Y$_1$ (MRS2500, 2 µM) receptors for ATP, or by blockade of $Ca^{2+}$ signalling (in zero $Ca^{2+}$/thapsigargin, 1 µM); (**d**) Summary data illustrating peak cAMP responses in astrocytes in response to ATP, ADP, glutamate, forskolin; ATP in the presence of SQ22536, PPADS, MRS2179, or MRS2500; and ATP in zero $Ca^{2+}$ conditions in the presence of thapsigargin. *P* values, one-way ANOVA followed by Sidak's post hoc test; (**e**) Summary data illustrating peak increases in PKA activity in astrocytes in response to ATP. *P* value, two-tailed Student's *t*-test. In panels (a), (b) and (c), traces illustrate averaged (mean ± s.e.m.) recordings from several individual cells in a representative experiment. In the box-and-whisker plots, the central dot indicates the mean, the central line indicates the median, the box limits indicate the upper and lower quartiles, and the whiskers extend to 1.5 IQR from the quartiles. Numbers in parentheses indicate the number of individual cells/number of separate cultures, prepared from the same number of animals.

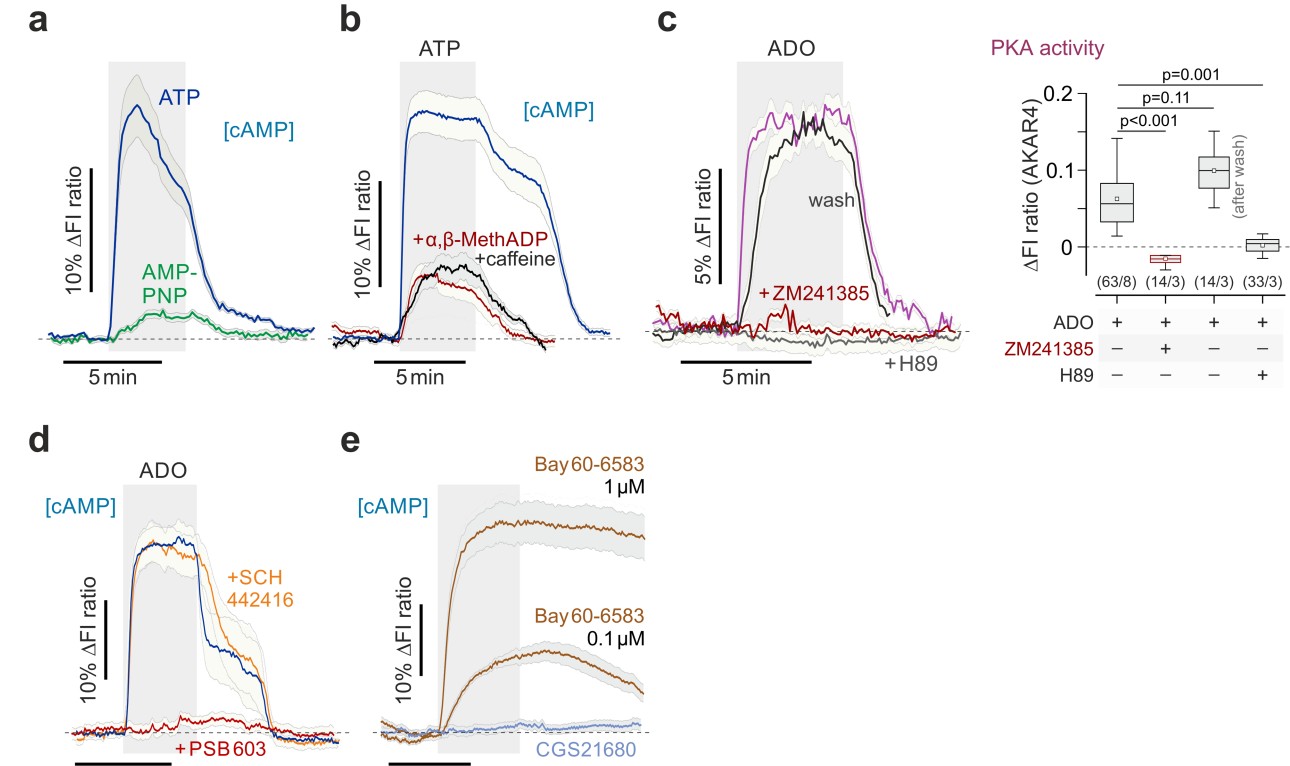

**Extended Data Fig. 3 | ATP-induced activation of cAMP/PKA signalling pathway in astrocytes is mediated by adenosine.** (**a**) Representative traces illustrating changes in $[cAMP]_i$ (Epac-S$^{H187}$ FI ratio mTurq./Ven.) in astrocytes induced by ATP (10 µM) or a non-hydrolysable ATP analogue AMP-PNP (10 µM); (**b**) Representative traces illustrating [cAMP] changes in astrocytes induced by ATP in the presence of ecto-5′-nucleotidase inhibitor α,β-methylene ADP (200 µM), or adenosine receptor antagonist caffeine (1 mM); (**c**) Representative traces and summary data illustrating changes in PKA activity (AKAR4 FI ratio Ven./Cer.) in astrocytes induced by adenosine (10 µM), and adenosine in the presence of ZM241385 (10 µM) or PKA inhibitor H89 (10 µM); (**d**) Adenosine-induced cAMP responses in astrocytes were blocked by A2B receptor antagonist PSB 603 (10 µM), whilst A2A antagonist SCH442416 (10 µM) had no effect; (**e**) cAMP responses in astrocytes induced by A2B receptor agonist BAY 60-6583. A2A agonist CGS21680 (1 µM) had no effect on cAMP in astrocytes. Traces illustrate averaged (mean ± s.e.m.) recordings from several individual cells in a representative experiment. In the box-and-whisker plot, the central dot indicates the mean, the central line indicates the median, the box limits indicate the upper and lower quartiles, and the whiskers extend to 1.5 IQR from the quartiles. Numbers in parentheses indicate the number of individual cells/ number of separate cultures, prepared from the same number of animals. *P* values, one-way ANOVA followed by Dunnett's post hoc test.

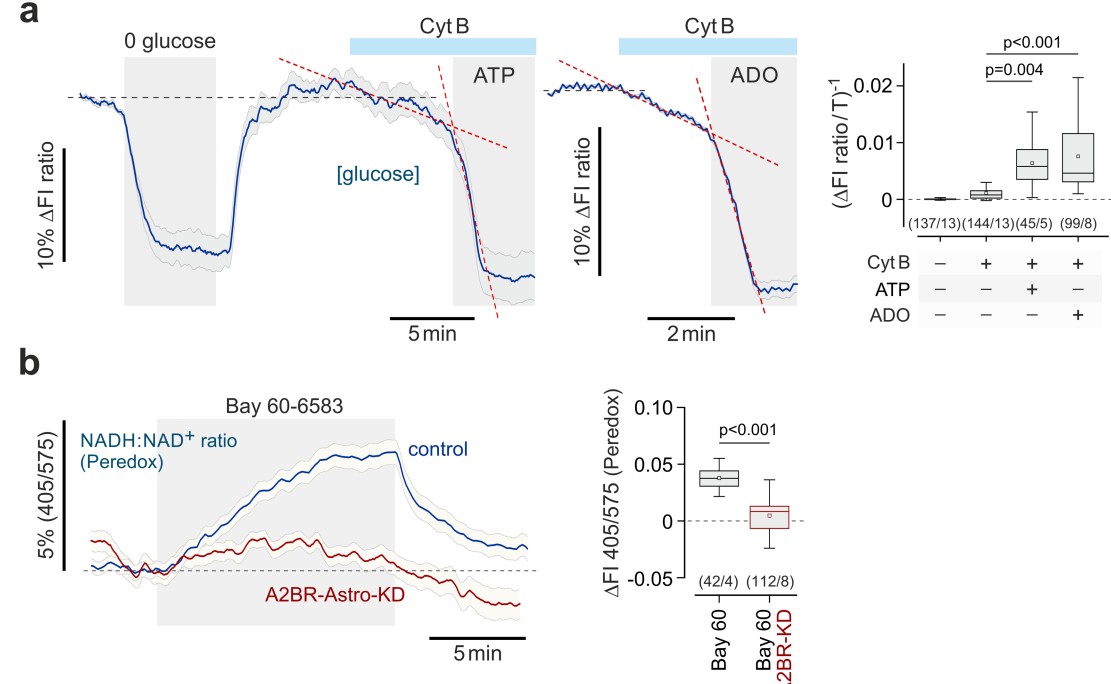

**Extended Data Fig. 4 | Activation of adenosine A2B receptors stimulates astrocyte glucose metabolism.** (**a**) Representative traces and summary data illustrating ATP- and adenosine-induced changes in astrocyte glycolytic rate, recorded using the fluorescent sensor of glucose FLIP[12]glu-700μΔ6 (FI ratio Citrine/CFP) in the presence of glucose transporter inhibitor Cytochalasin B (CytB, 20 μM). The slope of the sensor signal decline under conditions of glucose transporter blockade was used to calculate the glycolytic rate. The example also illustrates the response to the removal of extracellular glucose, demonstrating the functionality of the sensor. *P* values, one-way ANOVA followed by Sidak's post hoc test; (**b**) Representative traces and summary data illustrating the effect of A2B receptor agonist Bay 60-6583 on cytosolic

NADH-NAD⁺ redox state (reporting glucose consumption) recorded using *Peredox* sensor in cultured astrocytes. A2B receptor agonist had no effect on astrocyte glucose consumption in conditions of A2B receptor deletion (A2BR-Astro-KD). *P* value, two-tailed Student's *t*-test. Traces illustrate averaged (mean ± s.e.m.) recordings from several individual cells in a representative experiment. In the box-and-whisker plots, the central dot indicates the mean, the central line indicates the median, the box limits indicate the upper and lower quartiles, and the whiskers extend to 1.5 IQR from the quartiles. Numbers in parentheses indicate the number of individual cells/number of separate cultures, prepared from the same number of animals.

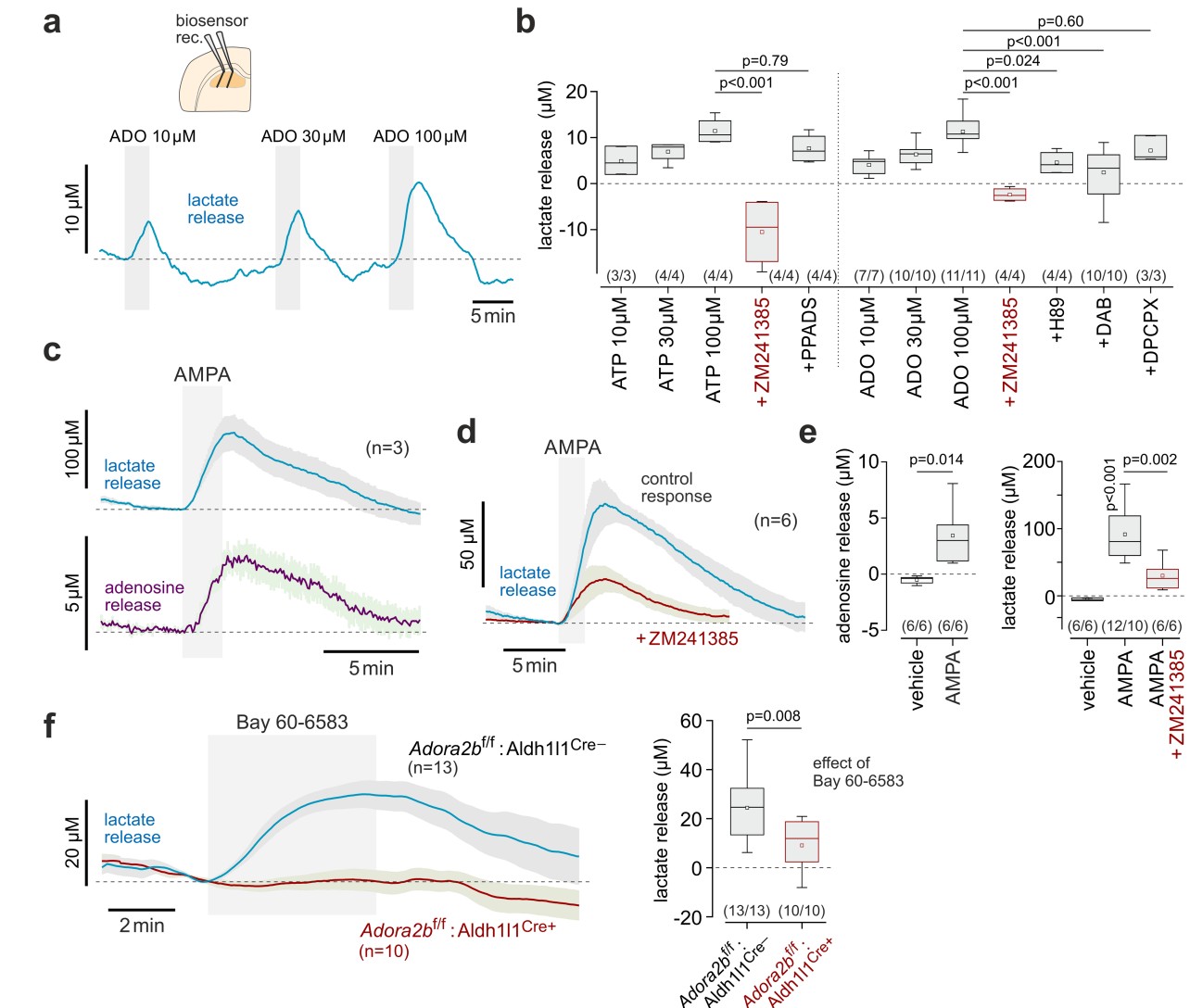

**Extended Data Fig. 5 | Activation of adenosine A2B receptors induces the release of lactate.** (**a**) A representative recording of changes in lactate microelectrode biosensor current illustrating facilitated release of lactate in response to adenosine, recorded in an acute brain slice; (**b**) Summary data illustrating peak increases in the release of lactate recorded in acute brain slices in response to ATP, ATP in the presence of ZM241385 (10 μM) or PPADS (30 μM); adenosine, and adenosine in the presence of ZM241385 (10 μM), H89 (30 μM), an inhibitor of glycogen phosphorylase DAB (150 μM), or A1 adenosine receptor antagonist DPCPX (1 μM). *P* values, one-way ANOVA followed by Sidak's post hoc test; (**c**) Averaged (means ± s.e.m.) traces illustrating the release of adenosine and lactate (recorded simultaneously using microelectrode biosensors) in response to neuronal network activation induced by application of AMPA (5 μM); (**d**) Averaged (means ± s.e.m.) traces and (**e**) summary data

illustrating the effect of A2 receptor blockade with ZM241385 (10 μM) on AMPA-induced lactate release in acute brain slices. *P* value, two-tailed Student's *t*-test; (**f**) Averaged (means ± s.e.m.) traces and summary data illustrating the release of lactate induced by A2B receptor agonist Bay 60-6583 (1 μM) in acute hippocampal slices prepared from the brains of *Adora2b*^flox/flox^:Aldh1l1^Cre-^ and *Adora2b*^flox/flox^:Aldh1l1^Cre/ERT2+^ mice treated with tamoxifen. The effect of Bay 60-6583 on lactate release was markedly reduced by genetic deletion of A2B receptors in astrocytes. *P* value, two-tailed Student's *t*-test. In the box-and-whisker plots, the central dot indicates the mean, the central line indicates the median, the box limits indicate the upper and lower quartiles, and the whiskers extend to 1.5 IQR from the quartiles. Numbers in parentheses indicate the number of independent slice experiments/number of animals per experimental group.

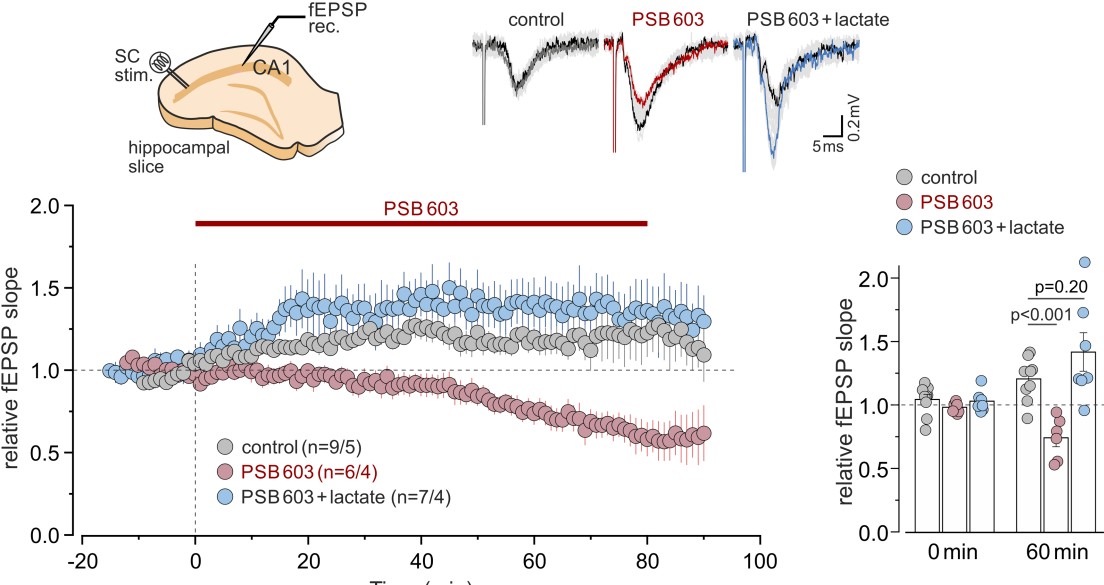

**Extended Data Fig. 6 | Adenosine A2B receptor-mediated signalling maintains synaptic function in the hippocampus.** Monitoring the excitatory transmission in the CA3-CA1 pathway (acute hippocampal slices). The graph illustrates the relative changes in the averaged field excitatory postsynaptic potential (fEPSP) slope recorded in the CA1 area in response to stimulation of Schaffer collateral (SC) fibers in the absence and presence of A2B receptor antagonist PSB 603 (10 μM). A2B receptor blockade led to a gradual decline in fEPSP slope; this effect of A2B receptor blockade was prevented by supplemental lactate (5 mM). Analysis was performed for the first fEPSP evoked in a low-frequency pulse train (5 pulses at 20 Hz). Data are presented as individual values and means ± s.e.m. Numbers in parentheses indicate the number of independent slice experiments/number of animals per experimental group. P values, one-way ANOVA followed by Sidak's post hoc test.

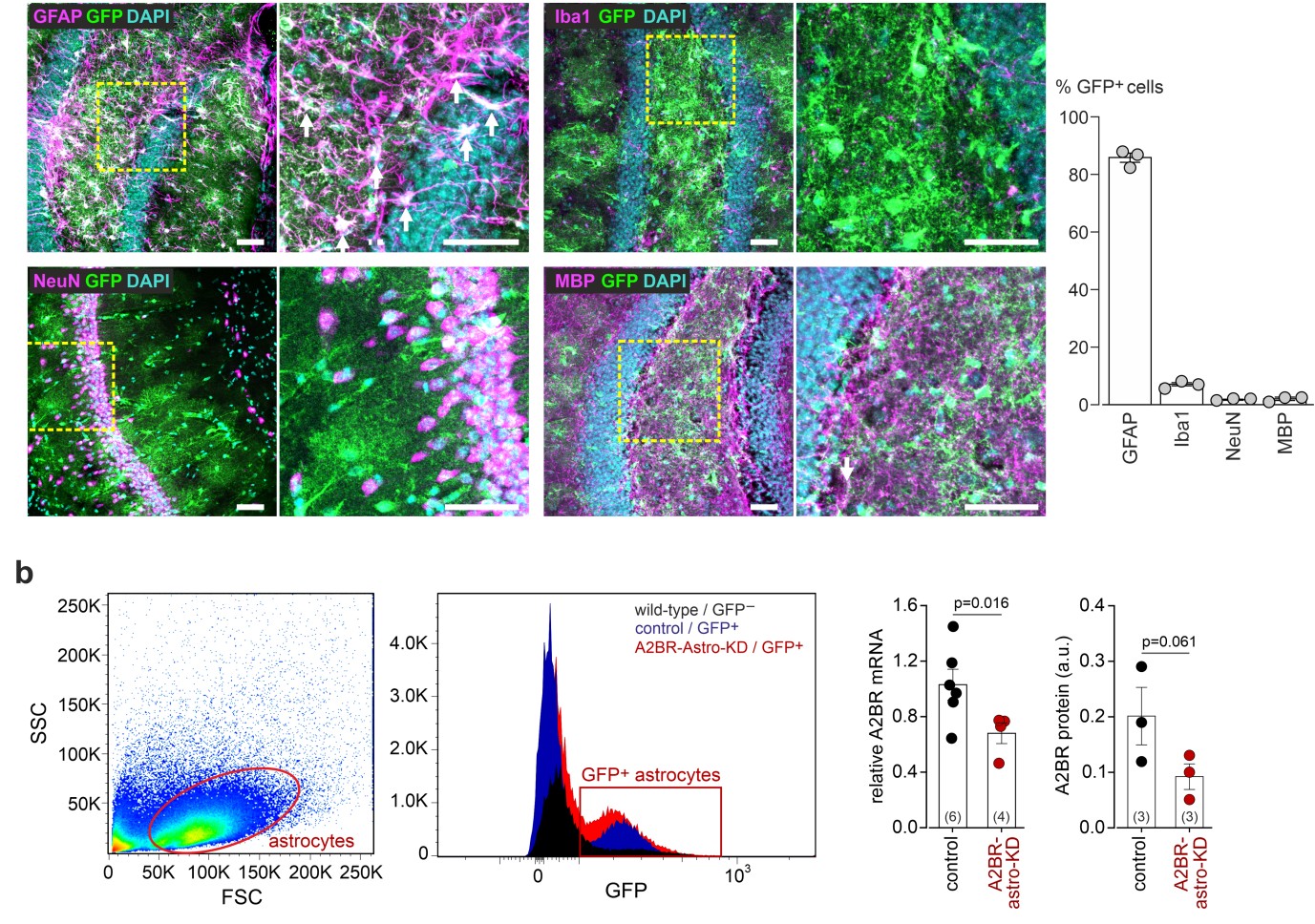

**Extended Data Fig. 7 | Deletion of A2B receptors in hippocampal astrocytes.** (**a**) Representative immunofluorescence images showing transgene expression after the microinjections of AAV5-Gfap-eGFP-iCre vector into the hippocampus in mice. Summary data illustrate quantification of brain cells expressing GFP and identified by immunohistochemical labeling to also express either glial fibrillary acidic protein (GFAP), Ionized calcium binding adaptor molecule 1 (Iba1), neuronal nuclear antigen (NeuN), or myelin basic protein (MBP) (n = 3 mice). Gfap promoter displayed high selectivity for astrocytes. Scale bars = 50 µm; (**b**) Hippocampal astrocytes of *Adora2b*^flox/flox mice were transduced to express iCre recombinase (microinjections of AAV5-Gfap-eGFP-iCre vector) or control transgene (microinjections of AAV5-Gfap-eGFP vector), followed by isolation of transduced astrocytes by FACS. Gating strategy used for isolation of transduced cells: hippocampal astrocyte population was first gated based on morphology by size and complexity (FSC and SSC) and then gated based on the expression of GFP (control/GFP^+: *Adora2b*^flox/flox mice injected with AAV5-Gfap-eGFP; A2BR-Astro-KD/GFP^+: *Adora2b*^flox/flox mice injected with AAV5-Gfap-eGFP-iCre; brain tissue of wild-type naïve mice was used as a negative control [wild-type/GFP-]). Expression of iCre recombinase in hippocampal astrocytes of *Adora2b*^flox/flox mice reduced A2B receptor expression. Data are presented as individual values and means ± s.e.m. Numbers in parentheses indicate the numbers of animals per experimental group. *P* values, one-tailed Student's *t*-test.

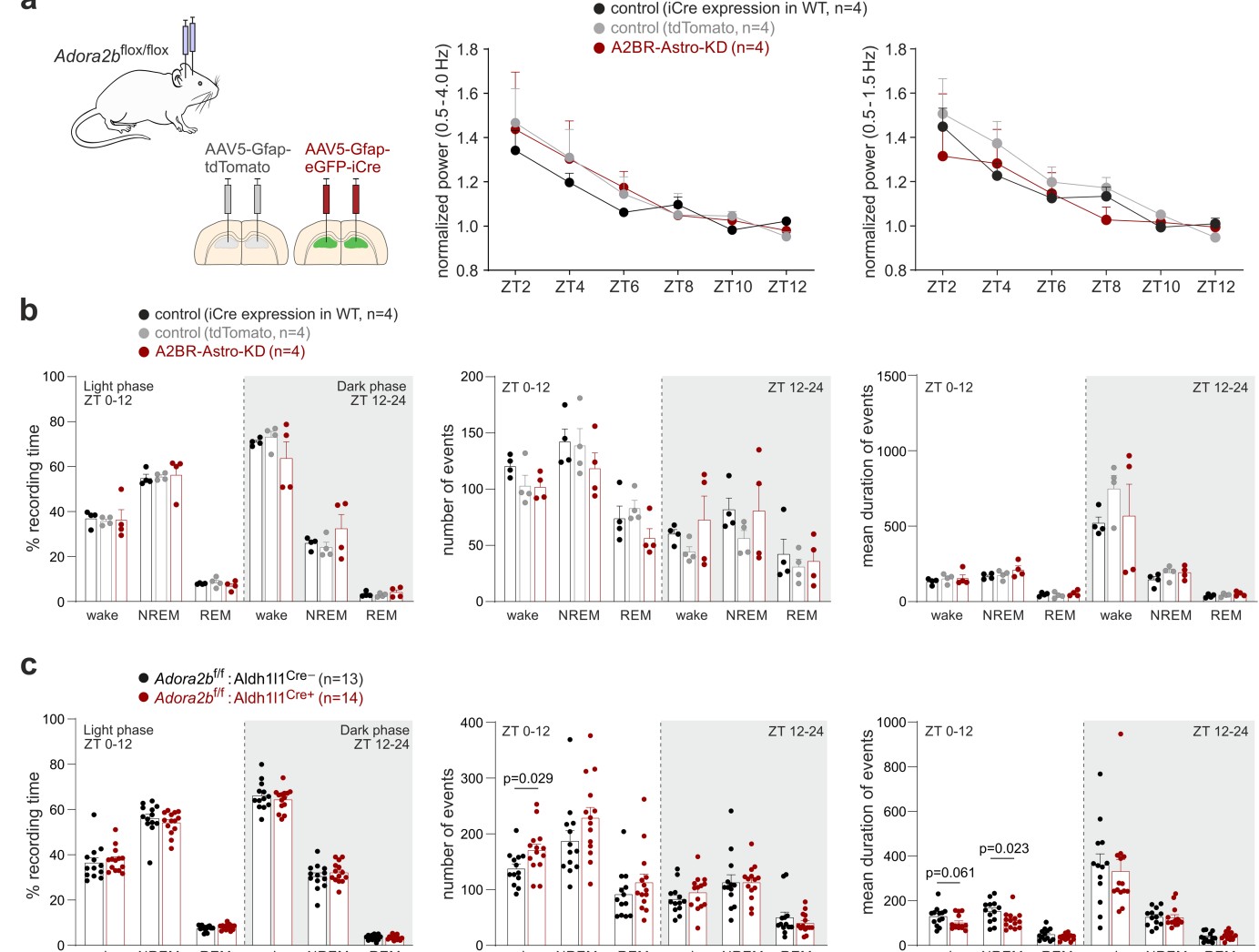

**Extended Data Fig. 8 | Sleep/wake architecture under conditions of A2B receptor deletion in astrocytes. (a)** Bilateral targeting of hippocampal astrocytes of *Adora2b*flox/flox or wild-type (WT) mice to express iCre recombinase or reporter protein (tdTomato). Summary data illustrating slow wave activity (0.5-4.0 Hz) and low-frequency slow wave activity (0.5-1.5 Hz) during NREM sleep across the light phase in two control groups of mice and animals with A2B receptor knockdown in hippocampal astrocytes (A2BR-Astro-KD). Data are presented as mean values ± s.e.m.; **(b)** Summary data illustrating the percentage of time spent in wakefulness, NREM (non-rapid eye movement) sleep, or REM (rapid eye movement) sleep during the 24-hour recordings as well as the number and duration of wake, NREM, and REM sleep episodes in two groups of control mice and A2BR-Astro-KD mice. Sleep/wake architecture and accumulation of sleep pressure were not affected when A2B receptor deletion was limited to the astrocytes of the hippocampus; **(c)** Summary data illustrating the percentage of time spent in wake, NREM or REM sleep during the 24 h recordings and the number and duration of wake, NREM and REM sleep episodes in *Adora2b*flox/flox: Aldh1l1Cre- and *Adora2b*flox/flox: Aldh1l1Cre/ERT2+ mice treated with tamoxifen. Deletion of A2B receptors in brain astrocytes led to fragmentation of sleep and wake during the light (resting) phase, with shorter periods of NREM sleep and increased number of wakefulness events. In panels (b) and (c), data are presented as individual values and means ± s.e.m. Numbers in parentheses indicate the numbers of animals per experimental group. *P* values, two-tailed unpaired *t*-test.

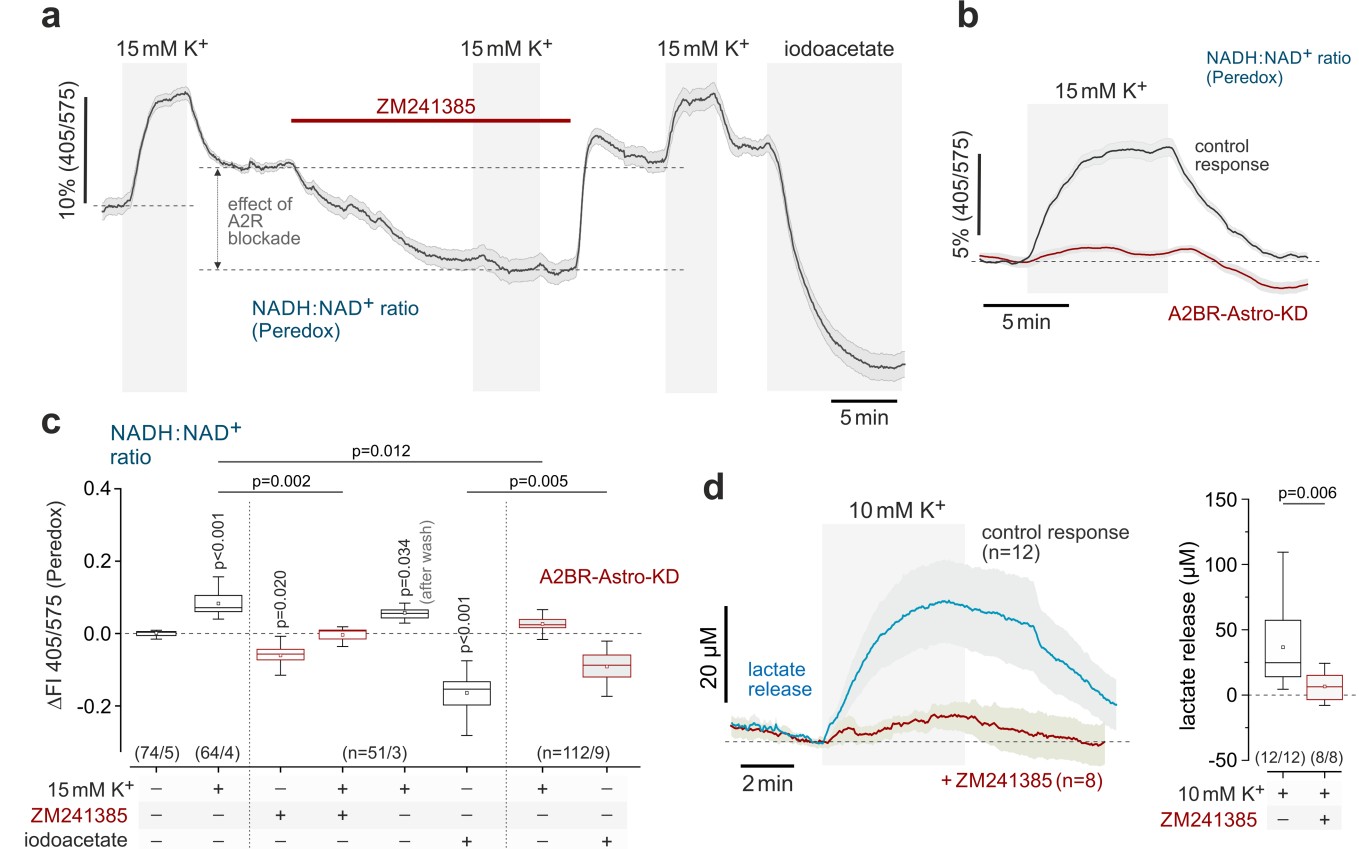

**Extended Data Fig. 9 | The effects of K+ on astrocyte glycolysis and lactate release are dependent on adenosine A2B receptor-mediated signalling.**
(**a**) Representative trace and (**c**) summary data illustrating changes in cytosolic NADH-NAD+ redox state in response to increases in extracellular [K+] (from 3 mM to 15 mM) in cultured astrocytes, recorded in the absence and presence of A2 receptor antagonist ZM241385 (10 µM). To estimate the cytosolic level of glycolysis-derived NADH:NAD+, the cells were treated with an inhibitor of glycolysis iodoacetate (1 mM) at the end of each experiment; (**b**) Traces illustrating changes in cytosolic NADH-NAD+ redox state in response to 15 mM extracellular [K+] in *Adora2b*flox/flox astrocytes transduced to express eGFP (control) and in astrocytes of *Adora2b*flox/flox mice transduced to express iCre recombinase (A2BR-Astro-KD). In panels (a) and (b), traces illustrate averaged (mean ± s.e.m.) recordings from several individual cells in a representative

experiment. In panel (c), numbers in parentheses indicate the number of individual cells/number of separate cultures, prepared from the same number of animals. *P* values, one-way ANOVA followed by Sidak's post hoc test; (**d**) Averaged (means ± s.e.m.) traces and summary data illustrating facilitated release of lactate in response to 10 mM extracellular [K+] recorded using lactate biosensors in acute hippocampal slices of mice in the absence and presence of A2 receptor antagonist ZM241385 (10 µM). *P* value, two-tailed Mann-Whitney test. In panel (d), numbers in parentheses indicate the number of independent slice experiments/number of animals per experimental group. In the box-and-whisker plots, the central dot indicates the mean, the central line indicates the median, the box limits indicate the upper and lower quartiles, and the whiskers extend to 1.5 IQR from the quartiles.

# Reporting Summary

## Statistics

For all statistical analyses, confirm that the following items are present in the figure legend, table legend, main text, or Methods section.

| n/a | Confirmed | |
|---|---|---|
| ☐ | ☒ | The exact sample size (*n*) for each experimental group/condition, given as a discrete number and unit of measurement |
| ☐ | ☒ | A statement on whether measurements were taken from distinct samples or whether the same sample was measured repeatedly |
| ☐ | ☒ | The statistical test(s) used AND whether they are one- or two-sided<br>*Only common tests should be described solely by name; describe more complex techniques in the Methods section.* |
| ☒ | ☐ | A description of all covariates tested |
| ☐ | ☒ | A description of any assumptions or corrections, such as tests of normality and adjustment for multiple comparisons |
| ☐ | ☒ | A full description of the statistical parameters including central tendency (e.g. means) or other basic estimates (e.g. regression coefficient) AND variation (e.g. standard deviation) or associated estimates of uncertainty (e.g. confidence intervals) |
| ☐ | ☒ | For null hypothesis testing, the test statistic (e.g. *F*, *t*, *r*) with confidence intervals, effect sizes, degrees of freedom and *P* value noted<br>*Give P values as exact values whenever suitable.* |
| ☒ | ☐ | For Bayesian analysis, information on the choice of priors and Markov chain Monte Carlo settings |
| ☒ | ☐ | For hierarchical and complex designs, identification of the appropriate level for tests and full reporting of outcomes |
| ☒ | ☐ | Estimates of effect sizes (e.g. Cohen's *d*, Pearson's *r*), indicating how they were calculated |

*Our web collection on statistics for biologists contains articles on many of the points above.*

## Software and code

Policy information about availability of computer code

| | |
|---|---|
| Data collection | Axon-pClamp (version 10.2); Spike2 (Cambridge Electronic Design Ltd, version 7); IQ3 imaging (version 6.3); Micromanager (ImageJ, version 1.4.23); Olympus FluoView (version 4); Sirenia Acquisition (Pinnacle, version 2.2.7); Skyline (MacCoss Lab Software, version 23.1); VisionWorksLS (Bio-Rad). |
| Data analysis | Axon-pClamp (version 10.2); GraphPad Prism (version 8); Image J (version 1.52P); MetaboAnalyst R package (version 5.0); MetaboAnalyst R package (version 5.0); Origin 2019 (version 9.6); ropls R package (version 1.26.4); Seurat package (version 4.2.2, "Innocent and Trusting") in R (version 3.6.0, "Planting of a Tree"); Skyline (version 23.1); SleepSign (Kissei Comtec, version 3.0); Spike2 (version 7); Viewer III software (Biobserve, version 3). |

For manuscripts utilizing custom algorithms or software that are central to the research but not yet described in published literature, software must be made available to editors and reviewers. We strongly encourage code deposition in a community repository (e.g. GitHub). See the Nature Portfolio guidelines for submitting code & software for further information.

## Data

Policy information about availability of data

All manuscripts must include a data availability statement. This statement should provide the following information, where applicable:
- Accession codes, unique identifiers, or web links for publicly available datasets
- A description of any restrictions on data availability
- For clinical datasets or third party data, please ensure that the statement adheres to our policy

The data that support the findings in this study are included within the Supplementary Material. The source data underlying Figs. 1b, e, f, 2c, f, g, i, 3c, d, f, g, h, i, l, 4b, c, e, f, g, h, 5c, d, f, i, and Extended Data Figs. 1e, 2d, e, 3c, 4a, b, 5b, c, d, e, f, 6, 7a, b, 8a, b, c, and 9c, d are provided as Source Data files. Single cell RNAseq source data underlying Fig.2a,b are available from a publicly available database (http://mousebrain.org/). The isotopically quantified LC–MS/MS data are deposited in MassIVE and available via the link https://doi.org/10.25345/C5X05XQ2B (accession number MSV000094445).

## Human research participants

Policy information about studies involving human research participants and Sex and Gender in Research.

| Reporting on sex and gender | Not applicable. |
|---|---|
| Population characteristics | Not applicable. |
| Recruitment | Not applicable. |
| Ethics oversight | Not applicable. |

Note that full information on the approval of the study protocol must also be provided in the manuscript.

# Field-specific reporting

Please select the one below that is the best fit for your research. If you are not sure, read the appropriate sections before making your selection.

☒ Life sciences          ☐ Behavioural & social sciences          ☐ Ecological, evolutionary & environmental sciences

For a reference copy of the document with all sections, see nature.com/documents/nr-reporting-summary-flat.pdf

# Life sciences study design

All studies must disclose on these points even when the disclosure is negative.

| Sample size | The paper describes the results of the experiments performed using in vitro (primary culture, organotypic and acute brain slices) and in vivo animal preparations. Power calculations for the in vivo animal studies are described below. The significance was set at 0.05 and the beta was set at 0.20. From years of relevant research experience, we expect to detect a significant difference with a minimum of 5 animals per experimental group, if the treatments cause differences between means that are as large as 2.25 standard deviations (SD), likely to be a physiologically significant difference. If the difference is as small as 1.75 SD, we increase sample sizes to 9. In the in vitro studies, the data in individual experiments were collected from a minimum of 3 different slices or cultures (termed 'samples' in the Source Data files) prepared from the same number of different animals. In each experimental preparation (sample), the recordings were made from several individual cells. This experimental design is based on many years of research experience in conducting experiments of this type. Statistical analysis was conducted on the collected data, taking into account the number of cells in each experimental sample and the number of biologically distinct samples. No specific statistical tools were used to predetermine the sample size. The sample sizes were chosen to provide sufficient statistical power to detect biologically significant differences between experimental treatments, considering the experimental design and objectives of the study. All key experiments supporting the main conclusions of the study were repeated several times in samples prepared from at least 8 different animals obtained from different litters. |
|---|---|
| Data exclusions | No data were excluded from the analysis |
| Replication | All experiments were independently conducted at least five times. All attempts to replicate the findings were consistently successful in the present study. |
| Randomization | Randomization is not relevant to the present study, since experimental animals were obtained from genetically homogeneous colonies and then assigned to different experimental groups according to the treatment/genetic status. |
| Blinding | The experiments conducted in acute brain slices were performed by the investigator who was blinded to the treatment/genetic status of the study animals. The investigator undertaking behavioral studies was not blinded , but the data analysis was performed by the individuals blinded to the condition, treatment and/or genetic status of experimental subjects and to the identity of the experimental groups. In the metabolomics study, sample separation, mass spectrometry and data analysis were performed by the investigators who were blinded to the |

identity of the samples.

# Reporting for specific materials, systems and methods

We require information from authors about some types of materials, experimental systems and methods used in many studies. Here, indicate whether each material, system or method listed is relevant to your study. If you are not sure if a list item applies to your research, read the appropriate section before selecting a response.

## Materials & experimental systems

| n/a | Involved in the study |
|-----|-----------------------|
| ☐ | ☒ Antibodies |
| ☒ | ☐ Eukaryotic cell lines |
| ☒ | ☐ Palaeontology and archaeology |
| ☐ | ☒ Animals and other organisms |
| ☒ | ☐ Clinical data |
| ☒ | ☐ Dual use research of concern |

## Methods

| n/a | Involved in the study |
|-----|-----------------------|
| ☒ | ☐ ChIP-seq |
| ☒ | ☐ Flow cytometry |
| ☒ | ☐ MRI-based neuroimaging |

## Antibodies

| Antibodies used | Anti-adenosine A2B receptor antibody (Merck Millipore, Cat # AB1589P); anti-actin antibody (Cell Signaling Technologies, Cat # 3700, clone 8H10D10); anti-chicken antibody AlexaFluor 488 (ThermoFisher, Cat #A-11039); anti-chicken antibody AlexaFluor 568 (ThermoFisher, Cat. # A-11041); anti-rabbit antibody AlexaFluor 568 (Abcam, Cat #175470); rabbit anti-GFAP antibody (Proteintech,Cat # 23935-1-AP); anti-GFP antibody (Aves Labs Cat. # GFP-1020); chicken anti-GFP antibody (Abcam  Cat#AB13970); rabbit anti-Iba1 antibody (GeneTex, Cat #GTX100042); anti-rabbit -HRP antibody (Santa Cruz, Cat. # sc-2054); anti-mouse-HRP antibody (Santa Cruz, Cat. # sc-2005); rabbit anti-MBP antibody (ThermoFisher, Cat #MA5-35074, clone 1Z9R5); rabbit anti-NeuN antibody (Abcam, Cat #AB236870, clone EPR21906). |
|---|---|
| Validation | Anti-adenosine A2B receptor antibody was used for the Western blots which showed reduced expression of A2B receptor in cells of Adora2Bf/f animals transduced to express Cre recombinase. This was expected and consistent with the data showing reduced expression of A2B receptor at mRNA level. Details of antibody validation are available on the manufacturer website: https://www.merckmillipore.com/GB/en/product/Anti-Adenosine-A2b-Receptor-Antibody,MM_NF-AB1589P?ReferrerURL=https%3A%2F%2Fwww.google.com%2F<br>Anti-GFP antibody was used for immunofluorescence detection and amplification of GFP expression in hippocampal astrcoytes of Adora2Bf/f mice transduced to express Cre recombinase. Details of antibody validation can be found on the manufacture's website: https://www.aveslabs.com/products/anti-green-fluorescent-protein-antibody-gfp; https://www.abcam.com/en-no/products/primary-antibodies/gfp-antibody-ab13970.<br>Anti-actin antibody was thoroughly validated for the use in western blot in cell/tissue samples of mice and other species.  Details of antibody validation can be found on the manufacture's website: https://www.cellsignal.com/products/primary-antibodies/b-actin-8h10d10-mouse-mab/3700). The use of this antibody has been reported in more than 4,000 publications, from which 314 studies used the antibody to perform western blot analysis in Mus musculus samples.<br>The following primary and secondary antibodies were validated by the manufacturers: rabbit anti-GFAP (https://www.ptglab.com/products/GFAP-Antibody-23935-1-AP.htm); rabbit anti-MBP (https://www.thermofisher.com/antibody/product/MBP-Antibody-clone-ARC0535-Recombinant-Monoclonal/MA5-35074); rabbit anti-NeuN (https://www.abcam.com/products/primary-antibodies/neun-antibody-epr21906-neuronal-marker-ab236870.html); rabbit anti-Iba1 (https://www.genetex.com/Product/Detail/Iba1-antibody/GTX100042); anti-chicken AlexaFluor 488 (https://www.thermofisher.com/antibody/product/Chicken-anti-Rabbit-IgG-H-L-Cross-Adsorbed-Secondary-Antibody-Polyclonal/A-21441); anti-rabbit AlexaFluor 568 (https://www.abcam.com/en-dk/products/secondary-antibodies/donkey-rabbit-igg-h-l-alexa-fluor-568-ab175470). |

## Animals and other research organisms

Policy information about studies involving animals; ARRIVE guidelines recommended for reporting animal research, and Sex and Gender in Research

| Laboratory animals | Young adult C57Bl/6J, Adora2bflox/flox and Aldh1l1Cre/ERT2 mice and their crosses (3-4 mo old) and pups (p2-10) of both sexes; young Sprague-Daweley rats (P21-25) and pups (p2-10) of both sexes. |
|---|---|
| Wild animals | The study did not involve any wild animals |
| Reporting on sex | Animals of both sexes were used in the experiments. Data analysis showed no differences in the effects of experimental condition, treatment or genetic status; therefore, the data obtained in male and female animals were pulled. Detailed sex-based analysis was not performed |
| Field-collected samples | The study did not involve any field collected samples |
| Ethics oversight | All animal experimentations were performed in accordance with the European Commission Directive 2010/63/EU (European Convention for the Protection of Vertebrate Animals used for Experimental and Other Scientific Purposes) and the UK Home Office |

Animals (Scientific Procedures) Act (1986) with project approval from the Institutional Animal Care and Use Committee of the University College London.

Note that full information on the approval of the study protocol must also be provided in the manuscript.

