## [Peer Review file · Nature]

Manuscript Title: Adenosine signaling to astrocytes coordinates brain metabolism and function

Editorial Notes:

Redactions – Third Party Material

Reviewer Comments & Author Rebuttals

Reviewer Reports on the Initial Version:

Referees' comments:

Referee #1 (Remarks to the Author):

The paper by Gourine and colleagues nicely shows that the metabolic coupling between neurons and astrocytes to maintain adequate energy supply to neurons, especially during low glucose availability and high neuronal firing, is mediated by neuronal adenosine release activating astrocytic A2B receptors. The activation of these receptors triggers intracellular PKA and cAMP cascades that leads to enhanced glucose metabolism and lactate availability. Overall, the paper is well written and the figures for the data are clear. My major concerns focus on novelty for Nature and aspects of rigor and controls.

Major concerns

1. This study follows many years of work on the lactate shuttle as a link between neuronal activity and energy supply via astrocytes. The various papers leading to this hypothesis are well cited in the current manuscript and this mechanism is probably one of the most well established mechanisms in the whole astrocyte field. In the current study the authors have used pharmacology and some genetic approaches to show that coupling between neurons and astrocytes occurs via neuronal adenosine release that activates A2B receptors on astrocytes. This is interesting, but identifying the link is incremental and not at the level of advance expected for a Nature paper. On page 8, the authors also do an excellent account of discussing past mechanisms that have been proposed and why they did not work out. Hence, the major advance in the current study is the proposal of another mechanism involving adenosine. Although this is interesting to specialists, it is not broadly appealing and not highly novel at the level expected of a Nature paper. They have added interesting details to a mechanism that is well known. I recognize this is a judgment call.

2. The author's evidence that astrocytes express A2B receptors is based on past RNA-seq data and pharmacological experiments. All of the key experiments for control and following A2B reduction need to be repeated with additional readouts of A2B receptors such as antibody labelling or in situ hybridization. These controls are critical to know that the reductions of A2B receptors did happen as the data suggest, but also to know what level of reduction happened in each experiment that led to the functional readouts. Right now, there is no direct way of knowing how much A2B was reduced. Such experiments should also assess A2B and A2A expression in neurons and oligodendrocytes. Oligos are important as they express A2B receptors and the AAV approaches using the GFAP promoter are expected to target astrocytes and some oligodendrocytes. Thus there is no direct evidence that A2B has been reduced only in astrocytes – although this is the simplest explanation. These types of control experiments are essential and should be reported in detail with Extended data.

3. The Gfap Cre virus studies use tdTomato controls. However, these controls do not show that the resultant A2B reduction was limited to astrocytes. Thus a series of experiments are needed to carefully explore the cell specificity of the Cre genetic interventions as being astrocyte specific. Without controls key aspects of the study are not on solid foundations. These types of control experiments are essential and should be reported in detail in Extended data.

4. I found the finishing comments on caffeine use and brain function in humans to be a stretch. This is highly speculative and not related to the main findings in the study. This type of speculation should be reserved for a review.

5. There is little information on how FRET was assessed in the reporters. From the very brief methods provided it seems they measure CFP and YFP emission after 2p excitation of CFP at 860nm. How did they convert that to FRET? If they measure only the two emissions they should not call it FRET. Changes in CFP and YFP emission should be shown. In sum, much more information and controls are needed for the optical measurements in order to understand what was done.

6. I do not follow the logic of performing experiments in Fig 1a-c in slices and the pharmacology in Fig 1d-g in cell culture. It would be more rigorous to perform the pharmacology in Fig 1 and all other figures in brain slices because this is closest to the final metrics of LTP and discrimination index.

7. Additional behaviors should be performed on the A2B astrocyte KO mice to assess other aspects of learning and memory and general brain function. If the mechanism of neuron to astrocyte metabolic coupling via astrocytes is important then perhaps multiple behavioral domains of the mice are altered. It would be surprising if only learning was changed. Or are the authors saying that there is something special about this mechanism for the hippocampus as implied by the title? If that is the case why is it special for the hippocampus because metabolic coupling is needed throughout the brain?

Referee #2 (Remarks to the Author):

This work reports that the astrocytic A2B receptor and activation by adenosine and ATP is involved in metabolic activation and in behavior. Whether the involvement in the behavioral changes is mediated by astrocytic metabolism may not be deduced from the present data. Most importantly, this pathway is not new, as claimed by the authors. There are technical issues regarding the characterization of metabolism that require solving with control experiments. There are also design issues that weaken the proposed causal chain involving ATP, adenosine, astrocytic glycolysis, lactate and behavior. Critically, the role of the A2B receptor on the stimulation of astrocytic metabolism by Schaffer collateral stimulation was not tested. In addition, there are significant omissions and misrepresentations of the literature that are misleading regarding novelty and relevance. Carefully demonstrating and characterizing the effects of adenosine on glycolytic rates would be interesting and important. There are technical issues, however, see below. Also, it would be key to put this pathway (which is not new per se) in the context of other known pathways. When, where, under what conditions would the adenosine pathway be important? What is the specific role of this pathway?

More specifically:

1. Novelty. The authors convey the message that a novel signaling mechanism is being presented, providing a molecular/cellular explanation for the phenomenon of activity-dependent aerobic glycolysis in brain tissue. Whereas the use of tissue slices and behavioral analysis and the specific role of A2B receptors are valuable contributions, authors should be aware that previous work by several laboratories had shown modulation of astrocytic glycolysis by ATP. For example, using Peredox, Koehler et al., *Glia* 2018 demonstrated that ATP increases NADH in astrocytes in culture and in slices, with similar kinetics as found presently (DOI: 10.1002/glia.23504). Horvat et al., *Cell Calcium* 2021, showed a fast increase in astrocytic lactate in response to ATP (doi.org/10.1016/j.ceca.2021.102368.), whereas Juaristi et al., *Glia* 2018, reported accumulation of pyruvate and inhibition of oxygen consumption in astrocytes exposed to ATP (DOI: 10.1002/glia.23574). Also, the impact of adenosine on metabolism was addressed in older papers that would need to be considered (Bruns RF. *Nucleosides Nucleotides* 10: 931–943, 1991; Magistretti PJ et al. *J Neurosci* 6: 2558–2562, 1986 and then Allaman et al., *Am J Physiol Cell Physiol* 284: C696–C704, 2003).

2. Technical. It is shown that Schaffer collateral stimulation lead to increased cAMP and PKA activity in astrocytes and that the response was insensitive to glutamate receptor blockage. Thus, the phenomenon is originated in a pre-synaptic signal. As other neuronal signals that induce aerobic glycolysis in astrocytes originate in the post-synaptic compartment (e.g., K⁺ and nitric oxide), the presumably presynaptic origin of the ATP should be discussed. However, further experiments are required to establish this mechanism. Firstly, it is known that electrical stimulation induces local permeabilization of cells, with the ensuing release of ATP. Conceivably, some of this ATP and/or adenosine may have reached the astrocytes diffusing from the stimulation area. Whereas extracellular ATP is toxic and stimulates astrocytic glycolysis (see below), adenosine provokes hyperexcitability in the hippocampus Rombo et al., *Hippocampus* 25:566–580, 2015. Please provide a demonstration that cAMP and PKA changes require neuronal action potentials, for example using TTX. Along the same line, if the ATP is actually released by neurons (not clear and not substantiated by the data), then the astrocytic response may be sensitive to presynaptic inhibition, e.g., blockers of synaptic vesicle fusion/recycling.

3. Design I. It is puzzling that there is no mention of extracellular adenosine and lactate recordings in response to Schaffer collateral stimulation. A positive result in such key experiment, namely a quick release of adenosine that precedes the release of lactate would support the proposed link between neuronal activity to astrocytic metabolism. Such temporal resolution is well within the capability of modern electrodes. If the role of A2B in behavior is actually mediated by astrocytic glycolysis, the release of lactate should be deficient in the A2B knockout, but not the accumulation of adenosine.

4. Fig. 3e, which shows adenosine and lactate release in response to post-synaptic activation with AMPA does not support the current story. Rather, the effect of AMPA indicates that a post-synaptic signal is at work, likely K⁺ and/or NO (see below).

5. Design II. Along a similar line, is also unclear why instead of Schaffer collateral stimulation, astrocytic metabolism was probed with ATP and adenosine. Both ATP and adenosine are bound not

only to affect astrocytes and many ways, including the opening of hemichannels (ref 32), but also neurons, with the secondary release of other mediators that are known to affect metabolism, like glutamate, K⁺, NO, NH₄⁺, etc. As above, demonstration of inhibited astrocytic NADH and glucose responses to Schaffer collateral stimulation in the A2B deficient animal support the proposed signaling pathway. Such type of experiment was instrumental to demonstrate the role of the NBCe1 bicarbonate cotransporter in the activation of astrocytic glucose consumption in response to Schaffer collateral stimulation (Ruminot et al., *JCBFM* 2017, DOI: 10.1177/0271678X17737012).

6. Belittling and omission of other signals. Page 8, third paragraph. This paragraph is misleading as it neglects relevant literature and contains statements that are problematic. i. It is stated that the response to glutamate is “rather slow”, citing ref. 43. However, ref. 43 shows a rapid stimulation of astrocytic glucose transport, which delivers glucose to glycolysis. ii. It is stated that “astroglial glycolysis is driven by astroglial depolarization, bicarbonate entry and intracellular alkalinization while citing ref. 43. This statement is not correct. The paper showing the astrocytic role of depolarization, bicarbonate entry and alkalinization is actually Ruminot et al. *J. Neurosci.* 2011, doi.org/10.1523/JNEUROSCI.5311-10.2011. iii. The role of K⁺ is played down by a vague mention to “gene sequencing data”, but no information is provided. iv. The role of K⁺ is played down again by citing “in vivo evidence that the neuronal activity-related increases in extracellular [K⁺] are relatively small (ref. 45)”. However, the study of ref. 45 measured cortex-wide K⁺ fluctuations. Much larger changes in local extracellular K⁺, reaching up to 10 mM, have been detected in response to somatic stimulation (e.g. Heinemann et al *Exp Brain Res* 79: 283-292, 1990). More recently, activity-dependent local extracellular K⁺ was also estimated at 10 mM using a voltage-sensitive probe in astrocytic PAPs (Ambruster et al., *Nat Neurosci* 2022, doi.org/10.1038/s41593-022-01049-x). It is well established that interstitial K⁺ reaches levels capable of modulating astrocytic glycolysis. v. Another key omission is the previous demonstration that extracellular K⁺, acting via the NBCe1 bicarbonate cotransporter, mediates the activation of astrocytic glucose consumption in response to Schaffer collateral stimulation (Ruminot et al., *JCBFM* 2017, DOI: 10.1177/0271678X17737012). vi. Additional relevant signals proposed to participate in activity-dependent modulation of astrocytic glycolysis are nitric oxide (San Martin et al., *JBC* 2017, doi.org/10.1074/jbc.M117.777243) and NH₄⁺ (Lerchundi et al. *PNAS* 2015, doi.org/10.1073/pnas.150825911).

7. Other signals II. The participation of astrocytic A2B receptors in behavior seems convincing. However, it is not clear that these receptors are the main mediators. There are multiple example of signaling pathways that play permissive roles on metabolism. Relevant examples aer the permissive roles of the NBCe1 and the Na⁺ pump on the glycolytic responses of astrocytes to K⁺, glutamate and ATP shown by Koehler et al., *GLIA* 2018 (DOI: 10.1002/glia.23504) and ref. 43. Looking at the present data in the context of the literature, a permissive role for the A2B receptor on the effects of other signals like glutamate, K⁺, NO and NH₄⁺ is a distinctive possibility.

8. Fig. 2g-i. It is surprising that a 54% reduction in A2B adenosine receptor mRNA expression fully cancelled the response to cAMP and PKA evoked by adenosine in cultured astrocytes. Is this explained by a more general change in astrocytic phenotype? Please discuss. Along the same line an experiment showing the effect of Schaffer collateral stimulation on cAMP and PKA in astrocytes from floxed *Adora2b* mice transduced to AAV5-Gfap-iCre-mCherry is required to validate the in vitro data showed in Fig, 2H.

9. Fig 3a. Cytosolic NADH is not a readout of glucose consumption or glycolysis, because glucose is also metabolized to glycogen or via the pentose-phosphate pathway. Crucially, cytosolic NADH may be confounding because is also sensitive to mitochondrial activity. For example, using Peredox, Koehler et al., *GLIA* 2018 showed that glutamate increases NADH (DOI: 10.1002/glia.23504) without affecting glycolysis (ref. 43). In this case cytosolic the NADH rise is provoked by mitochondrial failure.

10. Figs. 3h and 4f. It is shown that astrocytic A2B KO hampers excitatory neurotransmission and LTP. It is also shown that lactate supports both processes, but without knowing the effect of lactate on control slices, it is not possible to tell whether this an actual rescue or alternatively, that lactate works via a parallel pathway. Please provide the control experiment.

Referee #3 (Remarks to the Author):

This study show that activation of neurons led to adenosine release that in turn activate astrocytic adenosine 2B receptors. A2BR activation does in turn increase cAMP resulting in glucose degradation to lactate. Deletion of A2BR in astrocytes prevented synaptic long-term potentiation in the CA3-CA1 pathway of the hippocampus and severely impaired the recognition memory. The analysis highlights a previous unknown pathway of metabolic signaling in brain. This is a novel observation and therefore an interesting and important study. However, several points need to be addressed.

Major critique:

1. Most of the data are collected in acute brain slices that necessarily is exposed to traumatic injury when the slices are cut. Another concern is that the slices are prepared from 3-4 weeks old animals. It is well-known that ectonucleotidase activity is high during development and rapidly activated upon injury. See for example <https://www.ncbi.nlm.nih.gov/pmc/articles/PMC4922325/>. Also, HFS stimulation is not physiological and induces by itself ATP release. See: <https://www.nature.com/articles/nm1693>. Use of organotypic slices does not solve the problem, since these consist of immature cultured cells (harvested at postnatal P1-2) with high ectonucleotidase activity. Observation in the developing brain cannot be transferred to adult neuroglia signaling. See for example: <https://www.ncbi.nlm.nih.gov/pubmed/34245686>. Thus, it is imperative that key experiments, i.e. cAMP measurement in astrocytes are repeated in intact adult mice using physiological stimulation instead of relying on high-frequency electrical in immature hippocampal preparations.
2. How do the authors know that ATP is released from neurons? Same experiments as above will address this question.
3. A very large literature exists on adenosine and astrocytes. Most studies claim that astrocytes control neural activity by release of adenosine. See for example: Halassa et al.. Astrocytic modulation of sleep homeostasis and cognitive consequences of sleep loss. *Neuron*. 2009 61(2):213-9 and Pascual et al, Astrocytic purinergic signaling coordinates synaptic networks. *Science*. 2005

310:113-6. Florian et al., (2011). Astrocyte-derived adenosine and A1 receptor activity contribute to sleep loss-induced deficits in hippocampal synaptic plasticity and memory in mice. *Journal of neuroscience*: 31, 6956-6962. Xu et al. Astrocytes contribute to pain gating in the spinal cord. *Sci Adv.* 2021;7(45):eabi6287. doi: 10.1126/sciadv.abi6287. Ma et al., Neuromodulators signal through astrocytes to alter neural circuit activity and behaviour. *Nature.* 2016 7629:428-432. These are just some of many publications. Since these papers involve neuroglia adenosine signaling it would be expected that the authors discuss why these papers report the opposite signaling path as reported in the submitted report. In support of the authors finding is the recent publication based on adenosine biosensors reporting that adenosine is released by neurons, not astrocytes. See <https://www.biorxiv.org/content/10.1101/2020.05.04.075564v1>. This findings confirms the original finding that adenosine is released by neurons rather than astrocytes: <https://www.ncbi.nlm.nih.gov/pubmed/22421436>. It is important that the authors explain the background of their study and what they can conclude regarding the old literature based on their own observations.

Author Rebuttals to Initial Comments:

Responses to the referees' comments

We would like to thank the Reviewers and the Editors of *NATURE* for their time taken to evaluate our submission, and in general a very positive assessment of our work. We are grateful for the most thorough and constructive comments provided and appreciate an opportunity to resubmit our work. In this letter we describe the results of the additional experiments requested by the Reviewers and provide a full response to all the criticisms raised.

The key additional datasets included in the revised submission:

1. Results of the control and validation experiments showing specificity of genetic approaches used in this study to knockdown A2B receptor expression in astrocytes by immunohistochemistry, analysis of mRNA and protein expression, functional readouts, and metabolome analysis (to address the comments raised by Reviewer 1).
2. Results of the experiments conducted in mice with astrocyte-specific knockdown of A2B adenosine receptors illustrating the importance of this signaling pathway in the regulation of sleep, as one of the fundamental behaviors (to address the comments raised by Reviewer 1).
3. Results of brain metabolome analysis showing that A2B receptor knockdown specifically in astrocytes leads to a dramatic reduction in brain cAMP (in full agreement with the data obtained in reduced preparations) and has a major impact on brain energy metabolism (to address the comments raised by Reviewer 2 and 3). The observed metabolic re-programming caused by deficiency A2B receptor expression in astrocytes supports the critical role of this signaling pathway in regulation of brain energy metabolism.
4. Results of the experiments showing that the effects of K^+ on astrocyte glycolysis and lactate release are dependent on A2B receptor-mediated signaling, providing evidence for a critical permissive role of the identified mechanism, which is required for other signals of neuronal activity to modulate astrocyte metabolism (to address comments raised by Reviewer 2).

Referee #1:

The paper by Gourine and colleagues nicely shows that the metabolic coupling between neurons and astrocytes to maintain adequate energy supply to neurons, especially during low glucose availability and high neuronal firing, is mediated by neuronal adenosine release activating astrocytic A2B receptors. The activation of these receptors triggers intracellular PKA and cAMP cascades that leads to enhanced glucose metabolism and lactate availability. Overall, the paper is well written and the figures for the data are clear. My major concerns focus on novelty for Nature and aspects of rigor and controls.

Response: We thank this Referee for their time taken to review our manuscript and overall positive evaluation of our work. We are grateful for the detailed and constructive comments provided and appreciate the Reviewer's assessment of our study/paper as interesting, well written and clearly illustrated. For this resubmission we performed additional and control

experiments as requested by the Reviewer and revised the manuscript accordingly. Please review our arguments below on the novelty of the reported data as well as our full responses to all the other criticisms raised.

1. This study follows many years of work on the lactate shuttle as a link between neuronal activity and energy supply via astrocytes. The various papers leading to this hypothesis are well cited in the current manuscript and this mechanism is probably one of the most well established mechanisms in the whole astrocyte field. In the current study the authors have used pharmacology and some genetic approaches to show that coupling between neurons and astrocytes occurs via neuronal adenosine release that activates A2B receptors on astrocytes. This is interesting, but identifying the link is incremental and not at the level of advance expected for a Nature paper. On page 8, the authors also do an excellent account of discussing past mechanisms that have been proposed and why they did not work out. Hence, the major advance in the current study is the proposal of another mechanism involving Adenosine. Although this is interesting to specialists, it is not broadly appealing and not highly novel at the level expected of a Nature paper. They have added interesting details to a mechanism that is well known. I recognize this is a judgment call.

Response: Respectfully, we disagree with the Reviewer on this point. The molecular identity of the astrocyte sensor of neuronal activity remains unknown despite three decades of research since the lactate shuttle hypothesis was first proposed (below, we reproduce a summary figure from one of the most recent [March 2023] review articles in the field). We would argue that our study is novel because it identifies the key signaling mechanism that links neuronal activity and astrocyte metabolism and, for the first time, describes a clear functional role of cAMP signaling in astrocytes, demonstrated in a loss-of-function experiment. To the best of our knowledge, this study is the first in the field of brain energy metabolism which provides a comprehensive analysis “from molecule-to-behaviour” of the key signaling pathway of metabolic coupling between neurons and astrocytes in the mammalian brain. In contrast to the majority of studies of previously proposed mechanisms, the role of the identified mechanism is supported by genetic evidence, brain metabolomics, and *in vivo* behavioral readouts. For this revision we performed further experiments and obtained data showing that astrocyte A2B receptor-mediated signaling regulates global brain energy metabolism and is responsible for sleep-promoting effect of brain adenosine. Collectively, our data identify the adenosine A2B receptor as both the astrocyte sensor of neuronal activity (in the figure below labeled as “activity sensor”), which, as our new data suggest, is critically important for brain function, and an attractive, readily druggable target and therapeutic opportunity for brain energy rescue (see: <https://pubmed.ncbi.nlm.nih.gov/32709961/>).

We thank the Reviewer for raising this comment, and in hindsight, we now realize that perhaps the most novel aspects of our work - *the mechanisms and the importance of cAMP signaling in astrocytes* - were not effectively highlighted and may not have been fully appreciated by the reviewers. We revised the title and the text of the paper to better reflect the core findings, importance, and novelty of the study.

[REDACTED]

2. The author's evidence that astrocytes express A2B receptors is based on past RNA-seq data and pharmacological experiments. All of the key experiments for control and following A2B reduction need to be repeated with additional readouts of A2B receptors such as antibody labelling or in situ hybridization. These controls are critical to know that the reductions of A2B receptors did happen as the data suggest, but also to know what level of reduction happened in each experiment that led to the functional readouts. Right now, there is no direct way of knowing how much A2B was reduced. Such experiments should also assess A2B and A2A expression in neurons and oligodendrocytes. Oligos are important as they express A2B receptors and the AAV approaches using the GFAP promoter are expected to target astrocytes and some oligodendrocytes. Thus, there is no direct evidence that A2B has been reduced only in astrocytes – although this is the simplest explanation. These types of control experiments are essential and should be reported in detail with Extended data.

3. The Gfap Cre virus studies use tdTomato controls. However, these controls do not show that the resultant A2B reduction was limited to astrocytes. Thus, a series of experiments are needed to carefully explore the cell specificity of the Cre genetic interventions as being astrocyte specific. Without controls key aspects of the study are not on solid foundations. These types of control experiments are essential and should be reported in detail in Extended data.

Response: We agree with these comments and performed essential control experiments as requested by the Reviewer. In the revised manuscript we now report that:

1. In cultured astrocytes of Adora2b^{flox/flox} mice transduced to express Cre-recombinase, the expression of A2B receptor protein was reduced by >50% (revised Figure 2c).

2. Expression of Cre-recombinase in astrocytes of *Adora2b^{flox/flox}* mice fully blocked the effects of adenosine on intracellular [cAMP] and PKA activity (revised Figure 2d-g), indicating that these effects of adenosine are mediated exclusively by A2B receptors.

3. In hippocampal slices of *Adora2b^{flox/flox}* mice, expression of Cre-recombinase prevented increases in intracellular PKA in astrocytes induced by stimulation of Shaffer collateral fibers (revised Figure 2i), suggesting that the neuronal activity-dependent recruitment of cAMP/PKA signaling in astrocytes is mediated predominantly by A2B receptors.

4. Expression of Cre-recombinase in the hippocampus of *Adora2b^{flox/flox}* mice decreased the tissue A2B receptor expression at mRNA level by 40%, A2B receptor protein expression in astrocytes (isolated by FACS sorting of eGFP-expressing transduced astrocytes) by 54% (Extended Data Figure 8) and reduced the amount of lactate released (measured using the biosensors in acute brain slices) in response to adenosine and AMPA by 47% and 44%, respectively (Revised Figure 4b,c).

5. The viral vector used in this study (*AAV5-Gfap-eGFP-iCre*) was found to be highly specific in driving the expression of Cre-recombinase in hippocampal astrocytes. Extended Data Figure 8 provides images of expression and quantification of eGFP-iCre expression in hippocampal astrocytes, neurons, microglia, and oligodendrocytes. The data demonstrate highly specific targeting of hippocampal astrocytes by AAVs used in this study, similar to the previous reports on studies that used the same construct (PMID: 28479102).

6. Treatment of *Adora2b^{flox/flox}:Aldh1l1^{Cre/ERT2}* with tamoxifen reduces A2B receptor transcript level in the hippocampus by ~60% (revised Figure 3g). Biosensor recording confirmed depletion of A2B receptors in this model, as A2B agonist BAY 60-6583 failed to trigger any significant lactate release in slice preparations of *A2B^{flox/flox}:Aldh1l1^{Cre+}* mice (Extended Data Figure 5f).

7. We have also re-analyzed the transcriptome data and realized that we initially incorrectly classified some of the cells, where *Adora2b* was present as oligodendrocytes, while in fact they were found to be a small population of oligodendrocyte precursor cells (OPCs), which would be also consistent with the classification used in the original publication (PMID: 30096314). OPCs are present in postnatal brain, but their numbers are very low in most parts of the brain, outside of subventricular zone. The presence of some level of *Adora2b* expression in OPCs was also evident from the transcriptome reported by Zhang and colleagues (PMID: 25186741), but in mature oligodendrocytes this receptor is essentially absent. We apologize for this error; the results of the data re-analysis are illustrated by revised Figure 2a,b.

Collectively, the data reported in the revised submission demonstrate that in the brain A2B receptors are expressed almost exclusively by astrocytes and that the methods of genetic manipulation used in our study are highly specific in targeting astrocytes *in vitro* and *in vivo*, resulting in functionally significant knockdown of A2B receptor expression in these cells.

4. I found the finishing comments on caffeine use and brain function in humans to be a stretch. This is highly speculative and not related to the main findings in the study. This type of speculation should be reserved for a review.

Response: We agree with the Reviewer and removed this paragraph from the revised manuscript. Thank you.

5. There is little information on how FRET was assessed in the reporters. From the very brief methods provided it seems they measure CFP and YFP emission after 2p excitation of CFP at 860nm. How did they convert that to FRET? If they measure only the two emissions, they should not call it FRET. Changes in CFP and YFP emission should be shown. In sum, much more information and controls are needed for the optical measurements in order to understand what was done.

Response: We agree with the Reviewer and apologize for not providing this information in our original manuscript. In the revised submission we now illustrate the principles of cAMP and PKA activity detection using Epac-S^{H187} and AKAR4 sensors (Extended Data Figure 1a,d). We thank the reviewer for raising this comment and revised the text of the manuscript and figure labels accordingly to indicate that it was not the FRET *per se*, but the fluorescence intensity ratio (FI ratio) between the two fluorophores that we measured in our experiments. However, the fluorescence intensity changes of these two fluorophores are driven by FRET upon agonist binding to the sensor proteins (Extended Data Figure 1a,d).

6. I do not follow the logic of performing experiments in Fig 1a-c in slices and the pharmacology in Fig 1d-g in cell culture. It would be more rigorous to perform the pharmacology in Fig 1 and all other figures in brain slices because this is closest to the final metrics of LTP and discrimination index.

Response: We agree with the Reviewer but there are clear advantages of cell culture as an experimental model in studies of this type as the cells are studied in isolation and their responses to pharmacological agents are not affected by the potential effects of these agents on other cell types. In this study all the key tests were done both in cell culture and then repeated in brain slices.

7. Additional behaviors should be performed on the A2B astrocyte KO mice to assess other aspects of learning and memory and general brain function. If the mechanism of neuron to astrocyte metabolic coupling via astrocytes is important, then perhaps multiple behavioral domains of the mice are altered. It would be surprising if only learning was changed. Or are the authors saying that there is something special about this mechanism for the hippocampus as implied by the title? If that is the case why is it special for the hippocampus because metabolic coupling is needed throughout the brain?

Response: We agree and thank the Reviewer for making this suggestion. To address this comment of the Reviewer and in order to investigate the effect of specific A2B receptor knockdown in astrocytes on general brain function, we evaluated the effect of this genetic manipulation on sleep as one of the most fundamental behaviors, involving the whole brain. Undertaking a sleep study in the context of this investigation seemed to be most appropriate, considering that progressive elevation in extracellular adenosine in the brain during wakefulness is generally believed to be responsible for accumulation of sleep pressure (PMID: 9157887). Importantly, although it is widely accepted that adenosine promotes sleep via inhibitory A1 receptors expressed by neurons, earlier studies in A1

receptor deficient mice demonstrated normal sleep patterns in these animals (PMID: 14633239). In the revised manuscript we now report that

- astrocyte specific A2B receptor knockdown results in fragmented NREM sleep and wakefulness during the light (resting) phase (Figure 5h; Extended Data Figure 9c);
- astrocyte specific A2B receptor knockdown results in a significant reduction of slow wave activity and low-frequency slow wave activity during the light phase (revised Figure 5i), indicating a clear reduction in sleep pressure.

To disentangle the effects of A2B knockout on memory and sleep (sleep and slow wave activity facilitate memory consolidation PMID: 17086200, PMID: 12495631), we also analyzed sleep patterns in mice with A2B receptor deletion limited to the hippocampal astrocytes. In mice with A2B receptor knockdown in astrocytes of the hippocampus the memory was severely affected (as we reported in our original submission, revised Figure 5a-d), but this genetic manipulation had no effect on sleep architecture or slow wave activity (Extended Data Figure 9a,b).

These data strongly suggest that adenosine A2B receptor-mediated signaling to astrocytes promotes synchronous slow wave activity in NREM sleep and that astrocyte A2B receptors mediate (at least in part) the sleep-promoting effect of adenosine (sleep pressure).

Referee #2:

We thank this Referee for their time taken to review our manuscript. We are grateful for the most thorough and constructive comments provided. For this resubmission we performed additional experiments and data analysis as requested by the Reviewer and revised the manuscript accordingly. Please review our arguments below on the novelty of the reported data as well as our full responses to all the other criticisms raised.

This work reports that the astrocytic A2B receptor and activation by Adenosine and ATP is involved in metabolic activation and in behavior. Whether the involvement in the behavioral changes is mediated by astrocytic metabolism may not be deduced from the present data. Most importantly, this pathway is not new, as claimed by the authors.

Response: Respectfully, we strongly disagree with the Reviewer on this point. The molecular identity of the astrocyte sensor of neuronal activity remains unknown despite three decades of research since the lactate shuttle model was proposed (please review the illustration above reproduced in response to one of the comments raised by Reviewer 1). First, our study describes a novel, potentially unifying mechanism of metabolic communication between neurons and astrocytes and an astroglial sensor of neuronal metabolic demand. Second, for the first time, our study describes a clear functional role and the mechanisms driving cAMP signaling in astrocytes, demonstrated in a loss-of-function experiment, supported by genetic evidence in a mammalian model. And third, for the first time in this field of research our study identifies an attractive, readily druggable target and therapeutic opportunity for brain energy rescue (see: <https://pubmed.ncbi.nlm.nih.gov/32709961/>).

We thank the Reviewer for his/her evaluation, and in hindsight, we now realize that perhaps the most novel aspects of our work - the mechanisms and the importance of cAMP

signaling in astrocytes - were not effectively highlighted and may not have been fully appreciated by the reviewers. We revised the title and the text of the paper to better reflect the core findings, importance, and novelty of the study.

There are technical issues regarding the characterization of metabolism that require solving with control experiments. There are also design issues that weaken the proposed causal chain involving ATP, Adenosine, astrocytic glycolysis, lactate, and behavior. Critically, the role of the A2B receptor on the stimulation of astrocytic metabolism by Schaffer collateral stimulation was not tested.

Response: We thank the Reviewer and in the revised submission describe the results of the additional experiments conducted in response to the comments raised. To understand how A2B receptor mediated signaling impacts brain metabolism we performed untargeted metabolomics of the brain tissue in mice with conditional astrocyte-specific deletion of A2B receptors. In the revised submission we report that cAMP levels were dramatically reduced in brains of A2B^{flox/flox}:Aldh1l1^{Cre+} mice treated with tamoxifen (Figure 3i), indicating that the major pool of brain cAMP is maintained by the activity of A2B receptors in astrocytes. Pathway enrichment analysis demonstrated that depletion of A2B receptors in brain astrocytes results in metabolic re-programming with the most significantly downregulated pathways including the citric acid cycle, ketone body synthesis and the Warburg effect (Figure 3l; Supplementary Table 2). The Warburg effect, defined as an increase in the rate of glucose uptake and preferential production of lactate, is a core feature of astrocyte metabolism (PMID: 31981059). Together, these data and the analysis of brain metabolome strongly suggests that A2B receptor-mediated signaling controls astrocyte glucose metabolism.

In addition, there are significant omissions and misrepresentations of the literature that are misleading regarding novelty and relevance.

Response: We agree that in our original submission the description of the previously suggested pathways of metabolic communication between neurons and astrocytes might have appeared to be somewhat cursory (due to the limitations on the word count) and revised the text of the manuscript accordingly. Please review our detailed responses to the relevant comments in the text below.

Carefully demonstrating and characterizing the effects of Adenosine on glycolytic rates would be interesting and important. There are technical issues, however, see below. Also, it would be key to put this pathway (which is not new per se) in the context of other known pathways. When, where, under what conditions would the adenosine pathway be important? What is the specific role of this pathway?

Response: We thank the Reviewer for asking these important questions. Please see our argument above about the novelty of our study. We believe that the specific role of the described mechanism of adenosine-mediated communication between neurons and astrocytes is clearly illustrated by the reported data: genetic knockdown of A2B receptors specifically in astrocytes, depletes brain cAMP levels, results in a major re-programming of brain energy metabolism, prevents synaptic long-term potentiation in the CA3-CA1 pathway of the hippocampus, severely impairs the recognition memory and disrupts sleep. These data suggest that adenosine/A2B receptor mediated signaling between neurons and astrocytes is essential for normal functioning of brain neural circuits.

In response to the comments raised by Reviewer 1, we performed additional experiments and in this revised submission report data suggesting that adenosine A2B receptor-mediated signaling to astrocytes promotes synchronous slow wave activity in NREM sleep and that astrocyte A2B receptors mediate (at least in part) the sleep-promoting effect of adenosine (sleep pressure).

We further characterized the effect of specific A2B receptor activation on astrocyte glycolysis and the release of lactate in the experiments using pharmacological and genetic blockade of A2B receptors (revised Figure 3). In the revised submission we report data indicating that this signaling pathway is constitutively active. Adenosine-A2B receptor mediated signaling determines the level of intracellular cAMP in astrocytes (revised Figure 1f,h; Figure 3i), the amount of lactate released into the extracellular space (revised Figure 3e,f,h) and coordinates global brain energy metabolism, and in particular key metabolic pathways, including the citric acid cycle, ketone body synthesis and the Warburg effect (revised Figure 1; Supplementary Table 2 and 3)

For this revision we also conducted a series of experiments designed to “put this pathway in the context of other known pathways” in order to answer the key question of the Reviewer. The new data reported in the revised submission show that the effects of K^+ on astrocyte glycolysis and lactate release are dependent on A2B receptor-mediated signaling, providing evidence for a critical permissive role of the identified mechanism, which is required for other signals of neuronal activity to modulate astrocyte metabolism (Extended Data Figure 10).

We also performed further behavioral experiments and now report data showing that astrocyte A2B receptor-mediated signaling is responsible for sleep-promoting effect of brain adenosine (revised Figure 5g-i; Extended Data Figure 9).

Collectively, these data further support the hypothesis that adenosine acting via astrocyte A2B receptors is an important metabolic signaling pathway in the mammalian brain. Please review our detailed responses to all the specific comments in the text below.

Specific comments:

1. Novelty. The authors convey the message that a novel signaling mechanism is being presented, providing a molecular/cellular explanation for the phenomenon of activity-dependent aerobic glycolysis in brain tissue. Whereas the use of tissue slices and behavioral analysis and the specific role of A2B receptors are valuable contributions, authors should be aware that previous work by several laboratories had shown modulation of astrocytic glycolysis by ATP. For example, using Peredox, Koehler et al., *Glia* 2018 demonstrated that ATP increases NADH in astrocytes in culture and in slices, with similar kinetics as found presently (DOI: 10.1002/glia.23504). Horvat et al., *Cell Calcium* 2021, showed a fast increase in astrocytic lactate in response to ATP (doi.org/10.1016/j.ceca.2021.102368.), whereas Juaristi et al., *Glia* 2018, reported accumulation of pyruvate and inhibition of oxygen consumption in astrocytes exposed to ATP (DOI: 10.1002/glia.23574). Also, the impact of Adenosine on metabolism was addressed in older papers that would need to be considered (Bruns RF. *Nucleosides Nucleotides* 10: 931–943, 1991; Magistretti PJ et al. *J Neurosci* 6: 2558–2562, 1986 and then Allaman et al., *Am J Physiol Cell Physiol* 284: C696–C704, 2003).

Response: We thank the Reviewer for raising this comment. We are very familiar with the data reported in all the preceding literature, including studies mentioned by the reviewer. We all undertake research building on the existing knowledge and fully appreciate the contributions made by the authors of these publications, however, none of these studies provided evidence that adenosine is the actual signal and that A2B receptor is the astrocyte sensor of neuronal activity which mediates metabolic communication between neurons and astrocytes. The effects of ATP on astrocytes were reported in more than a hundred published studies (including papers from our laboratory), all focusing on astroglial Ca²⁺ signaling. In this study we show for the first time that the effects of ATP are indirect, mediated by adenosine (formed following ATP breakdown in the extracellular space) acting via A2B receptors to recruit cAMP/PKA signaling pathway (independently of Ca²⁺) and glucose metabolism. Our study is the first to describe a clear functional role of cAMP/PKA signaling in astrocytes, demonstrated in a loss-of-function experiments, and highlighting the impact of the genetic blockade of this signaling pathway on brain energy metabolism and the most fundamental behaviors.

To the best of our knowledge, this study is the first in the field of brain energy metabolism which provides a comprehensive (from molecule-to-behaviour) analysis of a key signaling pathway of metabolic communication between neurons and astrocytes in the mammalian brain. In contrast to the majority of studies focusing on other hypothesized mechanisms, the functional significance of the described mechanism is supported by the genetic evidence, brain metabolomics, and *in vivo* behavioral readouts.

(Specific comments on the data reported in studies mentioned by the reviewer:

1. The study by Koehler et al. (PMID: 30208253) investigated the effect of ATP on NADH/NAD⁺ redox state in cortical astrocytes. There is no mention of adenosine signaling and the conclusions reached by the authors are based on inhibition of NBCe1 activity using the compound S0859 which also inhibits monocarboxylate transporters (PMID: 26027796), and, therefore, may interfere with cellular metabolite uptake/release.

2. The studies by Horvat et al. (PMID: 33621899) and Juaristi et al. (PMID: 30623988) describe the effects of ATP focusing on Ca²⁺ responses. There is no mention of adenosine signaling and ATP-induced activation of astrocyte glycolysis reported is likely due to ATP breakdown to adenosine and the actions of adenosine, as our data suggest.

3. The review article by Bruns RF discusses the role of adenosine in specific conditions like hypoxia, ischemia, and exercise.

4. An earlier (1986) study by Magistretti et al. (PMID: 3018195) is indeed relevant as it shows that adenosine stimulates glycogenolysis in cortical tissue, thus it provides important support for our conclusions. The authors did not show which cells respond to adenosine and did not identify the adenosine receptor responsible for this effect. This idea of adenosine signaling was subsequently disregarded in favor of the mechanism involving glutamate uptake when the original hypothesis of lactate shuttle was put forward in 1994 (PMID: 7938003).

5. The study by Allaman et al. (PMID: 12421692) is somewhat relevant, as it describes long-term effects of adenosine on gene expression and glycogen synthesis in astrocytes. The data reported in that study did not show/suggest that adenosine-mediated signaling can rapidly recruit cAMP/PKA pathway in astrocytes to provide immediate metabolic support of increased neuronal activity. There is also no data on the impact of genetic blockade of this metabolic signaling pathway.)

2. Technical. It is shown that Schaffer collateral stimulation lead to increased cAMP and PKA activity in astrocytes and that the response was insensitive to glutamate receptor

blockage. Thus, the phenomenon is originated in a pre-synaptic signal. As other neuronal signals that induce aerobic glycolysis in astrocytes originate in the post-synaptic compartment (e.g., K⁺ and nitric oxide), the presumably presynaptic origin of the ATP should be discussed. However, further experiments are required to establish this mechanism. Firstly, it is known that electrical stimulation induces local permeabilization of cells, with the ensuing release of ATP. Conceivably, some of this ATP and/or Adenosine may have reached the astrocytes diffusing from the stimulation area. Whereas extracellular ATP is toxic and stimulates astrocytic glycolysis (see below), Adenosine provokes hyperexcitability in the hippocampus Rombo et al., *Hippocampus* 25:566–580, 2015 (A2A). Please provide a demonstration that cAMP and PKA changes require neuronal action potentials, for example using TTX. Along the same line, if the ATP is actually released by neurons (not clear and not substantiated by the data), then the astrocytic response may be sensitive to presynaptic inhibition, e.g., blockers of synaptic vesicle fusion/recycling.

Response: We thank the Reviewer for raising this comment and had performed the requested experiments. In the revised submission we report data showing that TTX fully blocks cAMP and PKA responses in hippocampal astrocytes induced by stimulation of Schaffer collateral fibers (revised Figure 1c; Extended Data Figure 1b,f). These data show that activation of cAMP/PKA pathway in astrocytes requires neuronal action potentials. We did not perform the experiments using blockers of exocytosis as these agents (such as bafilomycin A1 and others) would also interfere with the vesicular release of signaling molecules by astrocytes (see e.g., PMID: 12414798), therefore, the data obtained in the experiments of this type would be difficult to interpret. Presynaptic vesicular release of ATP at central synapses had been well documented in earlier studies (see e.g., PMID: 16639550, also from our laboratory PMID: 18617567) and generally accepted in the field (PMID: 32999463).

3. Design I. It is puzzling that there is no mention of extracellular adenosine and lactate recordings in response to Schaffer collateral stimulation. A positive result in such key experiment, namely a quick release of Adenosine that precedes the release of lactate would support the proposed link between neuronal activity to astrocytic metabolism. Such temporal resolution is well within the capability of modern electrodes. If the role of A2B in behavior is actually mediated by astrocytic glycolysis, the release of lactate should be deficient in the A2B knockout, but not the accumulation of Adenosine.

Response: We agree with the Reviewer but the release of lactate and adenosine in response to Schaffer collateral stimulation had been reported in earlier publications (see e.g. PMID: 23713028), including from our laboratory (PMID: 26661210). The results of the experiment suggested by the reviewer (involving simultaneous recordings of adenosine and lactate release) would be difficult to interpret because of significant differences in the response time between the adenosine (3 enzyme detections system) and lactate (1 enzyme detection) biosensors. Since we need essentially millisecond resolution, the different kinetic/response times of the sensors make it difficult to study with accuracy the temporal relationship between the release of the two analytes. However, to address this comment of the Reviewer we conducted a series of additional experiments and in the revised submission we report that:

1. The basal (tonic) release of lactate (the level of lactate measured using biosensors on the surface of the brain slice at resting conditions) was reduced by 60-70% during pharmacological blockade of A2B receptors (revised Figure 3e,f), and in conditions of

genetic A2B receptor knockdown (revised Figure 3h). We also report a marked reduction in [cAMP] and NADH/NAD⁺ ratio in astrocytes in response to pharmacological blockade of A2B receptors (revised Figure 1f,h; Extended Data Figure 10a).

2. Increased releases of lactate in brain slices induced by adenosine, A2B receptor agonist (Bay 60-6583), AMPA or K⁺ (7 mM) were significantly reduced or abolished by pharmacological blockade (revised Figure 3e; Extended Data Figure 5d,e) or genetic deletion (revised Figure 4b,c; Extended Data Figure 5f and 10d) of A2B adenosine receptors.

3. In hippocampal slices of Adora2b^{flox/flox} mice, expression of Cre-recombinase prevented increases in PKA activity in astrocytes induced by stimulation of Schaffer collateral fibers (revised Figure 2i), suggesting that the neuronal activity-dependent recruitment of cAMP/PKA signaling in astrocytes is mediated predominantly by A2B receptors.

In our opinion these results answer the reviewer's question: the release of lactate (basal and evoked by neuronal activity) is indeed deficient in the A2B knockout (and in conditions of pharmacological A2B receptor blockade). Collectively the data presented in our original submission and the new results show that the activity of the cAMP/PKA pathway in brain astrocytes and the release of lactate induced by neuronal activity are controlled by adenosine-mediated signaling via A2B receptors.

4. Fig. 3e, which shows Adenosine and lactate release in response to post-synaptic activation with AMPA does not support the current story. Rather, the effect of AMPA indicates that a post-synaptic signal is at work, likely K⁺ and/or NO (see below).

Response: Respectfully we disagree with this argument of the Reviewer. We acknowledge that the effect of AMPA is indeed specific to the post-synapse. However, AMPA in this type of experiments acts on the whole slice, including presynaptic neurons. Therefore, activation of AMPA receptors would be expected to induce action potential firing and release of neurotransmitters from the presynaptic terminals as well as trigger local somato-dendritic and axonal release from the AMPA-activated neurons in the area of the recording. Application of AMPA in acute brain slices was used in some of the biosensor experiments as a simple model of enhanced neuronal circuit activity. We now provide a brief justification of the model in the revised text of the manuscript.

5. Design II. Along a similar line, is also unclear why instead of Schaffer collateral stimulation, astrocytic metabolism was probed with ATP and Adenosine. Both ATP and Adenosine are bound not only to affect astrocytes and many ways, including the opening of hemichannels (ref 32), but also neurons, with the secondary release of other mediators that are known to affect metabolism, like glutamate, K⁺, NO, NH₄⁺, etc. As above, demonstration of inhibited astrocytic NADH and glucose responses to Schaffer collateral stimulation in the A2B deficient animal support the proposed signaling pathway. Such type of experiment was instrumental to demonstrate the role of the NBCe1 bicarbonate cotransporter in the activation of astrocytic glucose consumption in response to Schaffer collateral stimulation (Ruminot et al., JCBFM 2017, DOI: 10.1177/0271678X17737012).

Response: We thank the Reviewer for raising this comment. We believe that here the Reviewer is refereeing to the data illustrated by Figure 3a,b of the original submission (revised Figure 3b,d; Extended Data Figure 4a). These results were obtained in the

experiments that follow logically from the data described and illustrated by Figures 1-2 (leading the paper narrative to the conclusion that ATP is ultimately broken down to adenosine which is responsible for neuronal activity dependent activation of cAMP/PKA signaling pathway in astrocytes) and demonstrate that ATP and adenosine stimulate astrocyte glucose consumption via A2B receptor activation (as the effects of ATP and adenosine on NADH/NAD⁺ in individual astrocyte and lactate release in slices were abolished by A2B receptor blockade). In the revised submission we now report that the specific agonist of A2B receptors Bay 60-6583 increases the glycolytic rate in astrocytes and this effect is blocked by astrocyte specific A2B receptor knockout (revised Figure 3a,c). These experiments were conducted in astrocyte culture, where the contribution of neuronal mechanisms would be expected to be minimal. We also report that in acute hippocampal slices the effect of Bay 60-6583 on lactate release is markedly reduced by A2B receptor deletion in astrocytes (Extended Data Figure 5f). Reference 32 of the original submission is the publication from our laboratory, which indeed describes the mechanisms of lactate release via hemichannels but did not study the effects of adenosine or ATP.

We agree with the design of the experiment suggested by the Reviewer, however, Reviewer 3 raised concerns regarding the experimental models involving reduced preparations, including acute brain slices and organotypic slices (please see comment 1 by Reviewer 3). Therefore, to address the comments raised by both reviewers we made a “snapshot” of brain metabolomics in intact adult freely behaving mice in order to determine the effect of astrocyte-specific A2B adenosine receptor deletion on brain energy metabolism. The data obtained in these studies are illustrated by revised Figure 3i-l (please also see Supplementary Tables 1-3). We report that A2B receptor knockdown in astrocytes leads to a dramatic reduction in brain cAMP (in full agreement with the data obtained in reduced preparations), also suggesting that the major pool of brain cAMP is maintained by the activity of A2B receptors in astrocytes. Pathway enrichment analysis demonstrated that depletion of A2B receptors in brain astrocytes results in metabolic re-programming with the most significantly downregulated pathways including the citric acid cycle, ketone body synthesis and the Warburg effect (Figure 3l; Supplementary Table 2 and 3). The Warburg effect, defined as an increase in the rate of glucose uptake and preferential production of lactate, is a core feature of astrocyte metabolism (PMID: 31981059). Together, these data strongly suggest that A2B receptor-mediated signaling controls astrocyte glucose metabolism.

6. Belittling and omission of other signals. Page 8, third paragraph. This paragraph is misleading as it neglects relevant literature and contains statements that are problematic.
- i. It is stated that the response to glutamate is “rather slow”, citing ref. 43. However, ref. 43 shows a rapid stimulation of astrocytic glucose transport, which delivers glucose to glycolysis.
 - ii. It is stated that “astroglial glycolysis is driven by astroglial depolarization, bicarbonate entry and intracellular alkalinization while citing ref. 43. This statement is not correct. The paper showing the astrocytic role of depolarization, bicarbonate entry and alkalinization is actually Ruminot et al. *J. Neurosci.* 2011, doi.org/10.1523/JNEUROSCI.5311-10.2011.
 - iii. The role of K⁺ is played down by a vague mention to “gene sequencing data”, but no information is provided.
 - iv. The role of K⁺ is played down again by citing “in vivo evidence that the neuronal activity-related increases in extracellular [K⁺] are relatively small (ref. 45)”. However, the study of ref. 45 measured cortex-wide K⁺ fluctuations. Much larger changes in local extracellular K⁺, reaching up to 10 mM, have been detected in response to somatic

stimulation (e.g. Heinemann et al Exp Brain Res 79: 283-292, 1990). More recently, activity-dependent local extracellular K⁺ was also estimated at 10 mM using a voltage-sensitive probe in astrocytic PAPs (Ambruster et al., Nat Neurosci 2022, doi.org/10.1038/s41593-022-01049-x). It is well established that interstitial K⁺ reaches levels capable of modulating astrocytic glycolysis.

v. Another key omission is the previous demonstration that extracellular K⁺, acting via the NBCe1 bicarbonate cotransporter, mediates the activation of astrocytic glucose consumption in response to Schaffer collateral stimulation (Ruminot et al., JCBFM 2017, DOI: 10.1177/0271678X17737012).

vi. Additional relevant signals proposed to participate in activity-dependent modulation of astrocytic glycolysis are nitric oxide (San Martin et al., JBC 2017, doi.org/10.1074/jbc.M117.777243) and NH₄⁺ (Lerchundi et al. PNAS 2015, doi.org/10.1073/pnas.150825911).

Response: We thank the Reviewer for this detailed comment and apologize that our description of signaling mechanisms suggested by the preceding studies may not have been clearly written due to the limitations on the word count. We conclude from this comment that the Reviewer considers potassium ions (K⁺) to be the main signal of metabolic communication between neurons and astrocytes. Although one may argue that increases in brain extracellular [K⁺] to 10 mM are only observed in the experiments involving supraphysiological stimulations or in pathological conditions (please see below our specific comments on the data reported in studies mentioned by the reviewer) we acknowledge that signaling by K⁺ may play an important role. For this revision, and in order to put the results of our study in the context of previously reported data, we conducted a series of experiments designed to determine the effect of A2B receptor blockade on K⁺-induced metabolic activation of astrocytes. It was found that stimulation of astrocyte glycolysis induced by 7-15 mM K⁺ is markedly reduced by pharmacological A2B adenosine receptor blockade (Extended Data Figure 10a,c) or by genetic A2B receptor knockout (Extended Data Figure 10b,c). We also found that the release of lactate in acute hippocampal slices triggered by 7 mM K⁺ was strongly reduced by pharmacological or genetic A2B receptor blockade (Extended Data Figure 10d).

These data suggest that either (1) adenosine and A2B receptors mediate the effects of increased extracellular [K⁺] on astrocyte metabolism and/or (2) A2B receptor activity is critically important for the operation of mechanisms activated by K⁺, i.e. the A2B adenosine receptor plays a permissive role, as this Reviewer proposes below. In support of this hypothesis, we now report data indicating that this signaling pathway is constitutively active. Adenosine-A2B receptor mediated signaling determines the basal level of intracellular cAMP in astrocytes (revised Figure 1f,h; Figure 3i) and the level of lactate release into the extracellular space (revised Figure 3e,f,h). We now include the new data in the revised manuscript and have revised the text to acknowledge the importance of previously suggested pathways which contribute to neuronal activity-dependent modulation of astrocyte metabolism.

(Specific comments on the data reported in studies mentioned by the Reviewer:

i: The study by Bittner et al. (PMID: 21430169) reported that stimulation of astrocyte glycolysis in response to glutamate develops after a lag period of several minutes.

ii: We agree and apologize for citing an earlier work by the same group in the original submission. PMID: 21976511 is now cited in the revised manuscript.

iii and **iv**: Respectfully we disagree with the reviewer here. To the best of our knowledge there is no clear evidence to suggest that increases in extracellular $[K^+]$ to 10 mM occur under physiological conditions. Heinemann et al (PMID: 2323375) reported increases in extracellular K^+ to 1.7 mM in cat spinal cord in response to somatosensory stimulation. Similar or smaller in magnitude changes in extracellular $[K^+]$ were recorded during sleep and wakefulness by Ding et al (PMID: 27126038). Ambruster et al. (PMID: 35484406) *predicted* that $[K^+]_e$ can increase to 10 mM, but they did not measure changes in extracellular $[K^+]$ directly. Ambruster et al. acknowledged that "...during prolonged activity (>30 s) or pathological states, like seizures, $[K^+]_e$ can increase to ≈ 10 mM..", implying that changes in K^+ of this magnitude are likely to occur only under specific conditions.

v and **vi**: We agree and apologize for these omissions. The key relevant studies are now cited in the revised manuscript.)

7. Other signals II. The participation of astrocytic A2B receptors in behavior seems convincing. However, it is not clear that these receptors are the main mediators. There are multiple example of signaling pathways that play permissive roles on metabolism. Relevant examples aer the permissive roles of the NBCe1 and the Na^+ pump on the glycolytic responses of astrocytes to K^+ , glutamate and ATP shown by Koehler et al., GLIA 2018 (DOI: 10.1002/glia.23504) and ref. 43. Looking at the present data in the context of the literature, a permissive role for the A2B receptor on the effects of other signals like glutamate, K^+ , NO and NH_4^+ is a distinctive possibility.

Response: We appreciate that the Reviewer finds the reported behavioral data convincing. If we consider a permissive role for the identified mechanism, then this would mean that the A2B receptor-mediated cAMP/PKA signaling is critically important for other mechanisms to exert their effects on astrocyte metabolism. For this revision we performed untargeted brain metabolomics in the brains of mice with conditional astrocyte-specific A2B receptor deletion, focusing on metabolites of the central carbon metabolism. Metabolomic analysis showed that depletion of A2B receptors in brain astrocytes results in major metabolic re-programming with the most significantly downregulated pathways including the citric acid cycle, ketone body synthesis and the Warburg effect (Figure 3I; Supplementary Table 2 and 3). This analysis strongly suggests that A2B receptor-mediated signaling plays a major role in regulation of brain energy metabolism. Additional data reported in the revised submission showing that the effects of K^+ on astrocyte glycolysis and lactate release are largely prevented by pharmacological or genetic blockade of A2B receptors provide further evidence in support of this hypothesis. If the identified mechanism is indeed permissive, then in our opinion the results of this study would be even more important for our understanding of brain energy metabolism and the translational potential of this research. First, permissive role is (by definition) crucial and redundant (permissive) mechanisms may not exist (as our metabolomics data suggest). Second, a critical permissive metabolic pathway, which is relatively straightforward to modulate pharmacologically, may eventually become a key target for brain energy rescue for the ageing and diseased brain (<https://pubmed.ncbi.nlm.nih.gov/32709961/>). We thank the Reviewer for raising this comment and now briefly discuss this issue in the revised manuscript.

8. Fig. 2g-i. It is surprising that a 54% reduction in A2B adenosine receptor mRNA expression fully cancelled the response to cAMP and PKA evoked by Adenosine in cultured astrocytes. Is this explained by a more general change in astrocytic phenotype? Please discuss. Along the same line an experiment showing the effect of Schaffer collateral stimulation on cAMP and PKA in astrocytes from floxed Adora2b mice transduced to AAV5-Gfp-iCre-mCherry is required to validate the in vitro data showed in Fig, 2H.

Response: mRNA/protein quantifications were performed on brain tissue or entire tissue culture samples; these samples included transduced and non-transduced astrocytes and other cell types, therefore, the effects on A2B expression were found to be partial. cAMP/PKA signals were recorded from individual A2BR^{flox/flox} astrocytes, which were confirmed to express Cre recombinase (by observing mCherry expression), therefore complete blockade of the adenosine effects is not surprising (please note that stimulation of adenylate cyclase in response to forskolin was not affected in conditions of A2B receptor deletion). Following Reviewer's suggestion, we conducted the requested experiments for the revised submission. Revised Figure 2i provides summary data showing that in hippocampal slices of A2B^{flox/flox} mice transduced to express Cre recombinase (vector AAV5-Gfap-iCre-mCherry), stimulation of Schaffer collateral fibers fails to trigger PKA responses in astrocytes.

9. Fig 3a. Cytosolic NADH is not a readout of glucose consumption or glycolysis, because glucose is also metabolized to glycogen or via the pentose-phosphate pathway. Crucially, cytosolic NADH may be confounding because is also sensitive to mitochondrial activity. For example, using Peredox, Koehler et al., GLIA 2018 showed that glutamate increases NADH (DOI: 10.1002/glia.23504) without affecting glycolysis (ref. 43). In this case cytosolic the NADH rise is provoked by mitochondrial failure.

Response: We agree with the Reviewer and acknowledge that cytosolic NADH is not a definitive readout of glucose consumption or glycolysis. However, in our manuscript we also report data obtained in the experiments with direct measurements of glucose consumption clearly showing increased astrocyte glycolytic rate in response to adenosine, mediated by A2B receptors. The data obtained using these two complimentary methods as well as the results of brain metabolome analysis are in full agreement and suggest that A2B receptor stimulation enhances astrocyte glucose metabolism.

10. Figs. 3h and 4f. It is shown that astrocytic A2B KO hampers excitatory neurotransmission and LTP. It is also shown that lactate supports both processes, but without knowing the effect of lactate on control slices, it is not possible to tell whether this an actual rescue or alternatively, that lactate works via a parallel pathway. Please provide the control experiment.

Response: We agree with the Reviewer and conducted the suggested experiment, with the result reproduced below (HFS, high frequency stimulation of SC fibers).

We found that supplemental lactate (5 mM) tends to potentiate the LTP transiently, but in our experiments the differences between the groups did not reach statistical significance. Our hypothesis is that when A2B receptor-mediated signaling is blocked (pharmacologically

or genetically) the metabolic support of synaptic activity is impaired. In order to mimic these conditions, we also conducted experiments involving removal of extracellular glucose. It was found that supplemental lactate effectively supports the excitatory transmission in the absence of glucose (Extended Data Figure 7b). This effect of supplemental lactate was similar to the effect of lactate in supporting the synaptic activity and plasticity in conditions when A2B signaling was blocked experimentally. Please also review the new data included in the revised submission showing that pharmacological A2B receptor blockade or astrocyte-specific A2B receptor deletion markedly reduced the basal and neuronal activity-evoked release of lactate recorded in brain slices (revised Figure 3e,f,h). Collectively, these data suggest that genetic A2B receptor knockdown in astrocytes or pharmacological blockade of A2B receptors reduce the amount of lactate in the extracellular space and expose energetic vulnerability of synaptic transmission in the hippocampus, which can be supported/rescued by supplemental lactate.

Referee #3:

This study show that activation of neurons led to adenosine release that in turn activate astrocytic adenosine 2B receptors. A2BR activation does in turn increase cAMP resulting in glucose degradation to lactate. Deletion of A2BR in astrocytes prevented synaptic long-term potentiation in the CA3-CA1 pathway of the hippocampus and severely impaired the recognition memory. The analysis highlights a previous unknown pathway of metabolic signaling in brain. This is a novel observation and therefore an interesting and important study. However, several points need to be addressed.

Response: We thank this Reviewer for their time taken to evaluate our submission and overall positive assessment of our work. We are very grateful for the detailed and constructive comments provided and appreciate the Reviewer's assessment of our data/study as novel, interesting and important. For this resubmission we performed additional experiments as requested and appreciate an opportunity to submit our revised manuscript for this Reviewer's perusal.

1. Most of the data are collected in acute brain slices that necessarily is exposed to traumatic injury when the slices are cut. Another concern is that the slices are prepared from 3-4 weeks old animals. It is well-known that ectonucleotidase activity is high during development and rapidly activated upon injury. See for example <https://www.ncbi.nlm.nih.gov/pmc/articles/PMC4922325/>. Also, HFS stimulation is not physiological and induces by itself ATP release. See: <https://www.nature.com/articles/nm1693>. Use of organotypic slices does not solve the problem, since these consist of immature cultured cells (harvested at postnatal P1-2) with high ectonucleotidase activity. Observation in the developing brain cannot be transferred to adult neuroglia signaling. See for example: <https://www.ncbi.nlm.nih.gov/pubmed/34245686>. Thus, it is imperative that key experiments, i.e. cAMP measurement in astrocytes are repeated in intact adult mice using physiological stimulation instead of relying on high-frequency electrical in immature hippocampal preparations.

Response: We completely agree and for this revision conducted a number of additional experiments to address this comment of the Reviewer. That increased neuronal activity is associated with cAMP elevations in neighboring astrocytes had been shown previously by works of Dr Hirase and colleagues (PMID: 31980655), however the underlying mechanisms and the importance of these responses remain unknown. The major contribution of our

study is the discovery of the signaling mechanism responsible of the recruitment of cAMP-mediated pathway in astrocytes and demonstration of its functional significance in coordination of brain energy metabolism, neuronal plasticity, and behaviour. In the revised submission we now report data confirming the observations of Dr Hirase that increased neuronal activity is indeed associated with cAMP signals in astrocytes *in vivo* (Extended Data Figure 1c).

We felt, however, that these data would not be sufficient to answer this comment of the Reviewer and additionally made a “snapshot” of brain metabolomics in intact adult freely behaving mice in order to determine the effect of astrocyte-specific A2B adenosine receptor deletion on brain energy metabolism. The data obtained in these studies are illustrated by revised Figure 3i-l (please also see Supplementary Table 1-3). We report that A2B receptor knockdown in astrocytes leads to a dramatic reduction in brain cAMP (in full agreement with the data obtained in reduced preparations), also suggesting that the major pool of brain cAMP is maintained by the activity of A2B receptors in astrocytes. Pathway enrichment analysis demonstrated that depletion of A2B receptors in brain astrocytes results in metabolic re-programming with the most significantly downregulated pathways including the citric acid cycle, ketone body synthesis and the Warburg effect (Figure 3l; Supplementary Table 2 and 3). Together, these data suggest that the identified adenosine/A2B receptor-mediated mechanism of metabolic communication between neurons and astrocytes plays a major role in coordination of brain energy metabolism.

To address comments raised by Reviewer 1 we also studied the effect of astrocyte-specific deletion of A2B receptor on sleep (as one of the fundamental behaviors) and in the revised submission report that astrocyte specific A2B receptor knockdown results in a marked reduction in slow wave activity and fragmentation of wake and sleep (NREM stage) architecture during the light (resting) phase. These data strongly suggest that adenosine A2B receptor-mediated signaling to astrocytes promotes synchronous slow wave activity in NREM sleep and that astrocyte A2B receptors mediate (at least in part) the sleep-promoting effect of adenosine (sleep pressure) (revised Figure 5g-i; Extended Data Figure 9).

To the best of our knowledge, this study is the first in the field of brain energy metabolism that provides a comprehensive analysis “from molecule-to-behaviour” of the key signaling pathway of metabolic communication between neurons and astrocytes in the mammalian brain. In contrast to studies of the previously suggested pathways, the importance of the mechanism identified in our study is supported by genetic evidence, metabolomics, and *in vivo* behavioral readouts.

2. How do the authors know that ATP is released from neurons? Same experiments as above will address this question.

Response: We thank the Reviewer for this comment. Presynaptic release of ATP at central synapses had been documented in earlier studies (see e.g., PMID: 16639550, also from our laboratory PMID: 18617567) and generally accepted in the field (PMID: 32999463). Although the cellular source(s) of ATP/adenosine were not directly investigated in the current study, the data showing lack of an effect of glutamate receptor blockade on the neuronal activity-induced cAMP/PKA signals in astrocytes point to pre-synaptic origin of the release. Astroglial mechanisms of ATP/adenosine release had been shown to supplement the extracellular pool of adenosine as suggested by the results of studies

reported by Prof Dale and colleagues (PMID: 23713028), but the results of our experiments involving glutamate receptor blockade point to the key role of presynaptic release.

3. A very large literature exists on Adenosine and astrocytes. Most studies claim that astrocytes control neural activity by release of Adenosine. See for example: Halassa et al.. Astrocytic modulation of sleep homeostasis and cognitive consequences of sleep loss. *Neuron*. 2009 61(2):213-9 and Pascual et al, Astrocytic purinergic signaling coordinates synaptic networks. *Science*. 2005 310:113-6. Florian et al., (2011). Astrocyte-derived Adenosine and A1 receptor activity contribute to sleep loss-induced deficits in hippocampal synaptic plasticity and memory in mice. *Journal of neuroscience*: 31, 6956-6962. Xu et al. Astrocytes contribute to pain gating in the spinal cord. *Sci Adv*. 2021;7(45):eabi6287. doi: 10.1126/sciadv.abi6287. Ma et al., Neuromodulators signal through astrocytes to alter neural circuit activity and behaviour. *Nature*. 2016 7629:428-432. These are just some of many publications. Since these papers involve neuroglia Adenosine signaling it would be expected that the authors discuss why these papers report the opposite signaling path as reported in the submitted report. In support of the authors finding is the recent publication based on adenosine biosensors reporting that Adenosine is released by neurons, not astrocytes. See <https://www.biorxiv.org/content/10.1101/2020.05.04.075564v1>. This findings confirms the original finding that Adenosine is released by neurons rather than astrocytes: <https://www.ncbi.nlm.nih.gov/pubmed/22421436>. It is important that the authors explain the background of their study and what they can conclude regarding the old literature based on their own observations.

Response: We thank the Reviewer for raising this important comment. We are very familiar with this literature and were indeed surprised when our experiments revealed that adenosine has such a strong effect in stimulating astrocyte glucose metabolism. The majority of published studies (including the publications mentioned by the reviewer) focused on the effects of adenosine on brain neurons, mediated via inhibitory A1 receptors, and none of these studies explored the metabolic effects of adenosine, in particular on astrocytes. For example, there is significant evidence that progressive elevation in extracellular adenosine in the brain during wakefulness is responsible for accumulation of sleep pressure (PMID: 9157887). This view is supported by the data reported in the publication by Halassa and colleagues mentioned by the reviewer (PMID: 19186164). However, studies in A1 receptor deficient mice demonstrated normal sleep patterns in these animals (PMID: 14633239). To address comments raised by Reviewer 1 we investigated the effect of astrocyte-specific deletion of A2B receptors on sleep. The data illustrated by revised Figure 5g-i and Extended Data Figure 9 show that genetic blockade of this signaling mechanism reduces the accumulation of sleep pressure, pointing to the importance of A2B receptors expressed by astrocytes in mediating sleep-promoting effects of adenosine in the brain. To disentangle the effects of A2B knockout on memory and sleep we performed the experiments in mice with A2B receptor knockdown in astrocytes of the whole brain, and in a separate group of animals with A2B receptor deletion limited to hippocampal astrocytes. In the first group of animals, both sleep and memory were impaired (revised Figure 5e-i; Extended Data Figure 9c). In the other group, memory was severely affected (revised Figure 5a-d) but deletion of A2B receptor in hippocampal astrocytes had no effect on sleep architecture or sleep pressure (Extended Data Figure 9a). We thank the reviewer for raising the comment and in the revised manuscript we cite the key preceding literature in view of the data obtained in our study.

Reviewer Reports on the First Revision:

Referees' comments:

Referee #1 (Remarks to the Author):

The revised paper by Gourine and colleagues is improved and my previous concerns have mostly been addressed. Notably, the writing is much clearer and they have now performed essential controls that were almost completely lacking in the previous submission. The addition of these controls has not changed their major conclusions, which remain almost exactly the same. However, with the benefit of the controls they are now on stronger ground. The paper is written more clearly to emphasize that they have found that A2b receptors on astrocytes are the sensor for neuron-astrocyte metabolic coupling. This is the main novel aspect of the work.

I have suggestions for improving the manuscript a little more.

1. In Fig 1b, can the authors also show traces for "+TTX" condition? It will be useful for the reader to see if this condition is like "no stim" or if TTX changes the shape of the cAMP traces. I realize the average data are shown in panel C, but I think the paper would be stronger if the traces were shown as well.
2. In Fig 4G, why do the LTP data with "A2B Astro KD + lactate", i.e. the blue line trend down over 20-80 min? If the function of A2B is provide lactate from astrocytes, why does circumventing this pathway with bath application of lactate not fully look like "control"? Some discussion of this point is needed somewhere.
3. A general comment is that I thought the paper was a little wordy and could be reduced in length to focus on the identification of the A2B receptor as the link between neuronal activity and astrocytic lactate supply. Much of the work in cell culture could be placed in extended information since the physiological relevance of astrocyte cultures is questionable, although I recognize the authors disagreed with my previous comment on this.
4. In Fig 1a and all other figures where electrical stimulation is used, please specify the electrical stimulation conditions in the legend and indicate on the figure the total duration of electrical stimulation as "stim on" and "stim off" if it is of several seconds duration.
5. This last comment relates to points raised by Reviewers 2 and 3 on whether ATP is released presynaptically. I suggest the authors tone down their assertions that ATP is released presynaptically based on the papers they cite. A broader assessment of the literature supports the idea that ATP is released from presynaptic terminals but the evidence for this is not particularly strong and the ATP released is not very much. Perhaps it is released presynaptically and from astrocytes in response to a presynaptic stimulation?
6. The word "the" is used incorrectly in many places, but this could be corrected easily.

Overall, congratulations to the authors on some nice experiments.

Referee #2 (Remarks to the Author):

The authors have added many experiments and present many more data in their revised version. Whereas this was a major effort, my main criticism remains: The work introduces an interesting – yet not at all novel – signaling pathway in the complex framework of neural metabolism. In a fundamentally complex system, it is dangerous and conceptually wrong to claim that there is a principal signal that is the main driver. Experiments that address the relative strengths of different pathways are extremely difficult to design. The claim that the astrocytic A2B receptor is THE astrocyte sensor which mediates metabolic coupling between neurons astrocyte is not substantiated. As an example, potassium-induced metabolic coupling in astrocytes was reduced by blocking A2B receptors or by genetic A2B receptor knockout. This is conceptually not evidence for either a primary role of K⁺ nor adenosine. What would happen to the adenosine induced upregulation in case of a potassium signaling block? The paper is thus still much overstating the role of the adenosine/A2B receptor signaling cascade, which – again – is not a novel concept. Even in the abstract the author go as far as saying “These data identify the adenosine A3B receptor as the astrocytic sensor of neuronal activity ..”.

Interestingly, in the point-to-point reply, the authors themselves state that “the most novel aspects of our work – the mechanisms and the importance of cAMP signaling in astrocytes – were not effectively highlighted and may not have been fully appreciated by the reviewers.” However, this is not reflected in the abstract/introduction/conclusions, in which the novelty is claimed for the entire pathway.

With the data in this work, the authors have some legitimate new data for a known pathway and how adenosine and the A2B receptor are involved in astrocytic sensing of neural activity. This is important, yet incremental advance that should be published, but in my view clearly not in Nature. I apologize for the bluntness of this judgement.

Most of the newly presented data corroborate the involvement of adenosine/A2B receptor signaling. However, it is increasingly worrying that the authors jump from cultured astrocytes to acute slices to in vivo experiments, and it is extremely hard to understand where the data are taken or why the respective model was chosen in the first place.

Dynamics of the adenosine signals are not in line with the hypothesis. It could be learnt from reviewer 3 and the suggested literature, that neuronal adenosine release dynamics are relatively slow (around 40 seconds, Wu et al, PNAS, 2023). In the hippocampal CA1 region, it is released from the postsynaptic membrane via a vesicle independent, ENT-dependent mechanism. This significantly challenges adenosine to be the claimed major activity signal for astrocytes. The sluggishness of adenosine is consistent with the standard view that it is not an immediate signal but a slow messenger of metabolic states. Furthermore, ENT-mediated transport requires an intracellular build-up of adenosine, secondary to ATP depletion. However, neuronal ATP depletion is not observed, except at high frequency stimulation (<https://doi.org/10.1016/j.cell.2013.12.042> and

DOI:10.1016/j.cmet.2018.11.005). Recent in vivo imaging showed that neuronal ATP levels in fact increase at the onset of the wake state upon transition from quiet to active awake, as well as following micro-awakening during NREM sleep state, and in all cases was preceded by CBF increase (<https://doi.org/10.1038/s42003-020-01215-6>). The same study showed neuronal ATP decrease during REM sleep, in which adenosine is thought to be released. All this shows the complexity of the involve signaling and puts in question the primary role of adenosine as a straightforward and fast activity signal to astrocytes.

Referee #3 (Remarks to the Author):

The authors have addressed all the points I raised and I have no more critique. Nice study.

Author Rebuttals to First Revision:

Responses to the referees' comments

We would like to thank the Reviewers and the Editors of *NATURE* for their time taken to evaluate our revised submission. We are grateful for the additional feedback provided, which has contributed to the refinement of our manuscript, and greatly appreciate the opportunity to resubmit our work. We now provide a full response to all remaining points raised by the Referees and submit the second revision of the manuscript.

Referee #1:

The revised paper by Gourine and colleagues is improved and my previous concerns have mostly been addressed. Notably, the writing is much clearer and they have now performed essential controls that were almost completely lacking in the previous submission. The addition of these controls has not changed their major conclusions, which remain almost exactly the same. However, with the benefit of the controls they are now on stronger ground. The paper is written more clearly to emphasize that they have found that A2b receptors on astrocytes are the sensor for neuron-astrocyte metabolic coupling. This is the main novel aspect of the work. I have suggestions for improving the manuscript a little more.

Response: We thank this Referee for their time taken to review our revised manuscript and their very positive assessment of our work. We are grateful for the detailed and constructive comments provided on our initial submission, as well as the additional suggestions offered to further improve our manuscript. Please review our responses to all the additional comments raised. We found these comments very helpful and revised the text and the figures of the manuscript accordingly. Thank you.

1. In Fig 1b, can the authors also show traces for "+TTX" condition? It will be useful for the reader to see if this condition is like "no stim" or if TTX changes the shape of the cAMP traces. I realize the average data are shown in panel C, but I think the paper would be stronger if the traces were shown as well.

Response: We agree and have revised the manuscript Figures accordingly. Figure 1b and Extended Data Figure 1e of the revised manuscript now illustrate the neuronal activity-induced cAMP and PKA responses in hippocampal astrocytes in the absence and presence of TTX or under conditions of glutamate receptor blockade.

2. In Fig 4G, why do the LTP data with "A2B Astro KD + lactate", i.e. the blue line trend down over 20-80 min? If the function of A2B is provide lactate from astrocytes, why does circumventing this pathway with bath application of lactate not fully look like "control"? Some discussion of this point is needed somewhere.

Response: We thank the Reviewer for raising this comment. When we obtained these data, we initially thought that the amount of supplemental lactate provided in the bath solution (5 mM) might not be sufficient to fully compensate for impaired endogenous supply of this metabolic substrate (particularly in conditions where cAMP signaling in astrocytes was impaired by genetic deletion of the A2B receptor). However, further

experiments conducted using the same acute hippocampal slice preparations (fEPSP recordings in the CA1 area) showed that 5 mM supplemental lactate can effectively support excitatory neurotransmission under conditions of glucose deprivation (0 mM glucose in the media) (see Extended Data 7b). The observation that stimulation of other G_s coupled receptors (β -adrenoceptors with isoproterenol) fully rescued the LTP induction in slice preparations of mice with A2B receptor knockdown in hippocampal astrocytes suggests that while lactate metabolic signaling is involved in LTP initiation, other cAMP-dependent astroglial mechanisms are essential for LTP maintenance. In the revised manuscript, we include a brief discussion of this point as suggested by the Reviewer.

3. A general comment is that I thought the paper was a little wordy and could be reduced in length to focus on the identification of the A2B receptor as the link between neuronal activity and astrocytic lactate supply. Much of the work in cell culture could be placed in extended information since the physiological relevance of astrocyte cultures is questionable, although I recognize the authors disagreed with my previous comment on this.

Response: We thank the Reviewer for raising this comment. We have now revised the text of the manuscript to 3,365 words (the summary paragraph and body text). As suggested by the Reviewer, we have also revised and further simplified the main Figure 1, and now illustrate most of the results obtained in cell culture in the Extended Data file.

4. In Fig 1a and all other figures where electrical stimulation is used, please specify the electrical stimulation conditions in the legend and indicate on the figure the total duration of electrical stimulation as "stim on" and "stim off" if it is of several seconds duration.

Response: Thank you for this suggestion. This information is now provided within the revised figure legends, and all the Figures have been updated accordingly.

5. This last comment relates to points raised by Reviewers 2 and 3 on whether ATP is released presynaptically. I suggest the authors tone down their assertions that ATP is released presynaptically based on the papers they cite. A broader assessment of the literature supports the idea that ATP is released from presynaptic terminals but the evidence for this is not particularly strong and the ATP released is not very much. Perhaps it is released presynaptically and from astrocytes in response to a presynaptic stimulation?

Response: We agree with the reviewer. Indeed, there is evidence pointing to two sources of activity-dependent adenosine release involving transporter-mediated and astrocytic ATP exocytosis (PMID: 23713028; PMID: 36996110). Additionally, there is strong evidence for synaptic ATP release (PMID: 16639550). Together the data reported in a number of publications point to several potential sources of release which contribute to the robust immediate increases in local concentration of adenosine in response to increases in neuronal activity. To address this comment of the reviewer we now revised the text of the manuscript and summary illustration (Figure 5j) accordingly.

6. The word "the" is used incorrectly in many places, but this could be corrected easily.

Response: Thank you. We carefully proofread the manuscript several times and revised the text accordingly.

Overall, congratulations to the authors on some nice experiments.

We sincerely thank the Reviewer for dedicating their time to evaluating our revised submission and for their highly positive assessment of our work.

Referee #2

The authors have added many experiments and present many more data in their revised version. Whereas this was a major effort, my main criticism remains: The work introduces an interesting – yet not at all novel – signaling pathway in the complex framework of neural metabolism.

Response: We thank the Reviewer for their evaluation of our revised manuscript. We strongly disagree with the Reviewer and have provided detailed responses supported by new experimental data to all the comments raised by this Referee in the first review round. Collectively, the data we report show that adenosine mediates metabolic communication between neurons and astrocytes, and that the A2B receptor serves as an astrocyte sensor of neuronal activity, which is responsible for the recruitment of astrocyte glucose metabolism via activation of the cAMP/PKA signaling pathway. If the Referee strongly believes that the signaling pathway we describe is not novel, then we kindly request this Reviewer to draw our attention to a publication (or a series of publications) reporting that (i) neuronal activity-dependent stimulation of cAMP/PKA pathway in astrocytes is mediated by adenosine acting at A2B receptors, leading to the recruitment of astrocyte glucose metabolism, and (ii) genetic blockade of this signaling pathway alters brain energy metabolism, impairs synaptic function and disrupts behavior. We do not think such evidence exists.

In a fundamentally complex system, it is dangerous and conceptually wrong to claim that there is a principal signal that is the main driver.

Response: We fully acknowledge the complexity of metabolic pathways and the mechanisms underlying the regulation of metabolism. However, we can counter the Reviewer's comment by providing an example of a well-characterized fundamental physiological mechanism/function mediated by one principal signal. Please consider the regulation of hepatic glucose metabolism and circulating glucose levels by the hormones insulin and glucagon. In this fundamentally complex system, insulin serves as the principal signal and the main driver responsible for the regulation of glucose uptake by the liver (and other tissues). There are other examples of basic physiological functions mediated by one and only one principal molecule and one signaling pathway (PMID: 1378650). Yet, we are not claiming that adenosine is the ONLY signal; in our paper, we report data showing that the impaired function (LTP and memory) caused by experimental blockade of the identified pathway could be fully rescued by the stimulation of other astroglial G_s coupled receptors (β -adrenoceptors).

Experiments that address the relative strengths of different pathways are extremely difficult to design. The claim that the astrocytic A2B receptor is THE astrocyte sensor which mediates metabolic coupling between neurons astrocyte is not substantiated. As an example, potassium-induced metabolic coupling in astrocytes was reduced by blocking A2B receptors or by genetic A2B receptor knockout. This is conceptually not evidence for either

a primary role of K⁺ nor adenosine. What would happen to the adenosine induced upregulation in case of a potassium signaling block?

Response: Respectfully, we disagree with the Reviewer here. Experiments designed to study the interdependence and interactions between different signaling mechanisms are relatively straightforward. In response to the comments raised by this Reviewer in the first review round, we conducted a series of experiments to explore how signaling mediated by K⁺ may interact with the pathway described in our study. The data obtained in these experiments are illustrated by Extended Figure 10. It was found that:

1. The effect of increased extracellular [K⁺] (from 3 mM to 15 mM) on cytosolic NADH-NAD⁺ redox state (as a measure of glucose consumption) in astrocytes was prevented under conditions of pharmacological blockade of adenosine A2 receptors (ZM241385, 10 μM) or genetic deletion of A2B receptors.

2. The effect of increased extracellular [K⁺] (from 3 mM to 10 mM) on the release of lactate, recorded using lactate biosensors in acute hippocampal slices, was greatly reduced in the presence of the A2 receptor antagonist ZM241385.

To address this point of the Reviewer further, we now conducted experiments involving measurements of adenosine release in primary cortical cultures using an adenosine GRAB sensor (PMID: 36996110). Results illustrated by the Figure below show that increasing the extracellular concentration of K⁺ (from 3 mM to 10 mM) triggers release of adenosine. The data are consistent with the results of published studies that reported K⁺-induced release of both ATP and adenosine in hippocampal slices (PMID: 22394324). The most logical conclusion from these results is that adenosine release and A2B receptor activation mediate the effects of increased extracellular [K⁺] on astrocyte glucose metabolism. We are confident that this Reviewer would reach the same conclusion(s) if they were to undertake experiments of this type in their laboratory.

For imaging, cortical cultures were placed into a perfusion chamber mounted on the stage of an upright Leica SP5 microscope with a x40 water immersion lens. The GRAB_{ADO} sensor was expressed using an adenoviral vector. Imaging focused on the cellular membranes facing the media. A 488 nm argon line was used for excitation, and emission was sampled within the range of 495-540 nm.

The paper is thus still much overstating the role of the adenosine/A2B receptor signaling cascade, which – again – is not a novel concept. Even in the abstract the author go as far as saying “These data identify the adenosine A3B receptor as the astrocytic sensor of neuronal activity ..”.

Response: Thank you. We have revised the text of the manuscript to address this comment of the Reviewer; where relevant, we have toned down categorical statements and adopted a more conservative approach in making conclusions from the results of our experiments. Regarding the novelty of the reporting data, we can only repeat our argument provided above: if the Referee strongly believes that the signaling pathway we describe is not novel, then we kindly request this Reviewer to point our attention to a publication (or a series of publications) reporting that (i) neuronal activity-dependent stimulation of cAMP/PKA pathway in astrocytes is mediated by adenosine acting at A2B receptors, leading to the recruitment of astrocyte glucose metabolism, and (ii) genetic blockade of this signaling pathway alters brain energy metabolism, impairs synaptic function, and disrupts behavior.

Interestingly, in the point-to-point reply, the authors themselves state that “the most novel aspects of our work – the mechanisms and the importance of cAMP signaling in astrocytes – were not effectively highlighted and may not have been fully appreciated by the reviewers.” However, this is not reflected in the abstract/introduction/conclusions, in which the novelty is claimed for the entire pathway.

Response: Respectfully, we disagree with the Reviewer here. The manuscript was thoroughly revised to highlight the mechanisms underlying activity-dependent recruitment and the importance of cAMP signaling in astrocytes:

Title: “Brain energy metabolism and core behaviors coordinated by adenosine-mediated *cAMP* signaling in astrocytes”

Abstract: “Stimulation of A2B receptors recruits the canonical cyclic *adenosine 3',5'-monophosphate/protein kinase A signaling pathway*... [the data] show that *cAMP signaling in astrocytes* tunes brain energy metabolism to support fundamental behaviors such as sleep and memory”.

Introduction: “In peripheral tissues such as the liver and muscle, increased energy expenditure rapidly recruits intracellular glycogen stores via activation of the *canonical cyclic adenosine 3',5'-monophosphate (cAMP)/protein kinase A (PKA)-mediated signaling pathway*. Here we show that in the brain the same *cAMP/PKA signaling pathway* is responsible for neuronal-activity dependent control of astrocyte glucose metabolism and ensures unabated metabolic support of synaptic function essential for sleep and memory”.

Conclusions: “We show that neuronal metabolic needs are communicated to astrocytes by adenosine, which is responsible for metabolic activation of these glial cells via *cAMP/PKA-mediated signaling pathway*, - the same mechanism that controls glucose metabolism in muscle and the liver”.

With the data in this work, the authors have some legitimate new data for a known pathway and how adenosine and the A2B receptor are involved in astrocytic sensing of neural activity. This is important, yet incremental advance that should be published, but in my view clearly not in Nature. I apologize for the bluntness of this judgement.

Response: Progress in science is achieved through the competition between ideas. To the best of our knowledge, our study is the first in the field of brain energy metabolism to provide a comprehensive analysis ‘from molecule-to-behavior’ of a signaling pathway of metabolic communication between neurons and astrocytes in the mammalian brain. In contrast to all descriptions of previously suggested pathways, the functional significance of the mechanism identified in our study is supported by genetic evidence. In our opinion, this study contributes significantly to our understanding of the mechanisms underlying metabolic support of brain computation and, therefore, will be of great interest to the wide audience of the Journal.

Most of the newly presented data corroborate the involvement of adenosine/A2B receptor signaling. However, it is increasingly worrying that the authors jump from cultured astrocytes to acute slices to *in vivo* experiments, and it is extremely hard to understand where the data are taken or why the respective model was chosen in the first place.

Response: In our opinion, the comprehensive experimental analysis of the identified signaling pathway using a combination of *in vitro* and *in vivo* models is a major strength of our study. There are clear advantages of cell culture as an experimental model in studies of this type, as the cells are studied in isolation, and their responses to pharmacological agents/genetic manipulations are not affected by the potential effects of these treatments on other cell types. Electrophysiological recordings in acute slices are required to study excitatory synaptic transmission and synaptic plasticity. Ultimately, the functional significance of the mechanism under investigation can only be assessed in studies using *in vivo* animal models. We have carefully reviewed and revised the text of the manuscript to ensure that essential information on the experimental models used is illustrated by the figures and/or provided in the figure legends where appropriate.

Dynamics of the adenosine signals are not in line with the hypothesis. It could be learnt from reviewer 3 and the suggested literature, that neuronal adenosine release dynamics are relatively slow (around 40 seconds, Wu et al, PNAS, 2023). In the hippocampal CA1 region, it is released from the postsynaptic membrane via a vesicle independent, ENT-dependent mechanism. This significantly challenges adenosine to be the claimed major activity signal for astrocytes. The sluggishness of adenosine is consistent with the standard view that it is not an immediate signal but a slow messenger of metabolic states.

Response: We invite this Reviewer to carefully evaluate the data illustrated by the main and supplementary figures of the Wu et al. paper (PMID: 36996110). Figures 1j, 1n, S1, S2, and S3 show that electrical stimulation of neuronal activity in acute brain slices of the medial prefrontal cortex or CA1 area of the hippocampus triggers almost instantaneous release of adenosine. Fig S1h illustrates the profile of stimulation-evoked adenosine release with the time constant of 4.4 s.* Rapid neuronal activity-induced increases in the extracellular concentration of adenosine were observed in early studies involving adenosine detection using enzymatic microelectrode biosensors (PMID: 23713028).

*The authors reported that the release of adenosine evoked by optical stimulation of neurons transduced to express ChrimsonR appeared to be slower, but the overall profiles of the GRAB_{ADO} sensor signal changes to optical stimulation were also very different, with initial negative deflections, not observed when electrical stimulations were applied. The paper avoids discussion of the obvious discrepancy between the profiles of very fast adenosine responses triggered by electrical stimulation of neuronal activity and slower responses induced by optical stimulation.

Furthermore, ENT-mediated transport requires an intracellular build-up of adenosine, secondary to ATP depletion. However, neuronal ATP depletion is not observed, except at high frequency stimulation (<https://doi.org/10.1016/j.cell.2013.12.042> and DOI:10.1016/j.cmet.2018.11.005). Recent in vivo imaging showed that neuronal ATP levels in fact increase at the onset of the wake state upon transition from quiet to active awake, as well as following micro-awakening during NREM sleep state, and in all cases was preceded by CBF increase (<https://doi.org/10.1038/s42003-020-01215-6>). The same study showed neuronal ATP decrease during REM sleep, in which adenosine is thought to be released. All this shows the complexity of the involved signaling and puts in question the primary role of adenosine as a straightforward and fast activity signal to astrocytes.

Response: We thank the Reviewer for raising this discussion point. However, the Reviewer's reasoning is inconsistent with the large body of evidence obtained in many laboratories, which indicates that increased neuronal activity is associated with the release of ATP and adenosine into the extracellular space (PMID: 16210541; PMID: 16639550; PMID: 23713028; PMID: 32999463; PMID: 32883833; PMID: 36352228; PMID: 36996110). There is earlier evidence pointing to two sources of activity-dependent adenosine release involving transporter-mediated and astrocytic ATP exocytosis (PMID 23713028). Additionally, there is strong evidence for synaptic ATP release (PMID: 16639550). Together, the data reported in these publications point to several sources of release that contribute to the immediate and sustained increases in the concentration of extracellular adenosine in response to increases in neuronal activity.

Referee #3

The authors have addressed all the points I raised and I have no more critique. Nice study.

We sincerely thank the Reviewer for dedicating their time to evaluating our revised submission and for their highly positive assessment of our work.

Reviewer Reports on the Second Revision:

Referees' comments:

Referee #1 (Remarks to the Author):

The authors have addressed my previous comments and I have no additional clarifications to request.